# Decomposable Neuro Symbolic Regression

## Abstract

Symbolic regression (SR) models complex systems by discovering mathematical expressions that capture underlying relationships in observed data. However, most SR methods prioritize minimizing prediction error over identifying the governing equations, often producing overly complex or inaccurate expressions. To address this, we present a decomposable SR method that generates interpretable multivariate expressions leveraging transformer models, genetic algorithms (GAs), and genetic programming (GP). In particular, our explainable SR method distills a trained "opaque" regression model into mathematical expressions that serve as explanations of its computed function. Our method employs a Multi-Set Transformer to generate multiple univariate symbolic skeletons that characterize how each variable influences the opaque model's response. We then evaluate the generated skeletons' performance using a GA-based approach to select a subset of high-quality candidates before incrementally merging them via a GP-based cascade procedure that preserves their original skeleton structure. The final multivariate skeletons undergo coefficient optimization via a GA. We evaluated our method on problems with controlled and varying degrees of noise, demonstrating lower or comparable interpolation and extrapolation errors compared to two GP-based methods, three neural SR methods, and a hybrid approach. Unlike them, our approach consistently learned expressions that matched the original mathematical structure.

## 1 Introduction

Deep learning-based systems have achieved remarkable success across various domains due to their ability to model complex, nonlinear functions. However, these systems are often described as "opaque" models[1] in the literature, emphasizing the high complexity of the functions they learn and their many required parameters. Their lack of interpretability and traceability poses challenges in applications where understanding the underlying mechanisms is crucial (Linardatos et al., 2021).

In scientific research, particularly in the physical sciences, interpretability is essential for uncovering governing equations that describe observed phenomena (Camps-Valls et al., 2023; Lee & Kumar, 2023). Data-driven methods play a crucial role in this process by identifying patterns and relationships directly from empirical data. Symbolic regression (SR) emerges as a powerful technique, offering an alternative to opaque models for discovering interpretable mathematical representations from data (Filho et al., 2020). Unlike conventional regression techniques that assume predefined model structures, SR searches the space of mathematical expressions to identify compact and accurate equations describing the observed data (Kronberger et al., 2024).

Most existing SR approaches focus primarily on minimizing prediction error rather than extracting the governing equations (Bertschinger et al., 2024). This often leads to expressions that fit the data well but are overly complex and difficult to interpret (La Cava et al., 2021). Thus, recent neural SR techniques (Biggio et al., 2021; Kamienny et al., 2022; Bertschinger et al., 2024) leverage transformer-based models to generate symbolic skeletons or full mathematical expressions. However, they process all variables simultaneously and often fail to capture the functional form between each variable and the system's response correctly [Ref. 1].[2]

To tackle these limitations, we introduce a novel multivariate SR method called **S**ymbolic **R**egression using **T**ransformers, **G**enetic **A**lgorithms, and genetic **P**rogramming (SeTGAP), illustrated in Fig. 1. SeTGAP is

---

[1]This term is preferred over "black-box": acm.org/diversity-inclusion/words-matter
[2]These references are hidden for double-blind review purposes.

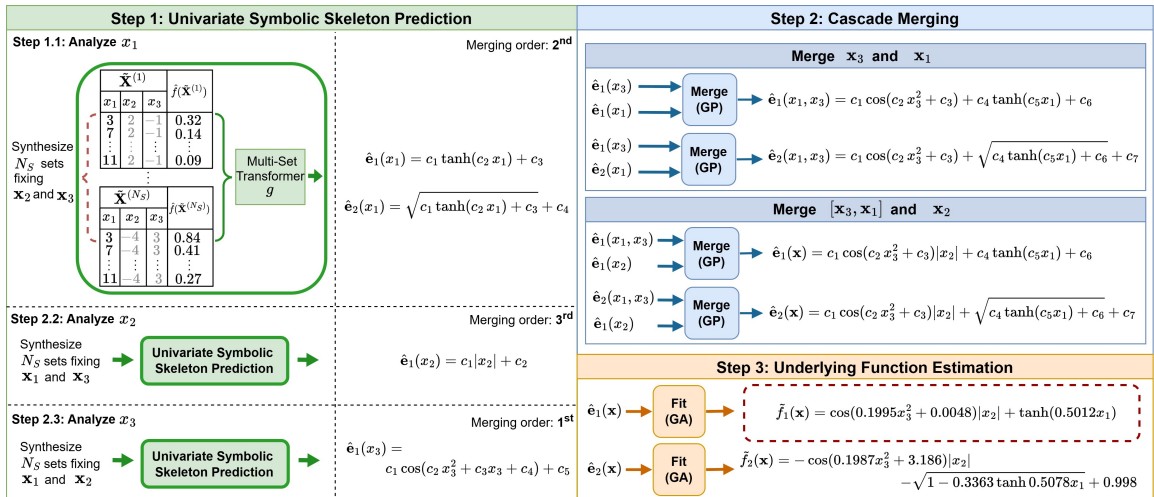

Figure 1: Overview of the SeTGAP symbolic discovery process.

an explainable SR approach that extracts mathematical expressions that explain a trained opaque model's learned function. In Sect. 3.2, we describe how a Multi-Set Transformer generates multiple univariate symbolic skeletons that capture the functional relationships between each independent variable and the opaque model's response. A genetic algorithm (GA)-based selection process then filters out low-quality skeletons, retaining only the most informative ones. In Sec. 3.3, these skeletons are merged through an incremental genetic programming (GP)-based procedure. Sec. 3.3.1 details how, given two skeletons to be merged, a pool of candidate skeleton combinations is produced, ensuring that the merged expressions remain aligned with the original structures. Sec. 3.3.2 explains the selection of the most appropriate combination from the pool of skeleton combinations. Sec. 3.3.3 outlines our cascade approach, which incorporates one variable at a time into the merging process. Finally, Sec. 3.3.4 details a GA-based refinement of the numerical coefficients in the resulting multivariate expression. Our experiments show that SeTGAP consistently recovers the correct functional form of the underlying equations, unlike existing GP-based and neural SR methods, which often fail to do so. In addition, SeTGAP achieves lower or comparable interpolation and extrapolation errors across various problem settings, including scenarios with controlled and varying noise levels.

## 2   Related Work

SR constitutes an NP-hard problem that becomes increasingly complex as the number of observations, operators, and variables increases (Virgolin & Pissis, 2022). As such, brute-force approaches become infeasible. Most SR approaches are based on GP (Kronberger et al., 2024). They evolve candidate solutions, or programs, to identify suitable functional forms and optimize their parameters. This process involves applying genetic operators such as mutation, crossover, and selection to populations of expressions (Cranmer, 2023; Makke & Chawla, 2022; Orzechowski et al., 2018; Schmidt & Lipson, 2009). A candidate's fitness is evaluated based on its ability to minimize prediction error, guiding the evolution toward more accurate models.

GP's flexibility in generating programs of varying lengths allows it to explore a vast search space. One significant drawback is called "code bloat," the uncontrolled growth of program size during evolution. This phenomenon leads to increasingly complex solutions, inflating computational costs and reducing model interpretability (Dignum & Poli, 2007; Poli et al., 2007). The redundancy in the search space further exacerbates this issue, as many solutions represent nearly equivalent functions within the studied domain but differ syntactically (Ebner, 1999; Amir Haeri et al., 2017). This redundancy increases the difficulty of identifying optimal solutions and contributes to GP's bias toward generating larger expressions (Langdon et al., 1999).

Another limitation is that fitness evaluation typically focuses on the overall output of a program, neglecting evaluation of the intermediate components that contribute to the final result. Thus, Arnaldo et al. (2014)

proposed to optimize the generated programs before the selection step by optimizing the fitness contributions of its subexpressions. Conversely, Jiang & Xue (2023) proposed Control Variable Genetic Programming (CVGP), which introduces independent variables into the analysis sequentially through a series of controlled experiments. In each stage, a GP learns simplified expressions by holding all but one variable constant, progressively refining candidate equations as more variables are incorporated. This iterative procedure enables CVGP to recover complex symbolic expressions more reliably than end-to-end approaches.

GP has been combined with deep learning techniques for SR. Mundhenk et al. (2021) introduced a recurrent neural network (RNN) to generate initial GP populations. The best-performing individuals are then used to retrain the RNN. Similarly, Petersen et al. (2021) presented a method that utilizes RNNs to generate mathematical expressions in a reinforcement learning framework. In addition, Udrescu & Tegmark (2020) proposed a recursive algorithm called AI Feynmann that uses NNs to identify properties such as symmetry, separability, and compositionally, which are exploited to define sub-problems that are easier to solve.

Several methods have proposed training NNs and pruning irrelevant parts of them until a simple equation can be distilled from the network weights (Martius & Lampert, 2016; Sahoo et al., 2018; Tsoi et al., 2025; Werner et al., 2021). One of their main limitations is that they do not leverage past experiences, and each problem is learned from scratch. Transformer-based methods have been proposed recently as an alternative. Biggio et al. (2021) presented a Set Transformer-based model (Lee et al., 2019) and pre-trained it on a synthetic dataset of multivariate equations to predict symbolic skeletons from input-output data pairs. During inference, the observed data is used to predict the equation's skeleton using the pre-trained transformer. Next, the constants within the skeleton are optimized using the BFGS algorithm (Fletcher, 1987). To extend this approach, Landajuela et al. (2022) presented a unified deep SR (uDSR) framework that combines multiple SR strategies. It first attempts to decompose the target problem into subproblems using AI-Feynman (Udrescu & Tegmark, 2020). For each subproblem, an RNN-based controller (Petersen et al., 2021) generates candidate mathematical expressions via a neural SR model based on that proposed by Biggio et al. (2021). At each optimization step, candidates are evolved and refined using GP. The refined expressions are fed back to update the RNN. The final solution assembles the subproblem expressions using a linear model.

Alternatively, Kamienny et al. (2022) proposed an end-to-end (E2E) transformer model that directly predicts complete mathematical expressions, bypassing skeleton prediction. Learned constants can be fine-tuned using a non-convex optimizer such as BFGS. Shojaee et al. (2023) introduced Transformer-based Planning for Symbolic Regression (TPSR), a decoding strategy that uses Monte Carlo Tree Search (MCTS) to guide equation generation via lookahead planning. TPSR is model-agnostic and works with any pre-trained transformer-based SR model, enabling optimization for non-differentiable objectives like reward and complexity. Their experiments show that applying TPSR on the E2E model boosts performance on benchmark datasets.

In addition, Bertschinger et al. (2024) proposed an SR neuro-evolution approach that trains a population of transformer models using two objective functions: prediction error and symbolic loss. Finally, in [Ref. 1][2], we noted that SR methods analyzing all variables simultaneously often fail to identify the functional relationships between each variable and the system's response. To address this, we introduced the Multi-Set Transformer, trained on synthetic data to identify symbolic skeletons shared across multiple input-response sets. Then, we proposed a *post-hoc* explainability method that extracts univariate skeletons, providing interpretable approximations of the function learned by an opaque regression model.

## 3 Deep Evolutionary Symbolic Regression

We consider a system with response $y \in \mathbb{R}$ and $t$ explanatory variables $\mathbf{x} = \{x_1, \ldots, x_t\}$. We assume that its underlying function $f(\mathbf{x}) = f(x_1, \ldots, x_t)$ can be constructed using a finite number of unary and binary operations. The response is expressed as $y = f(\mathbf{x}) + \varepsilon_a$, where $\varepsilon_a$ represents the error term due to aleatoric uncertainty. Below, we define the SR problem formally and describe our SeTGAP methodology.

### 3.1 Problem Definition

Let us define our SR problem formally:

**Definition 1.** *Given a dataset $(\mathbf{X}, \mathbf{y})$, with inputs $\mathbf{X} = \{\mathbf{x}_1, \mathbf{x}_2, \ldots, \mathbf{x}_{N_R}\} \subseteq \mathbb{R}^t$ and target responses $\mathbf{y} = \{y_1, y_2, \ldots, y_{N_R}\} \subseteq \mathbb{R}$, the SR problem seeks to discover a function $\tilde{f} : \mathbb{R}^t \to \mathbb{R}$ that approximates the unknown underlying function $f(\mathbf{x})$. The function $\tilde{f}$ is represented as a composition of a finite number of unary and binary operators, such that $\tilde{f}(\mathbf{x}) \approx f(\mathbf{x})$ while capturing its functional structure and behavior.*

Thus, $\tilde{f}(\mathbf{x})$ can be expressed as $\tilde{f}(\mathbf{x}) = f(\mathbf{x}) + \varepsilon_a + \varepsilon_e$, where $\varepsilon_e$ is the error due to epistemic uncertainty, which is attributable to a lack of knowledge about $f$ and can be reduced by acquiring additional information and improving the predictive model. In this context, solving the SR problem implies minimizing $\varepsilon_e$ by selecting a suitable representation for $\tilde{f}(\mathbf{x})$ and identifying an optimal set of parameters $\boldsymbol{\theta}_{\tilde{f}}$ for such representation.

## 3.2 Univariate Symbolic Skeleton Prediction

We deviate from SR approaches that prioritize the minimization of the prediction error of the learned functions. We argue that a correctly identified functional form $\tilde{f}$ inherently leads to a low estimation error and emphasizes interpretability and faithfulness to the underlying system's governing principles. Existing works on Symbolic Skeleton Prediction (SSP) (Bertschinger et al., 2024; Biggio et al., 2021; Petersen et al., 2021; Valipour et al., 2022) attempt to address this issue by generating multivariate symbolic skeletons that describe the behavior of $f$, with numerical coefficients subsequently optimized to minimize prediction error.

### 3.2.1 Background: Multi-Set Skeleton Prediction

Given a mathematical expression, its symbolic skeleton is an expression that replaces the numerical constants with placeholders. For example, if $f(\mathbf{x}) = 5 \log x_1 (\sin(x_2^2) + 1) - 4$, its skeleton is expressed as $\mathbf{e}(\mathbf{x}) = \kappa(f(\mathbf{x})) = c_1 \log x_1 (\sin(x_2^2) + c_2) + c_3$, where $\kappa(\cdot)$ represents the skeleton function and $c_i$ are placeholders. In prior work [Ref. 1][2], we showed that SSP methods struggle to identify the correct functional form of all variables in a system. Thus, we introduced a univariate skeleton prediction method that produces univariate skeletons explaining the relationships between each variable and the system's response. Following the previous example, its univariate skeletons are: $\mathbf{e}(x_1) = \kappa(f(\mathbf{x}); x_1) = c_4 (\log x_1) + c_5$, and $\mathbf{e}(x_2) = \kappa(f(\mathbf{x}); x_2) = c_6 \sin(x_2^2) + c_7$. Here, $\kappa(\cdot; x_v)$ considers the remaining variables $\mathbf{x} \setminus x_v$ irrelevant when describing the functional form between $x_v$ ($v \in [1, \ldots, t]$) and the system's response. In this case, the placeholders $c_i$ may represent numeric constants or functions of other variables.

This method uses a trained regression model $\hat{f}$ (e.g., a neural network) that approximates the underlying function $f$ by capturing the association between $\mathbf{X}$ and $\mathbf{y}$. Then, each variable of the system, $x_v$, is analyzed separately. To do this, $N_s$ artificial sets of input–response pairs $\{(\tilde{\mathbf{X}}^{(1)}, \tilde{\mathbf{y}}^{(1)}), \ldots, (\tilde{\mathbf{X}}^{(N_s)}, \tilde{\mathbf{y}}^{(N_s)})\}$ are generated. To isolate the influence of variable $x_v$, the set $\tilde{\mathbf{X}}^{(s)}$ ($s \in [1, \ldots, N_s]$) is constructed such that the variable $x_v$ (i.e., the $v$-th column $\tilde{\mathbf{X}}_v^{(s)}$) is allowed to vary while the other variables are fixed to random values. Thus, $\tilde{\mathbf{y}}^{(s)}$ denotes the estimated response of inputs $\tilde{\mathbf{X}}^{(s)}$ using the trained model $\hat{f}$ as $\tilde{\mathbf{y}}^{(s)} = \hat{f}(\tilde{\mathbf{X}}^{(s)})$.

Fixing the columns corresponding to the variables $\mathbf{x} \setminus x_v$ ensures that $\tilde{\mathbf{y}}^{(s)}$ depends only on the column $\tilde{\mathbf{X}}_v^{(s)}$. However, this process may project the function into a space where its functional form becomes less recognizable. To address this, the influence of $x_v$ on the system's response is analyzed using $N_s$ sets of input–response pairs, each reflecting a different effect of the variables $\mathbf{x} \setminus \{x_v\}$. As such, each set $(\tilde{\mathbf{X}}^{(s)}, \tilde{\mathbf{y}}^{(s)})$ is generated independently by fixing $\mathbf{x} \setminus \{x_v\}$ at different values. The relationship between each $\tilde{\mathbf{X}}_v^{(s)}$ and $\tilde{\mathbf{y}}^{(s)}$ can be described by a univariate function $f_v^{(s)}$. Note that functions $f_v^{(1)}, \ldots, f_v^{(N_S)}$ have been derived from the same function $f(\mathbf{x})$ and should share the same symbolic skeleton $\mathbf{e}(x_v)$, which is unknown. The task of predicting a skeleton $\hat{\mathbf{e}}(x_v)$ that describes the shared function form of input $\tilde{\mathbf{D}}_v = \{\tilde{\mathbf{D}}_v^{(1)}, \ldots, \tilde{\mathbf{D}}_v^{(N_s)}\}$ (i.e., $\hat{\mathbf{e}}(x_v) \approx \mathbf{e}(x_v)$), s.t. $\tilde{\mathbf{D}}_v^{(s)} = (\tilde{\mathbf{X}}_v^{(s)}, \tilde{\mathbf{y}}^{(s)})$, is known as multi-set symbolic skeleton prediction (MSSP).

The MSSP problem has been addressed in [Ref. 1][2] by designing a Multi-Set Transformer. The model's function and parameters are denoted by $g(\cdot)$ and $\Theta$, respectively. It was trained on a large dataset of synthetically generated MSSP problems to produce accurate estimated skeletons. Given an input collection $\tilde{\mathbf{D}}_v$, the estimated skeleton obtained for variable $x_v$ is computed as $\tilde{\mathbf{e}}(x_v) = g(\tilde{\mathbf{D}}_v, \Theta)$. Key details about the model architecture, training procedure, and vocabulary are provided in Appendix A.

---

**Algorithm 1** Univariate Skeleton Generation

---

1: **function** GENERATEUNIVSKS($v, n, N_s, \hat{f}, g, n_{\text{cand}}, n_B$)
2:     $\text{genSks}_v \leftarrow []$
3:     **for** each $i \in (1, n_{\text{cand}})$ **do**
4:         $\tilde{\mathbf{D}}_v \leftarrow \texttt{generateCollection}(v, n, N_s, \hat{f})$
5:         $\text{genSks}_v.\texttt{append}(g(\tilde{\mathbf{D}}_v, \Theta; n_B))$
6:     $\text{genSks}_v \leftarrow \text{removeDuplicates}(\text{genSks}_v)$                          $\triangleright\ \text{genSks}_v = \{\hat{\mathbf{e}}_1(x_v), \ldots, \hat{\mathbf{e}}_{|\text{genSks}_v|}(x_v)\}$
7:     $\tilde{\mathbf{D}}_v^{(\text{test})} \leftarrow \texttt{generateCollection}(v, n, N_s, \hat{f})$
8:     $\text{corrVals}_v \leftarrow \text{zeros}(|\text{genSks}_v|)$
9:     **for** each $k \in (1, n_{\text{cand}})$ **do**
10:        $\text{corrVals}_v[k] \leftarrow \texttt{fitCoefficients}(\hat{\mathbf{e}}_k(x_v), \tilde{\mathbf{D}}_v^{(\text{test})})$
11:     $\text{genSks}_v \leftarrow \texttt{sortSkeletons}(\text{genSks}_v, \text{corrVals}_v)$
12:     **if** $|\text{genSks}_v| > n_{\text{cand}}$ **then**
13:        $\text{genSks}_v, \text{corrVals}_v \leftarrow \text{genSks}_v[1 : n_{\text{cand}}], \text{corrVals}_v[1 : n_{\text{cand}}]$
14:     **return** $\text{genSks}_v, \text{corrVals}_v$

---

### 3.2.2 Extended Univariate Symbolic Skeleton Prediction

Unlike previous work, we generate up to $n_{\text{cand}}$ distinct candidate skeletons, as described in Algorithm 1. We generate a $\tilde{\mathbf{D}}_v$ collection and feed it into $g$ to obtain $n_B$ skeletons using a diverse beam search (DBS) strategy (Vijayakumar et al., 2018) to promote variability among the $n_B$ generated skeletons. This process is repeated $n_{\text{cand}}$ times, yielding a total of $n_{\text{cand}} n_B$ skeletons. Each repetition yields a new $\tilde{\mathbf{D}}_v$ collection using different combinations of fixed values of $\mathbf{x} \setminus \{x_v\}$, increasing input diversity and potentially leading to different skeletons. Then, we discard identical and some mathematically equivalent skeletons, determined using basic trigonometric rules (see Appendix B). For example, we consider the skeleton $c_1 \sin(c_2 x_v + c_3)$ is equivalent to $c_4 \cos(c_5 x_v + c_6)$ if $c_1 = c_4$, $c_2 = c_5$, and $c_3 = c_6 - \pi/2$. The motivation to identify such equivalences is to avoid redundancy and reduce computational cost. However, this simplification does not affect the final results. A more comprehensive treatment of equivalence identification is left for future work.

If the generated skeleton list, $\texttt{genSks}_v$, exceeds $n_{\text{cand}}$ elements, we evaluate their performance and select the top $n_{\text{cand}}$ candidates. To do this, an additional collection $\tilde{\mathbf{D}}_v$ is generated. Since a skeleton expression for variable $\mathbf{x}_v$ is expected to describe the functional form of all sets in $\tilde{\mathbf{D}}_v$, we choose a random set $\tilde{\mathbf{D}}_v^{(\text{test})} = (\tilde{\mathbf{X}}_v^{(\text{test})}, \tilde{\mathbf{y}}^{(\text{test})}) \subset \tilde{\mathbf{D}}_v$ and use it to fit the coefficients of each skeleton $\hat{\mathbf{e}}_k(x_v)$, where $k \in \{1, \ldots, |\texttt{genSks}_v|\}$. The coefficient fitting problem is described as follows. Let $f_{est}(x_v) = \texttt{setConstants}(\hat{\mathbf{e}}_k(x_v), \mathbf{c})$ be the function obtained when replacing the $n_c$ coefficients in $\hat{\mathbf{e}}_k(x_v)$ with the numerical values in a given set $\mathbf{c} \in \mathbb{R}^{n_c}$. Then, the objective is to find an optimal set $\mathbf{c}^*$ that maximizes the following Pearson correlation:

$$\mathbf{c}^* = \underset{\mathbf{c}}{\operatorname{argmax}} \operatorname{corr}(f_{est}(\tilde{\mathbf{X}}_v^{(\text{test})}), \tilde{\mathbf{y}}^{(\text{test})}).$$

Note that in generating $\tilde{\mathbf{D}}_v$, we fixed the values of $\mathbf{x} \setminus x_v$, so we can assume that all coefficients in $\mathbf{c}$ are numerical values. However, the learned coefficients are then discarded, as they serve only to evaluate the fit of a univariate skeleton to the data. We use the Pearson correlation between $f_{est}(\tilde{\mathbf{X}}_v^{(\text{test})})$ and the estimated response $\tilde{\mathbf{y}}^{(s)}$ instead of mean squared error (MSE) to evaluate the functional suitability of skeleton $\hat{\mathbf{e}}_k(x_v)$. This choice is motivated by the fact that we aim to assess whether the candidate skeleton captures the functional form regardless of scale or vertical shift. Unlike MSE, the correlation coefficient is invariant to linear transformations, thus allowing us to compare skeletons without explicitly optimizing scale and offset parameters, which would otherwise increase computational cost.

This problem is solved using a genetic algorithm (GA) (Holland, 1992). The individuals of our GA are arrays of $n_c$ elements that represent potential $\mathbf{c}$ sets. Then the optimization process is carried out by function $\texttt{fitCoefficients}(\hat{\mathbf{e}}_k(x_v), \tilde{\mathbf{D}}_v^{(\text{test})})$. This optimization process is repeated for all system variables to derive their univariate skeleton expressions with respect to the system's response.

### 3.3 Merging Univariate Symbolic Skeletons

The next step is to merge the univariate skeleton candidates to produce multivariate expressions. This process is carried out in a cascade fashion until a final expression incorporating all variables is formed.

### 3.3.1 Merging Skeleton Expressions

Given two skeleton expressions, multiple mathematically valid ways to combine them may exist. Here, we explore how to generate such combinations. We start with the following proposition:

**Proposition 1.** *Let $f(\mathbf{x})$ be a scalar-valued function defined by a finite composition of real-valued unary and binary operators applied to scalar sub-expressions. Then, $f(\mathbf{x})$ can always be expressed as: $f(\mathbf{x}) = c_0 + \sum_i c_i \prod_j \nu_{i,j}(T_{i,j}(\mathbf{x}))$, where $c_0, c_i \in \mathbb{R}$, $\nu_{i,j}$ is a unary operator (including the identity function $I(f(\mathbf{x})) = f(\mathbf{x})$), and $T_{i,j}(\mathbf{x})$ is a sub-expression. Moreover, each $T_{i,j}(\mathbf{x})$ can be recursively decomposed in the same structure as $f$, continuing until the decomposition reduces to variables or constants.*

Proposition 1 is proved in Appendix C. Then, let $\mathbf{e}_1(\mathbf{x}_S)$ and $\mathbf{e}_2(x_q)$ be two candidate skeletons to be merged. Here, $S$ is an index set, $S \subset \{1, \ldots, t\}$, specifying the variables that $\mathbf{e}_1$ depends on; i.e., $\mathbf{x}_S = \{x_r | r \in S\}$. In contrast, $x_q$ is a variable distinct from those in $\mathbf{x}_S$ ($q \notin S$). Following from Proposition 1, the skeletons can be expressed as $\mathbf{e}_1(\mathbf{x}_S) = c_0 + \sum_i c_i \prod_j \nu_{i,j}(T_{i,j}(\mathbf{x}_S))$ and $\mathbf{e}_2(x_q) = c_0' + \sum_i c_{i'}' \prod_{j'} \nu_{i',j'}'(T_{i',j'}'(x_q))$. Below, we explain how the subtrees of both expressions can be merged recursively.

The key idea is that a constant placeholder in $\mathbf{e}_1(\mathbf{x}_S)$ may be replaced by a subtree of $\mathbf{e}_2(x_q)$, and vice versa. Given two skeletons, $\mathbf{e}_1(\mathbf{x}_S) = c_1 T_1(\mathbf{x}_S)$ and $\mathbf{e}_2(x_q) = c_2 T_2(x_q)$, a straightforward way to merge them is by recognizing that part of $c_1$ may be a function of $x_q$, while part of $c_2$ may be a function of $\mathbf{x}_S$. Thus, their combination, denoted by the operation $\bowtie$, is given by $\mathbf{e}_3(\mathbf{x}_S \cup x_q) = \mathbf{e}_1(\mathbf{x}_S) \bowtie \mathbf{e}_2(x_q) = c_3(c_4 + T_1(\mathbf{x}_S))(c_5 + T_2(x_q))$. Expanding this expression conforms to the expected functional form stated in Proposition 1. Note that the skeletons with respect to the corresponding initial variable sets remain unchanged; that is, $\kappa(\mathbf{e}_1(\mathbf{x}_S); \mathbf{x}_S) = \kappa(\mathbf{e}_3(\mathbf{x}_S \cup x_q); \mathbf{x}_S) = c_1 T_1(\mathbf{x}_S)$ and $\kappa(\mathbf{e}_2(x_q); x_q) = \kappa(\mathbf{e}_3(\mathbf{x}_S \cup x_q); x_q) = c_q T_2(x_q)$.

Applying the same principle to a more general case in which $\mathbf{e}_1(\mathbf{x}_S) = c_1 + c_2 \prod_j T_{1,j}(\mathbf{x}_S)$ and $\mathbf{e}_2(x_q) = c_3 + c_4 \prod_{j'} T_{2,j'}(x_q)$, we obtain $\mathbf{e}_3(\mathbf{x}_S \cup x_q) = \mathbf{e}_1(\mathbf{x}_S) \bowtie \mathbf{e}_2(x_q) = c_5 + c_6 \prod_j (c_{7,j} + T_{1,j}(\mathbf{x}_S)) \prod_{j'} (c_{8,j'} + T_{2,j'}(\mathbf{x}_S))$. This case is referred to as the "wrapped–product merge" for future reference. Nevertheless, more combinations are possible if the candidate skeletons share functions with compatible mathematical structures. For example, if $\mathbf{e}_1(x_1, x_2) = c_1 \sin(c_2 x_1 x_2 + c_3)$ and $\mathbf{e}_2(x_3) = c_4 \sin(c_5 x_3 + c_6)$, we could obtain $\mathbf{e}_3(x_1, x_2, x_3) = \mathbf{e}_1(x_1, x_2) \bowtie \mathbf{e}_2(x_3) = c_7(c_8 + \sin(c_9 x_1 x_2 + c_{11}))(c_{12} + \sin(c_{13} x_3 + c_{14}))$, as shown before, but also $\mathbf{e}_3(x_1, x_2, x_3) = c_{15} \sin(c_{16} x_1 x_2 + c_{17} x_3)$ and $\mathbf{e}_3(x_1, x_2, x_3) = c_{18} \sin(c_{19} x_1 x_2 x_3 + c_{20})$, which yield to the same skeletons with respect to the initial variable sets.

Algorithm 2 implements our recursive merging procedure, $\mathtt{merge}(\mathbf{ex}_1, \mathbf{ex}_2)$, which takes as inputs two skeleton expressions $\mathbf{ex}_1$ and $\mathbf{ex}_2$, and constructs multivariate skeleton candidates by respecting the decomposition guaranteed by Proposition 1. The high-level invariant maintained throughout the recursion is that every intermediate expression is represented in a form consistent with the proposition; i.e., as a constant plus a sum of terms, each being a product of sub-expressions affected by unary operators. The procedure begins by extracting the immediate subtrees (the summands or multiplicative factors) of each input expression using $\mathtt{getSubtreesLists}(\mathbf{ex}_1, \mathbf{ex}_2)$, which returns the subtree lists ordered so that $\mathbf{exShort}$ contains fewer or equal elements than $\mathbf{exLong}$. The lists are shuffled to introduce diversity in the merging order. If $\mathbf{exShort}$ lacks an explicit constant term, the routine appends a constant placeholder and ensures it is the last element; this placeholder enables absorption of unmatched terms and enforces the proposition's canonical form.

When both inputs are sums (Line 4), the algorithm iterates the subtrees (i.e., summands) in $\mathbf{exShort}$, attempting to merge each with compatible subtrees from $\mathbf{exLong}$; i.e., subtrees that share the same unary or binary operator. The compatibility check is performed by $\mathtt{findComp}$, and a subset of matching subtrees is randomly selected for merging, as depicted in Fig. 2. A random subset $\mathbf{selectedArgs}$ of these compatible subtrees is chosen (Line 10). If $\mathbf{selectedArgs}$ is empty, the loop continues; if it contains a single element, that element is merged recursively with the current summand, $\mathbf{exShort}[i]$, via $\mathtt{merge}$. However, if multiple subtrees are selected, the only way to maintain the form required by Proposition 1 is to treat their sum as a single, indivisible block, effectively a constant term with respect to $x_q$, which is then multiplied by $\mathbf{exShort}[i]$ (Line 16). The last element of $\mathbf{exShort}$ (the constant placeholder) then absorbs any remaining unmatched summands from $\mathbf{exLong}$, again preserving the canonical structure (Line 7).

If neither expression is a sum or a product but they share the same outer operator, we keep that operator and merge their inner arguments recursively. This is shown in Line 20, where $\mathtt{ex1.func}$ is the SymPy

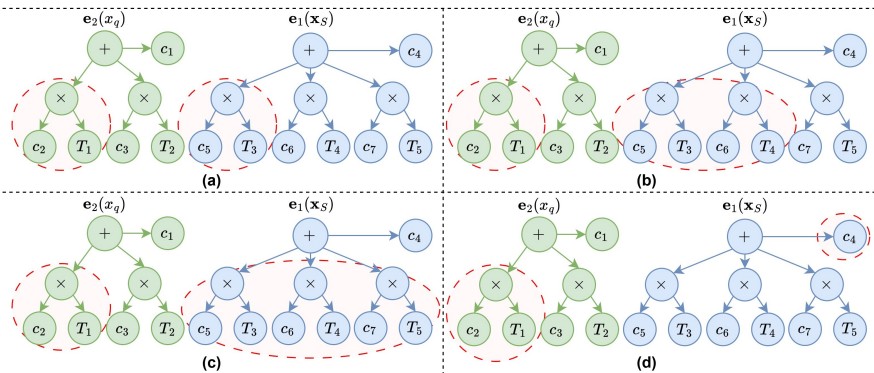

Figure 2: Example of a selected subtree of $\mathbf{e}_2(x_q)$ within a sum merging with one or more subtrees from $\mathbf{e}_1(\mathbf{x}_S)$, illustrating four out of the nine possible cases.

attribute that returns the outermost operator of `ex1`, and `ex1.args` returns its tuple of inner arguments. The call `ex1.func(...)` constructs a new expression by applying this same operator to the result of merging the arguments `ex1.args` and `ex2.args`. The `merge` call is recursive because each argument can itself be a composite expression, and the same merging logic can be applied at deeper levels of the expression tree.

If both `ex1` and `ex2` are multiplications (Line 22), we distinguish two subcases. If `isAllSymbols(exShort)` is true, each factor of `exShort` represents a symbol, and the merged expression is built as wrapped–product merge, $c_1 \prod_i (c_{2,i} + \text{exShort}[i]) \prod_j (c_{3,j} + \text{exLong}[j])$ (Line 24), which fits into the topology of Proposition 1. If non-symbol factors are present, the algorithm attempts one-to-one merges between factors. It iterates through the factors of `exShort`, uses `findComp` to locate compatible candidates in `exLong` (Line 31), selects a single candidate uniformly at random (Line 33), and accepts the recursive merge with probability 0.5 to introduce variability (Line 34–36). When analyzing the last `exShort` factor, Line 29 implements the absorption of any remaining unmatched `exLong` factors using a constant-wrapped product. Otherwise, if `exLong` has no remaining factors, the merged expression is the product of the updated `exShort` factors.

If the outermost operators differ (Line 40), the merged expression is the product $\text{ex}_1 \text{ex}_2$. This preserves the canonical form because the product of two canonical sub-expressions expands into a sum of products of unary-applied subexpressions (see Appendix C), as required by Proposition 1. Finally, the function returns the merged expression, which integrates elements from both inputs while preserving their initial structures. An illustrative example of the merging process, including its intermediate steps, is provided in Appendix D.

### 3.3.2 Selecting Combined Skeleton Expressions

Algorithm 2 generates a single combination of skeletons. Here, we extend this process to construct a population of candidate skeletons and select the top-performing one. Thus, we employ an evolutionary strategy to assess the performance of each skeleton combination. The population of skeletons, `candSks`, is constructed by repeatedly applying Algorithm 2. Since the merging process is stochastic, each application of the algorithm may yield a different skeleton combination. The process continues until a population size $P_{\max}$ is reached or a patience criterion is met. As such, if no new valid skeleton is found after a certain number of attempts, the generation process halts, assuming that all viable combinations have been explored.

We aim to identify the most promising combination in `candSks` by evaluating how well each skeleton can fit the test data. Algorithm 3 provides a high-level overview of this process. We generate test data $\tilde{\mathbf{D}}^{(\text{test})} = (\tilde{\mathbf{X}}^{(\text{test})}, \tilde{\mathbf{y}}^{(\text{test})})$. Unlike in Sec. 3.2, here the columns in $S' = \{S \cup q\}$ (i.e., the indices of variables present in the combined skeletons) are allowed to vary, while all other variables are fixed to random values. The set $\tilde{\mathbf{D}}_{S'}^{(\text{test})} = (\tilde{\mathbf{X}}_{S'}^{(\text{test})}, \tilde{\mathbf{y}}^{(\text{test})})$ is then used for evaluation. Each skeleton in `candSks` is replicated `rep` times, with randomly assigned coefficient values, to form an initial pool of candidate expressions `Pop`.

The population then undergoes an evolution process. In each iteration, the fitness of every expression in `Pop` is evaluated using `evalCorr`, which computes the Pearson correlation between $\tilde{\mathbf{y}}^{(\text{test})}$ and the output produced

---

**Algorithm 2** Recursive Skeleton Merging

---

1: **function** MERGE($ex_1, ex_2$)
2:     exShort, exLong ← **getSubtreesLists**($ex_1, ex_2$)                    ▷ Return lists of subtrees in sum
3:     exShort.**shuffle**(), exLong.**shuffle**()
4:     **if** isSum($ex_1$) **and** isSum($ex_2$) **then**
5:         **for** each $i \in (1, |\text{exShort}|)$ **do**
6:             **if** $i = |\text{exShort}|$ **and** $|\text{exLong}| > 0$ **then**
7:                 exShort[$i$] ← exShort[$i$] + $\sum_j$ exLong[$j$]
8:             **else**
9:                 args ← **findComp**(exLong, exShort[$i$])                    ▷ Find exLong args compatible w/exShort[$i$]
10:                selectedArgs ← **sample**(args, randInt(0, |args|))
11:                **if** $|\text{selectedArgs}| = 0$ **then continue**
12:                exLong.**remove**(selectedArgs)
13:                **if** $|\text{selectedArgs}| = 1$ **then**
14:                    exShort[$i$] ← **merge**(exShort[$i$], selectedArgs[0])
15:                **else**
16:                    exShort[$i$] ← exShort[$i$] $\sum_j$ selectedArgs[$j$]
17:         mergedEx ← $\sum_i$ exShort[$i$]
18:     **else**
19:         **if** $ex_1$.**func** $= ex_2$.**func then**
20:             **if** !isMult($ex_1$) **then**                              ▷ If compatible, merge inner arguments
21:                 mergedEx ← $ex_1$.**func**(merge($ex_1$.**args**, $ex_2$.**args**))
22:             **else**
23:                 **if** isAllSymbols(exShort) **then**
24:                     mergedEx ← $c_1 \prod_i (c_{2,i} + \text{exShort}[i]) \prod_j (c_{3,j} + \text{exLong}[j])$
25:                 **else**
26:                     mergedEx ← **null**
27:                     **for** each $i \in (1, |\text{exShort}|)$ **do**
28:                         **if** $i = |\text{exShort}|$ **and** $|\text{exLong}| > 0$ **then**
29:                             mergedEx ← $c_1 \prod_{i'=0}^{i} (c_{2,i'} + \text{exShort}[i']) \prod_j (c_{3,j} + \text{exLong}[j])$
30:                         **else**
31:                             args ← **findComp**(exLong, exShort[$i$])
32:                             **if** $|\text{args}| = 0$ **then continue**
33:                             selectedArg ← **choice**(args)
34:                             **if** random(0, 1) < 0.5 **then continue**
35:                             exLong.**remove**(selectedArg)
36:                             exShort[$i$] ← **merge**(exShort[$i$], selectedArg)
37:                     **if** mergedEx is **null then**
38:                         mergedEx ← $\prod_i$ exShort[$i$]
39:         **else**
40:             mergedEx ← $ex_1 \, ex_2$
41:     **return** mergedEx

---

**Algorithm 3** Skeleton Combination with Genetic Programming

---

1: **function** SELECTCOMBINATION(candSks, rep, maxG, $\tilde{\mathbf{D}}_{S'}^{(\text{test})}$)
2:     Pop ← [ ]
3:     **for** each $s \in (1, |\text{candSks}|)$ **do**
4:         exps ← [ ]
5:         **for** $i = 1$ to rep **do** exps.**append**(assignValues(candSks[$s$]))
6:         Pop.**append**(exps)
7:     **for** each gen $\in (1, \text{maxG})$ **do**
8:         fitnesses, bestFitnesperSk ← **evalCorr**(Pop, $\tilde{\mathbf{D}}_{S'}^{(\text{test})}$)
9:         **for** each $p \in (1, |\text{Pop}|)$ **do**
10:            Pop[$p$] ← **evolveSks**(Pop[$p$], fitnesses)    ▷ Expressions derived from the same skeleton combination are evolved together
11:    **return** candSks[**argmax**(bestFitnesperSk)], **max**(bestFitnesperSk)

---

by the expression when evaluated on $\tilde{\mathbf{X}}^{(\text{test})}$. After the evaluation, `evolveSks` updates each subpopulation in `Pop` independently. As such, expressions derived from the same skeleton combination are evolved together, and no crossover is allowed between expressions coming from different skeletons. This grouping enforces the structural constraint because crossover is implemented as an exchange of coefficient vectors. In other words, when two individuals are selected for crossover, their skeleton structure is kept fixed and only their numeric coefficient assignments are swapped. Likewise, mutation acts only on coefficients, so the expression trees remain unchanged throughout evolution. Finally, after `maxG` generations, the algorithm returns the skeleton whose evolved instances achieved the highest fitness. This approach bears similarities to GP, with the distinction that the search for viable skeletons is decoupled from the evolutionary optimization process. Rather than evolving the expressions dynamically, as in standard GP, we enumerate all admissible skeleton combinations via Algorithm 2 and then apply our constrained evolution, as described in Algorithm 3, where the structure of all mathematical expressions is fixed to the precomputed combined skeletons.

### 3.3.3 Cascade Merging

The construction of multivariate skeletons follows an incremental merging process in which univariate skeleton candidates are combined progressively. For each variable $x_v$ Algorithm 1 generates up to $n_{\text{cand}}$ candidate skeletons $\texttt{genSks}_v$ along with their corresponding correlation values $\texttt{corrVals}_v$. To determine the merging order, the variables are ranked according to the maximum correlation value obtained across their generated skeletons. Variables with higher correlation scores are prioritized in the merging order, as their generated skeletons exhibit stronger relationships with the data, reducing the risk of propagating structural uncertainty.

Let $S$ denote the indices of the variables whose skeletons have already been merged. Initially, $S$ contains only the index of the variable that corresponds to the skeleton with the highest correlation value. At each iteration, a new variable $x_q$ is selected according to the ranking, and its univariate skeletons are merged with those corresponding to $S$. Specifically, Algorithms 2 and 3 are applied to combine each skeleton generated for $\mathbf{x}_S$ with each skeleton generated for $x_q$, producing at most $n_{\text{cand}}^2$ skeleton combinations. Since Algorithm 3 returns both the selected merged skeleton and its fitness, we apply a greedy selection strategy, retaining only the $n_{\text{cand}}$ highest-performing skeletons at each step. This process repeats until skeletons incorporating all $t$ variables have been generated, which serve as interpretations that describe the functional form of model $\hat{f}$.

We argue that our cascade approach ensures a structured integration of variables into the final multivariate skeleton. Directly constructing a multivariate skeleton from all univariate candidates would prioritize minimizing overall prediction error, potentially obscuring individual contributions and leading to overfitting. Instead, incorporating one variable at a time allows for a more interpretable learning process, where each newly added variable's effect is evaluated in the context of the previously analyzed ones.

It is worth pointing out that our cascade-fashion approach shares similarities with the CVGP method proposed by Jiang & Xue (2023). CVGP analyzes the $t$ system variables in a fixed order: $x_1, x_2, \ldots, x_t$. First, multiple datasets (trials) are generated where only $x_1$ varies while others remain fixed. A GP algorithm extracts a skeleton involving $x_1$. Then, new trials are generated where $x_1$ and $x_2$ vary, and the previously discovered skeleton is fixed while GP extends it to include both variables. This process repeats until all variables are included. Our approach differs in both motivation and implementation. SeTGAP starts by independently identifying univariate skeletons for each variable, focusing on interpreting the individual functional relationships learned by the opaque model $\hat{f}$. Once fixed, it aims to discover mathematically valid ways to merge them into coherent multivariate expressions. In contrast, CVGP builds successive expressions atop a single skeleton for $x_1$. If this initial skeleton is poorly fitted, subsequent expressions tend to compensate for the misspecification rather than recover accurate functional relationships for the remaining variables.

### 3.3.4 Underlying Function Estimation

We utilize the $N_e$ multivariate skeletons $\hat{\mathbf{e}}_1(\mathbf{x}), \ldots, \hat{\mathbf{e}}_{N_e}(\mathbf{x})$ produced by our cascade merging procedure, where $N_e \leq n_{\text{cand}}$. The goal is to construct functions $\tilde{f}_i(\mathbf{x})$ that approximate the underlying function $f(\mathbf{x})$ using the corresponding skeleton $\hat{\mathbf{e}}_i(\mathbf{x}), \forall i \in (1, \ldots, N_e)$. This defines a coefficient fitting problem similar to the one presented in Sec. 3.2. Unlike earlier sections, we minimize the MSE of the response calculated by evaluating $\tilde{f}_i(\mathbf{x})$ on the original dataset $(\mathbf{X}, \mathbf{y})$. This is feasible because $\hat{\mathbf{e}}(\mathbf{x})$ contains all system variables and there is no need to generate a synthetic set of estimated points using $\hat{f}$. Hence, the new coefficient fitting problem consists of finding an optimal set of coefficient values that minimizes the prediction MSE:

$$\mathbf{c}^* = \underset{\mathbf{c}}{\text{argmin}} \, \frac{1}{|\mathbf{X}|} \sum_{(\mathbf{x}_j, y_j) \in (\mathbf{X}, \mathbf{y})} \left( \texttt{setConstants}(\hat{\mathbf{e}}_i(\mathbf{x}_j), \mathbf{c}) - y_j \right)^2$$

such that $\tilde{f}_i(\mathbf{x}) = \texttt{setConstants}(\hat{\mathbf{e}}_i(\mathbf{x}), \mathbf{c}^*)$. This optimization is carried out using the GA-based function $\texttt{fitCoefficients}(\hat{\mathbf{e}}_i(\mathbf{x}), (\mathbf{X}, \mathbf{y}))$ introduced in Sec. 3.2 but modified to minimize MSE.

## 4 Experimental Results

We evaluated SeTGAP on 13 synthetic SR problems derived from equations inspired by previous work. Table 1 lists these equations along with their domain ranges. The 13 synthetic SR problems were chosen to

Table 1: Equations used for experiments

| Eq. | Underlying equation | Reference | Domain range |
|-----|---------------------|-----------|--------------|
| E1 | $(3.0375 x_0 x_1 + 5.5 \sin(9/4(x_0 - 2/3)(x_1 - 2/3)))/5$ | Jin et al. (2020) | $[-5, 5]^2$ |
| E2 | $5.5 + (1 - x_0/4)^2 + \sqrt{x_1 + 10} \sin(x_2/5)$ | [Ref. 1][2] | $[-10, 10]^2$ |
| E3 | $(1.5 e^{1.5 x_0} + 5 \cos(3 x_1))/10$ | Jin et al. (2020) | $[-5, 5]^2$ |
| E4 | $((1 - x_0)^2 + (1 - x_2)^2 + 100(x_1 - x_0^2)^2 + 100(x_3 - x_2^2)^2)/10000$ | Rosenbrock-4D | $[-5, 5]^4$ |
| E5 | $\sin(x_0 + x_1 x_2) + \exp(1.2 x_3)$ | [Ref. 1][2] | $x_1 \in [-10, 10], x_2 \in [-5, 5],$ $x_3 \in [-5, 5], x_4 \in [-3, 3]$ |
| E6 | $\tanh(x_0/2) + |x_1| \cos(x_2^2/5)$ | [Ref. 1][2] | $[-10, 10]^3$ |
| E7 | $(1 - x_1^2)/(\sin(2\pi x_0) + 1.5)$ | Werner et al. (2021) | $[-5, 5]^2$ |
| E8 | $x_0^4/(x_0^4 + 1) + x_1^4/(x_1^4 + 1)$ | Trujillo et al. (2016) | $[-5, 5]^2$ |
| E9 | $\log(2 x_1 + 1) - \log(4 x_0^2 + 1)$ | Trujillo et al. (2016) | $[0, 5]^2$ |
| E10 | $\sin(x_0 e^{x_1})$ | Bertschinger et al. (2023) | $x_1 \in [-2, 2], x_2 \in [-4, 4]$ |
| E11 | $x_0 \log(x_1^4)$ | Bertschinger et al. (2023) | $[-5, 5]^2$ |
| E12 | $1 + x_0 \sin(1/x_1)$ | Bertschinger et al. (2023) | $[-10, 10]^2$ |
| E13 | $\sqrt{x_0} \log(x_1^2)$ | Bertschinger et al. (2023) | $x_1 \in [0, 20], x_2 \in [-5, 5]$ |

cover a range of functional forms and difficulties. Equations E1, E3, E4, E7, E8, and E9 correspond to expressions frequently used in prior SR studies (Trujillo et al., 2016; Jin et al., 2020; Werner et al., 2021), while E10–E13 were adapted to the multivariate setting from the suite proposed by Bertschinger et al. (2023). We also included E2, E5, and E6 [Ref. 1][2] to increase the proportion of non-separable problems, a class where neural SR approaches have been observed to struggle. All equations were evaluated over extended input ranges (e.g., $[-5, 5]$ and $[-10, 10]$) rather than the narrow domains commonly used in earlier works, thereby increasing problem difficulty. The datasets generated from these equations consisted of 10,000 points and each variable was sampled according to a uniform distribution. Appendix H presents the results of SeTGAP evaluated on functions from the SRBench++ benchmark (de Franca et al., 2025).

The regression models $\hat{f}$ tested by SeTGAP are assumed to be opaque functions trained to approximate the target response. In our experiments, we implemented $\hat{f}$ as feedforward neural networks whose architectures were tuned independently for each problem to minimize MSE, as is standard in regression tasks. Different problems vary in complexity, and accordingly, the number of hidden layers. They featured three hidden layers for problem E2, five for E1, E4, E5, and E7, and four for all other cases, with each layer containing 500 ReLU-activated nodes. Tuning these models is not part of SeTGAP's process, but ensures that $\hat{f}$ provides a reasonable approximation of the underlying function. In addition, the Multi-Set Transformer $g$, used to infer univariate skeletons, follows the design and hyperparameter settings established in [Ref. 1][2]. Their generated collections $\tilde{\mathbf{D}}_v$ consist of $N_S$ sets of $n = 3000$ elements, with these parameters tuned in prior work.

For SeTGAP, the hyperparameters in Algorithm 1 include the beam size $n_B$ and the number of univariate skeleton candidates $n_{\text{cand}}$. We set $n_B = 3$ and $n_{\text{cand}} = 3$, as higher values did not yield more distinct skeletons across all problems. The GAs in Secs. 3.2 and 3.3 share the same configuration but differ in loss functions: the former maximizes Pearson correlation, while the latter minimizes MSE. The GA runs with a population of 500 individuals and terminates when the objective function change remains below $10^{-6}$ for 30 consecutive generations. It uses tournament selection, binomial crossover, and generational replacement. Algorithm 3 uses the number of instance expressions per candidate skeleton combination `rep` and the maximum number of generations `maxG`, with values set to `rep` = 150 and `maxG` = 300. In addition, the initial population `candSks` was generated with a maximum size $P_{max} = 5000$, though none of the cases reached this limit. This setup was chosen for its consistent and effective optimization results across all problems. Please refer to Appendix E for more results on hyperparameter tuning.

For each problem, SeTGAP generates up to $n_{\text{cand}}$ multivariate expressions, but for brevity, we report only the one with the lowest MSE. These results are compared against expressions obtained from two GP-based methods, PySR (Cranmer, 2023) and TaylorGP (He et al., 2022), three neural SR approaches, NeSym-ReS (Biggio et al., 2021), E2E (Kamienny et al., 2022), and TPSR (Shojaee et al., 2023), and a hybrid approach, uDSR (Landajuela et al., 2022). The comparison did not include the SR neuro-evolution method proposed by Bertschinger et al. (2024), as training and evolving their large transformer architectures in a multivariate setting is computationally prohibitive. NeSymReS could not be applied to E4 and E5 due to its limitation to systems with at most three variables. The only hyperparameters tuned were the beam size and the number of BFGS restarts, both initially set to 5 and 4, respectively. However, it was found that a beam

Table 2: Comparison of predicted equations (E1—E4) with rounded numerical coefficients — Iteration 1

| Method | E1 | E2 | E3 | E4 |
|---|---|---|---|---|
| PySR | $0.61x_0x_1$ | $0.41x_2 + ||x_0 - 3.51| - 1.95| + 4.11$ | $0.34e^{x_0}|\sinh(0.47x_0)|$ | $0.21x_0^2 - 0.18x_1 + 0.21x_2^2 - 0.18x_3 - 0.76$ |
| TaylorGP | $0.64x_0x_1$ | $-0.5x_0 + 0.001x_1 + 0.39x_2 + 8.62$ | $0.23x_0e^{x_0}$ | $0.29x_0^2$ |
| NeSymReS | $0.59x_0x_1 + \cos(0.01(x_1 - x_0 - 0.08)^2)$ | $-x_0 + 0.40x_2 + e^{e^{-0.001x_1}} + 5.88$ | $9.10e^{0.72x_0}\cos(0.15x_1)$ | — |
| E2E | $1.08(0.56x_0x_1 - 0.03x_0 + 0.02x_1 - \sin(0.01x_2^2 + 8.6x_0 + 0.45) - 0.01)$ | $0.06x_0^2 - 0.51x_0 - 0.22x_1\cos(0.18x_2 + 1.43) - 0.01x_1 + 0.01x_2 - 3.25\cos(0.18x_2 + 1.43) + 6.56$ | $0.14e^{1.52x_0} + 0.52\cos(3.45x_1 + 0.05) + 0.11$ | $0.001|8.99(-0.88x_1 + (x_0 + 0.01)^2 + 0.62)^2 + 9.72(-x_3 + 0.98(x_2 + 0.01)^2 + 0.01)^2| + 0.0023$ |
| uDSR | $(-0.006x_0^2 - 0.001x_0x_1^2 + 0.61x_0x_1 + 0.001x_0 - 0.001x_1^2 + 0.031)\cos(e^{-4x_1^2 + x_1})$ | $0.063x_0^2 - 0.501x_0 + 0.029x_1x_2 - 0.001x_2^3 + 0.351x_2 + \sin(0.25x_2) + 6.498$ | $0.013x_0^4 - 0.032x_0^3 - 0.001x_0^2x_1 - 0.41x_0^2 + 0.001x_0x_1^2 + 0.002x_0x_1 - 0.902x_1^2 - 0.087x_1^4 - 0.001x_1^3 + 0.397x_1^2 + 0.006x_1 + e^{x_0} - \cos(x_1) + \cos(2x_1) - 0.477$ | $0.01x_0^4 - 0.02x_0^2x_1 + 0.01x_1^2 + 0.01x_2^4 - 0.02x_2^2x_3 + 0.01x_3^2$ |
| TPSR | $0.146x_1(4.161x_0 - 0.071) - 0.036\sin(5.687x_1 - 1.848) - 0.002$ | $\left(0.014 + \frac{1}{x_0 + 14.596}\right)(0.039x_0 + 3.247)\left(-0.059x_1 - 83.999 + \frac{0.061}{x_2}\right)(0.028x_2 - 0.409\arctan(0.14474x_2 + 0.0015) - 0.334)$ | $0.15e^{1.5x_0} + 0.5\cos(3.0x_1)$ | $0.175x_1 - 0.124x_3 + 0.3626 + (8.858 - 5.843x_1)(0.001x_1 + 0.073) + 0.0016\left(x_0 + 0.595(0.944x_0 - 1)^2\right)^3$ |
| SeTGAP | $0.61x_0x_1 + 1.15\sin((2.24x_0 - 1.5)(x_1 - 0.68))$ | $0.06x_0^2 - 0.5x_0 + (3.37\sqrt{0.1x_1 + 1} - 0.19)(\sin(0.2x_2) + 0.01) + 6.49$ | $0.15e^{1.5x_0} + 0.5\sin(3x_1 - 4.71)$ | $0.01x_0^4 - 0.02x_0^2x_1 - 0.001x_0 + 0.01x_1^2 + 0.01x_2^4 + 0.01x_3^2 - (0.02x_2^2 + 0.004)(x_3 - 0.11) - 0.02$ |

Table 3: Comparison of predicted equations (E5—E8) with rounded numerical coefficients — Iteration 1

| Method | E5 | E6 | E7 | E8 |
|---|---|---|---|---|
| PySR | $e^{1.2x_3} + \sin(x_0 + x_1x_2)$ | $\tanh(e^{x_2})$ | $(0.56 - 0.59x_0^3)/(\sinh(\sinh(\tanh(e^{\sinh(\sin(6.28x_1))}))))$ | $(\tanh(\cosh(x_0) - 1.04) + \tanh(\cosh(x_1) - 1.04))$ |
| TaylorGP | $0.51e^{x_3}e^{\sin(0.87x_3)}$ | $-\frac{\sin(0.34x_2^2)}{-\sqrt{|x_2|} + \sin(\sqrt{|x_2|})}$ | $\sqrt{|x_1|} - x_1^2$ | $2$ |
| NeSymReS | — | $-0.39x_0 + x_1\sin(\frac{x_0}{x_1} - 0.001x_2)$ | $\frac{0.12x_0 + x_1^2}{\cos(3.1(-0.02x_1 - 1)^2) - 0.31}$ | $\cos(\sin(1.69x_1)/(x_0x_1)) + 0.71$ |
| E2E | $e^{1.2x_3} - 0.91\cos((2.62x_0 + 0.15)(24.66x_1 + 1.24)) - 0.05$ | $0.01x_1(-7.5\cos(15.41x_1 + 0.21) - 0.18) + 0.69\,\text{atan}(0.75x_0 + 0.05) + 0.47$ | $(-0.03x_1 - 0.03)(0.34x_1 - 0.35)(41.59(1 - 0.5\sin(6.74x_0 + 0.23))^2 + 40)$ | $2.01 - 1.05e^{-0.06|x_0 2.73 - 0.14||0.59x_1 + 0.1|}$ |
| uDSR | $0.001x_0^2 - 0.001x_0x_2x_3 + 0.001x_0x_2 - 0.002x_0x_3 + 0.002x_1x_3 + 0.003x_1 + 0.002x_2^2 + 0.004x_2x_3 + 0.003x_2 + 0.152x_3^4 + 0.568x_3^3 + 0.506x_3^2 + 0.609x_3 + \sin(x_0 + x_1x_2) + 1.074$ | $-0.002x_0^3 + 0.001x_0^2 - 0.005x_0x_1 - 0.003x_0x_2 + 0.234x_0 + 0.046x_1^2 - 0.001x_1x_2 + 0.005x_1 + 0.003x_2^2 - 0.257x_2^2 - 0.012x_2 + 3\cos(x_2) - \cos(2x_2) + 2.512$ | $-0.001x_0^3x_1 - 0.017x_0^3 + 0.002x_0^2x_1 + 0.003x_0^2 - 0.019x_0x_1^2 + 0.017x_0x_1 + 0.283x_0 - 0.001x_1^4 - 0.89x_1^2 - 0.042x_1 + e^{\sin(6x_0)} - 0.372$ | $0.008x_0^4 - 0.179x_0^2 + 0.001x_0x_2 + 0.001x_0 - 0.004x_1^4 + 0.107x_1^2 + 0.004x_1 - e^{\cos(x_1)} - \cos(x_0) + \cos(x_1) + 2.721$ |
| TPSR | $1.032e^{-0.001x_2 + 1.191x_3} - 0.054 + 0.063\cos(-2.276x_0 + x_1 + 5.52)$ | $(0.003 - 0.024(\cos(39.378(1 - 0.005x_2)^2 - 38.823) + 0.56)^2)(-3.997x_0 + 51.732x_2 + 16.608\cos(1.089x_1 - 0.479) - 135.856)$ | $0.681 - 0.848(-x_1 - 0.134\cos(x_0 + 2.153) - 0.069)^2$ | $2.002 - 1.474e^{-0.009|(39.99x_0 - 0.28)(1.08x_1 + 0.03)|}$ |
| SeTGAP | $0.999e^{1.2x_3} - \sin(x_0 + x_1x_2 + 9.42)$ | $\cos(0.2x_2^2 + 0.05)|x_1| + \tanh(0.5x_0)$ | $\frac{4.53 - 4.54x_1^2}{4.54\sin(6.28x_0 + 6.28) + 6.81}$ | $2 - \frac{19.76}{19.31x_0^4 + 0.12x_0^3 + 0.42x_0^2 + 19.72} - \frac{5.33}{5.44x_1^4 - 0.09x_1^2 + 5.34}$ |

size above 2 and more than 2 BFGS restarts were unnecessary. The transformer architecture used by UDSR follows a similar configuration. For E2E, we limited the number of generated candidates to a maximum of $K = 10$, as increasing this value provided no observable improvement. TPSR employed E2E as its backbone, and its regularization parameter $\lambda$, which controls the trade-off between accuracy and expression complexity, was set to 1, as smaller values led to unnecessarily complex expressions.

For GP-based methods, we set an iteration limit of 10,000, though all cases converged earlier. Population sizes of 100, 200, 500, and 1000 were tested, with no observed benefits beyond 500. All methods, including uDSR, were configured to use the same set of operators as those present in the vocabularies used by the neural SR methods. In addition, uDSR was configured to include polynomial terms up to degree four and executed for up to 2 million expression evaluations per sub-problem. Although our approach may appear similar to CVGP, as discussed in Sec. 3.3.3, we did not include it in our experiments. This is because CVGP relies heavily on a data oracle and does not apply to a standard SR setting where a fixed initial dataset is available, making a fair comparison infeasible. Tables 2–4 present the learned expressions, with shaded cells indicating that the learned expression's functional form matches that of the underlying function. To ensure a fair comparison, the evaluation was repeated nine additional times, each with a newly generated dataset using a different random seed (Appendix F).

We evaluated the extrapolation capability of the learned expressions by testing them on an extended domain range. The original domain range, or interpolation range, for a variable $x_v$ is denoted as $[x_v^\ell, x_v^u]$, while its extrapolation range is defined as $[2x_v^\ell, x_v^\ell[ \cup ]x_v^u, 2x_v^u]$. This evaluation was repeated for each of the 10 expressions learned by each method. We also evaluated the extrapolation capability of the opaque models

Table 4: Comparison of predicted equations (E9—E13) with rounded numerical coefficients — Iteration 1

| Method | E9 | E10 | E11 | E12 | E13 |
|---|---|---|---|---|---|
| PySR | $\log(\frac{x_1+0.5}{0.5+2x_0^2})$ | $\sin(x_0 e^{x_1})$ | $x_0\log(x_1^4)$ | $\sin(\frac{x_0}{x_1/0.12})|x_0|+0.99$ | $\sqrt{x_0}\log(x_1^2)$ |
| TaylorGP | $\log(\frac{0.79}{|x_0|})-$ $2.36e^{-x_1}$ | $x_0 e^{-\sqrt{|x_1|}}$ | $4x_0\log(|x_1|)$ | $(x_0+\sqrt{|x_1|}-$ $0.91)\sin(\frac{0.73}{x_1})$ | $\sqrt{e^{\sqrt{x_0}}}\log(|x_1|)+$ $\log(|x_1|)+0.58$ |
| NeSymReS | $1.12\log(|x_1/x_0|)-1.37$ | $\sin(x_0 e^{x_1})$ | $x_0\log(x_1^4)$ | $(x_0+x_1)\sin(1/x_1)$ | $0.31x_0+3.19\log(x_1^2)-3.25$ |
| E2E | $2-0.60\log(13.36(0.004-$ $x_0)^2(1-0.13/$ $(-0.06x_1-0.02))^2+0.8)$ | $-0.98\sin((0.06-2.86e^{1.03x_1})$ $(0.32x_0+0.002))-0.007$ | $-0.74x_0(-5.62$ $\log(0.07|-6.94x_1+0.13|$ $+0.01)-3.74)$ | $(0.79x_0-0.04)$ $\sin(4.5/(3.4x_1+0.08))$ | $(-90.0+\frac{9}{0.12|3.4x_1+0.12|+0.04})$ $(0.09-0.1\log(0.17x_0+3.29))$ |
| uDSR | $\log\left(\frac{x_1(2x_1+1)}{4.0x_0^2x_1+1.0x_1}\right)$ | $\sin(x_0 e^{x_1})$ | $x_0\log(1.0x_1^4)$ | $(x_0x_1\sin(1/x_1)+x_1)/x_1$ | $\log(x_1^2)\log(0.004x_0^3+0.06x_0^2$ $+0.001x_0x_1+1.189x_0$ $-0.003x_1+1.422)$ |
| TPSR | $0.401x_1-0.025|8.656x_0-$ $120.665\arctan(-0.611x_0-$ $0.004)+0.098|+0.94$ | $1.0\sin(1.002x_0 e^{x_1})$ | $(0.022-0.601x_0)$ $(-0.223x_1^2-7.502-$ $\frac{36.162}{-4.305|x_1|-0.971})$ | $0.001x_0(17.945x_1-$ $0.666+\frac{1.203}{x_1})-$ $0.008x_0+0.941$ | $\left(-0.008+\frac{0.742}{5.792|1.369x_1+0.005|+1.035}\right)$ $\left(7.888-\frac{28.241}{x_0+4.024}\right)(x_1-0.995)$ $(1.462x_0+44.99)(0.112x_1+0.111)$ |
| SeTGAP | $-\log(13.95x_0^2+3.48)+$ $|\log(8.32x_1+4.18)|-0.18$ | $\sin(x_0 e^{0.999x_1})$ | $1.998x_0\log(x_1^2)$ | $x_0\sin(1/x_1)+1$ | $2\sqrt{x_0}\log|x_1|$ |

Table 5: Extrapolation MSE Comparison

| Eq. | NN | PySR | TaylorGP | NeSymRes | E2E | uDSR | TPSR | SeTGAP |
|---|---|---|---|---|---|---|---|---|
| E1 | 431.1 ± 0.39 | 0.37 ± 0.40 | 1.22 ± 1.31 | 3.37 ± 3.49 | 1.22 ± 0.13 | 1.94 ± 1.32 | 0.65 ± 0.61 | **0.12 ± 0.09** |
| E2 | 30.77 ± 0.37 | 98.99 ± 64.70 | 238.50 ± 75.63 | 8.59e+7 ± 1.91e+8 | 68.85 ± 68.22 | 1.32e+03 ± 3.19e+02 | 1.38e+7 ± 4.04e+7 | **0.08 ± 7.62e-2** |
| E3 | 2.14e+4 ± 594.9 | 2.07e+3 ± 6.19e+3 | 3.15e+4 ± 3.96e+3 | 4.65e+4 ± 9.21e+3 | 1.45e+5 ± 3.68e+5 | 4.84e+04 ± 1.16e+04 | 9.42e+7 ± 2.83e+8 | **10.28 ± 17.89** |
| E4 | 2869 ± 15.4 | 2.21e+5 ± 9.04e+4 | 6.62e+3 ± 1.06e+3 | — | 876.20 ± 2.32e+3 | 2.44e+5 ± 7.26e+5 | | 1.62 ± 1.89 |
| E5 | 9.29e+4 ± 1.19e+3 | **5.01e-2 ± 0.15** | 2.22e+4 ± 2.99e+4 | — | 2.05e+3 ± 2.81e+3 | 5.726e+04 ± 5.035e+02 | 5.73e+4 ± 1.72e+5 | 1.02 ± 1.33 |
| E6 | 200.8 ± 0.52 | 18.26 ± 34.46 | 116.10 ± 3.86 | 189.20 ± 40.71 | 130.20 ± 25.14 | 2.631e+04 ± 1.876e+03 | 213.9 ± 191.3 | **3.83 ± 5.89** |
| E7 | 2330 ± 14.28 | 570.70 ± 1.11e+3 | 1.04e+3 ± 6.62 | 1.62e+3 ± 819.30 | 1.78e+3 ± 525.90 | 1.384e+03 ± 1.071e+02 | 1.28e+3 ± 656.6 | **3.60e-2 ± 4.49e-2** |
| E8 | 1.68e-2 ± 4.66e-5 | **3.32e-4 ± 4.76e-4** | 1.18 ± 2.12 | 0.12 ± 1.91e-2 | 0.13 ± 0.38 | 5.731e+02 ± 2.295e+02 | 1.25e-2 ± 1.83e-2 | **3.38e-8 ± 6.31e-8** |
| E9 | 6.09e-2 ± 3.73e-4 | 779.40 ± 2.33e+3 | 0.30 ± 0.15 | 1.03 ± 4.76e-2 | 0.43 ± 0.17 | 1.06e+01 ± 2.65e+01 | 0.77 ± 1.11 | **2.72e-6 ± 4.09e-6** |
| E10 | 2.39 ± 3.38e-2 | **7.3e-11 ± 1.5e-10** | 0.17 ± 0.21 | **0.00 ± 0.00** | 0.36 ± 4.76e-2 | 0.00 ± 0.00 | 4.96e-12 ± 1.45e-11 | 3.72e-4 ± 5.62e-4 |
| E11 | 240.2 ± 3.09 | 1.94e-2 ± 5.83e-2 | 3.08 ± 8.02 | **0.00 ± 0.00** | 98.80 ± 104.80 | **2.35e-11 ± 5.34e-11** | 200.8 ± 125.0 | 9.36e-5 ± 2.81e-4 |
| E12 | 4.35 ± 0.24 | 0.80 ± 2.41 | 2.97 ± 1.57 | 2.57e-2 ± 0.00 | 5.31 ± 1.66 | 6.78e-03 ± 2.03e-02 | 15.42 ± 20.69 | **1.11e-6 ± 2.22e-6** |
| E13 | 1.43 ± 5.5e-2 | 2.72 ± 8.16 | 9.43 ± 23.11 | 60.55 ± 15.63 | 182.90 ± 330.00 | 1.47e+05 ± 4.41e+05 | 26.17 ± 22.79 | **8.59e-7 ± 1.78e-6** |

$\hat{f}$ trained for each problem. Each extrapolation set comprised 10,000 points sampled uniformly within this range. Table 5 shows the rounded mean and standard deviation of the extrapolation MSE across these runs. Bold entries indicate the method with the lowest mean error and a statistically significant difference, determined by Tukey's honestly significant difference test at the 0.05 significance level.

Finally, we tested SeTGAP under increasingly noisy conditions. We considered a normal error term, defined as $\varepsilon_a = \mathcal{N}(0, \sigma_a\sigma_\mathbf{y})$, and four noise levels: $\sigma_a = \{0, 0.01, 0.03, 0.05\}$. Here, $\sigma_\mathbf{y}$ denotes the standard deviation of the response variable so that the noise is scaled relative to the dispersion of each problem. The obtained interpolation and extrapolation MSE are shown in Table 6, where shaded cells indicate an incorrectly identified functional form. The corresponding learned expressions are provided in Appendix G.

## 5    Discussion

SeTGAP can be viewed as a *post-hoc* interpretability method, as it extracts mathematical expressions that align with the functional response learned by a given opaque regression model. Our decomposable approach learns and preserves functional relationships between input variables and the system's response, and increments them progressively, allowing for an interpretable evolution.. This prevents the resulting expressions from focusing solely on error minimization, encouraging alignment with the true functional form instead.

From Tables 2–4, we confirmed that SeTGAP successfully learned mathematical expressions equivalent to the underlying functions in Table 1 across all tested problems and all 10 iterations. In contrast, none of the competing methods correctly identified the underlying functions in more than seven out of the 13 problems in any of the 10 iterations. Notably, some methods only captured the functional form of the most influential variables; i.e., those contributing most to the response value. For instance, E2E recovered correctly the term $0.06x_0^2-0.51x_0+6.5$ for $x_0$ in E2 but failed for the other variables. E2E and TPSR were the only methods that produced expressions longer than reported, requiring simplification via a symbolic manipulation library[3].

---

[3]SymPy: https://www.sympy.org/

Table 6: MSE comparison using SeTGAP with noisy data

| Eq. | Interpolation ($\sigma_a = 0$) | Interpolation ($\sigma_a = 0.01$) | Interpolation ($\sigma_a = 0.03$) | Interpolation ($\sigma = 0.05$) | Extrapolation ($\sigma_a = 0$) | Extrapolation ($\sigma_a = 0.01$) | Extrapolation ($\sigma_a = 0.03$) | Extrapolation ($\sigma_a = 0.05$) |
|-----|------|------|------|------|------|------|------|------|
| E1 | 7.159e-03 | 1.259e-02 | 1.236e-01 | 5.455e-01 | 1.012e-01 | 4.433e-01 | 1.645e+00 | 3.845e+00 |
| E2 | 7.024e-03 | 4.459e-03 | 1.563e-02 | 4.754e-02 | 1.300e-01 | 5.232e-02 | 1.017e-01 | 3.504e-01 |
| E3 | 1.063e-03 | 1.022e-03 | 7.219e-03 | 2.003e-02 | 2.961e-01 | 2.055e+02 | 7.467e+01 | 1.652e+02 |
| E4 | 1.158e-04 | 2.969e-03 | 2.061e-01 | 2.228e-02 | 4.121e-02 | 1.022e+01 | 7.420e+01 | 3.105e+01 |
| E5 | 7.211e-06 | 6.807e-03 | 5.288e-01 | 1.694e-01 | 3.069e-01 | 1.097e+01 | 8.402e+01 | 2.312e+02 |
| E6 | 5.619e-03 | 1.147e-02 | 1.726e-02 | 4.819e-02 | 4.508e-02 | 4.651e+00 | 2.081e-01 | 5.598e-01 |
| E7 | 2.655e-06 | 1.076e-02 | 6.822e-02 | 1.903e-01 | 6.447e-05 | 2.517e-01 | 1.247e+00 | 3.520e+00 |
| E8 | 1.742e-06 | 2.413e-05 | 2.103e-04 | 5.828e-04 | 1.817e-10 | 1.686e-07 | 7.539e-09 | 1.672e-07 |
| E9 | 4.907e-08 | 2.160e-04 | 1.943e-03 | 5.408e-03 | 2.780e-07 | 3.063e-04 | 2.756e-03 | 7.943e-03 |
| E10 | 2.282e-06 | 3.298e-05 | 2.950e-04 | 8.192e-04 | 1.857e-03 | 2.233e-04 | 4.793e-04 | 3.379e-03 |
| E11 | 1.178e-04 | 1.838e-02 | 1.654e-01 | 4.595e-01 | 9.308e-04 | 1.453e-01 | 1.308e+00 | 3.632e+00 |
| E12 | 8.029e-06 | 4.948e-04 | 4.436e-03 | 1.232e-02 | 7.273e-06 | 1.031e-03 | 9.257e-03 | 2.571e-02 |
| E13 | 2.583e-07 | 4.200e-03 | 3.991e-02 | 1.047e-01 | 1.015e-06 | 9.599e-03 | 9.348e-02 | 2.389e-01 |

Table 5 confirms that SeTGAP achieved lower or comparable extrapolation MSE values across all problems. These results suggest that other methods, which optimize purely for in-domain MSE, tend to overfit and learn expressions lacking the structural flexibility needed for extrapolation. In contrast, SeTGAP's decompositional approach learns functional forms that better capture underlying relationships, enabling superior generalization beyond the training domain. For example, in problem E8, most SR methods effectively minimized prediction MSE. TaylorGP, prioritizing parsimony, evolved the expression $\tilde{f}(\mathbf{x}) = 2$, effectively smoothing the data but offering no meaningful solution. This illustrates that the simplest solution isn't always best, as small data variations may reflect functional components important for generalization. Table 5 also shows cases where NeSymRes achieved zero extrapolation error across all 10 iterations for E10 and E11, while SeTGAP had low but nonzero MSE. This is because some methods learned expressions that exactly matched the underlying form, eliminating the need for coefficient fitting. PySR, for instance, identified $\sqrt{x_0}\log(x_1^2)$, an expression with no numerical coefficients, for E13 in its first iteration. SeTGAP, by contrast, produced $2.000137\sqrt{x_0}\log|x_1|$, where the fitted constant caused minor errors. Similarly, uDSR achieved near-zero extrapolation error for E4 consistently, whereas SeTGAP obtained higher MSE despite recovering the functional form correctly. We also compare SeTGAP's extrapolation with that of the opaque NNs from which its symbolic expressions are derived. Despite being trained on the same in-domain data, the original networks consistently show significantly higher extrapolation errors, as shown in Table 5. This highlights SeTGAP's value not only as a regression method but also as an interpretability tool that extracts functional approximations with superior generalization.

Since SeTGAP was the only method that consistently identified the correct functional form across all tested problems, we evaluated its robustness under different noise levels, as shown in Table 6. As expected, interpolation and extrapolation errors increased with higher noise levels. Interpolation errors remained low across all cases, while extrapolation errors showed a few exceptions due to poor coefficient fitting or incorrect functional form identification. For example, E3 and E5 exhibited high extrapolation errors because their functions contain exponential terms. Even when the learned expressions matched the expected functional form, small coefficient errors in the exponential term led to significant deviations for larger values of $x_1$. A similar case was observed in E1, where extrapolation errors were high for $\sigma_a \geq 0.03$ despite correctly identifying the functional form. The ability to recover the correct functional form in the presence of noise can be attributed to two factors. First, the Multi-Set Transformer was trained with small noise levels, making it more resilient to perturbations. Second, the regression NN $\hat{f}$ used during inference to generate the multiple input sets $\tilde{\mathbf{D}}_v$ smooths the estimated response values, mitigating the impact of noise. In the remaining cases, we observed two outcomes. SeTGAP correctly identified the functional forms of individual variables, but noise hindered the detection of relationships between variables. Otherwise, incorrect but reasonable univariate skeletons were identified, leading to expressions with small errors; e.g., for E9 and $\sigma_a = 0.05$, SeTGAP produced $\tilde{f}(\mathbf{x}) = 5.965\sqrt{0.63\log(9.4x_1 + 6.4) + 1} - \log(11.26x_0^2 + 2.82) - 7.71$.

SeTGAP is limited by the complexity of skeletons and the set of unary and binary operators used to train the Multi-Set Transformer. As shown in Appendix H, SeTGAP fails on problem F4 of the SRBench++ benchmark. The univariate skeleton of the underlying function with respect to $x_0$ is

$(c_1 x_0 + \sin(x_0 + c_2))/(c_3 x_0^2 + c_4)$, which requires eight operators, including three unary operators (`sin`, `sqr`, and `inv`). This level of complexity exceeds the capabilities of our approach, since the Multi-Set Transformer used in the experiments was trained only on expressions containing up to seven operators and at most two unary operators. This limitation, inherent to any neural SR method, can be addressed through transfer learning, where the model is fine-tuned on more complex tasks to expand its expressivity. Another limitation is the computational cost due to multiple intermediate optimization processes, making SeTGAP less efficient than all compared approaches. However, in applications like scientific discovery, where the goal is to derive interpretable and reliable mathematical expressions rather than simply optimizing predictive accuracy, the additional computational effort is warranted. In this work, we do not aim to achieve competitive computational complexity, but rather to demonstrate that a decomposable approach reconstructs the correct functional form of a system more effectively. Future work will focus on developing fast and scalable merging strategies capable of synthesizing high-dimensional expressions efficiently.

## 6 Conclusion

Symbolic regression aims to find interpretable equations that represent the relationships between input variables and their response. As such, SR represents a promising avenue for building interpretable models, a key aspect of modern machine learning. By distilling opaque regression models into transparent mathematical expressions, SR provides insight into their inner workings and enhances confidence in their reliability.

In this work, we introduced SeTGAP, a decomposable SR method that integrates transformers, genetic algorithms, and genetic programming. It first generates multiple univariate skeletons that capture the functional relationship between each variable and the system's response. These skeletons are systematically merged using evolutionary approaches, ensuring interpretability throughout the process. Experimental results demonstrated that SeTGAP consistently recovered the correct functional forms across all tested problems, unlike the compared methods. In addition, it exhibited robustness against varying noise levels.

Future work will investigate how the order of skeleton merging influences the learned expressions and their performance. We also aim to extend SeTGAP beyond unary and binary operators to identify more complex functions, such as differential operators and integral transforms. This expansion would significantly enlarge the search space, increasing the challenge of exploration. To address this, we plan to refine the Multi-Set Transformer's structure, incorporating prior knowledge to better guide the search process. Finally, while effective in consistently recovering ground-truth functions in low-dimensional settings, the evolutionary merging process becomes computationally infeasible as dimensionality increases. Future work will focus on developing faster, scalable merging strategies, grounded in a formal theoretical framework, to determine when correctly inferred skeletons can be reliably combined into coherent multivariate expressions.

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

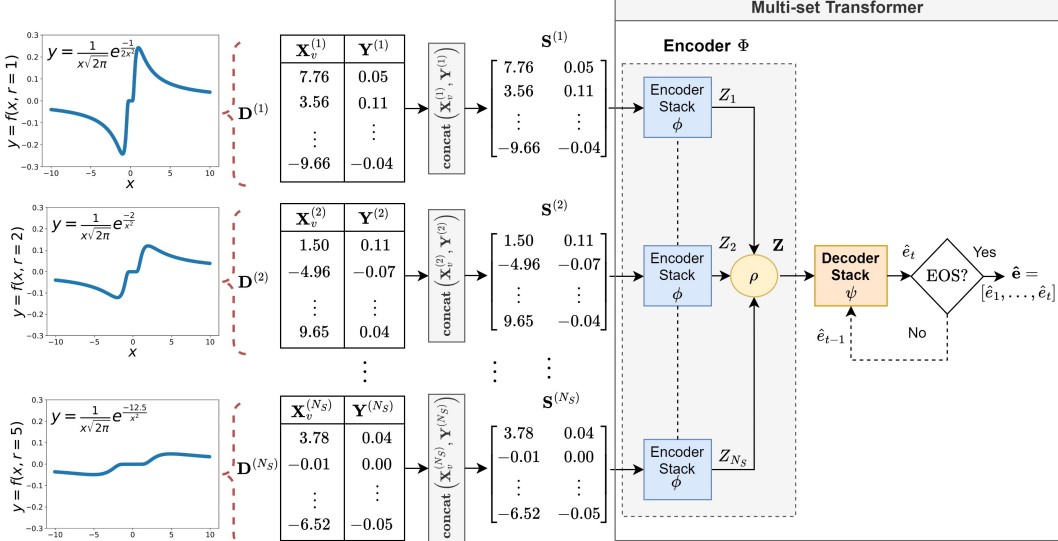

Figure 3: An example of an MSSP problem using the Multi-Set Transformer.

# A  Multi-Set Transformer

This section provides background on the Multi-Set Transformer introduced in our prior work [Ref. 1][2], summarizing its formulation and role within the symbolic skeleton prediction task.

## A.1  Multi-Set Transformer Architecture

Recall from Sec. 3.2.1 that each input set $\tilde{\mathbf{D}}_v^{(s)} = (\tilde{\mathbf{X}}_v^{(s)}, \tilde{\mathbf{y}}^{(s)})$ is defined as a set of $n$ input–response pairs. Here, we rearrange $\tilde{\mathbf{D}}_v^{(s)}$ into a two-column matrix $\mathbf{S}^{(s)}$ by concatenating $\tilde{\mathbf{X}}_v^{(s)}$ and $\tilde{\mathbf{y}}^{(s)}$ column-wise. The Multi-Set Transformer comprises two primary components: an encoder and a decoder (Fig. 3). The encoder maps the information of all input sets into a unique latent representation $\mathbf{Z}$. To do so, an encoder stack $\phi$ transforms each input set $\mathbf{S}^{(s)}$ into a latent representation $Z^{(s)} \in \mathbb{R}^d$ (where $d$ is context vector length or the "embedding size") individually. Our encoder, denoted as $\Phi$, comprises the use of the encoder stack $\phi$ to generate $N_S$ individual encodings $Z^{(1)}, \ldots, Z^{(N_S)}$, which are then aggregated into $\mathbf{Z}$:

$$\mathbf{Z} = \Phi\left(\mathbf{S}^{(1)}, \ldots, \mathbf{S}^{(N_S)}, \boldsymbol{\theta}_e\right) = \rho\left(\phi\left(\mathbf{S}^{(1)}, \boldsymbol{\theta}_e\right), \ldots, \phi\left(\mathbf{S}^{(N_S)}, \boldsymbol{\theta}_e\right)\right) = \rho\left(Z^{(1)}, \ldots, Z^{(N_s)}, \boldsymbol{\theta}_e\right), \qquad (1)$$

where $\rho(\cdot)$ is a pooling function, and $\boldsymbol{\theta}_e$ represents the trainable weights of the encoder stack. We define $\phi$ as a stack of $\ell$ ISAB blocks so that it encodes high-order interactions among the elements of an input set in a permutation-invariant way. Furthermore, unlike the Set Transformer's encoder, we include a PMA layer in $\phi$ to aggregate the features extracted by the ISAB blocks, whose dimensionality is $n \times d$, into a single $d$-dimensional latent vector. Finally, the function $\rho(\cdot)$ that is used to aggregate the latent representations $Z^{(s)}$ is implemented using an additional PMA layer.

On the other hand, the objective of the decoder, denoted as $\psi$, is to generate sequences conditioned on the representation $\mathbf{Z}$ generated by $\Phi$. This objective is aligned with that of the standard transformer decoder (Vaswani et al., 2017); thus, the same architecture is used for our Multi-Set Transformer. Specifically, $\psi$ consists of a stack of $M$ identical blocks, each of which is composed of three main layers: a multi-head self-attention layer, an encoder–decoder attention layer, and a position-wise feedforward network.

Let $\hat{\mathbf{e}} = \{\hat{e}_1, \ldots, \hat{e}_{N_{out}}\}$ denote the output sequence produced by the Multi-Set Transformer, which represents the symbolic skeleton as a sequence of indexed tokens in prefix notation. For instance, the skeleton $\frac{c}{x}e^{\frac{c}{x^2}}$ would be expressed as the sequence of tokens $\{\texttt{mul}, \texttt{div}, \texttt{c}, \texttt{x}, \texttt{exp}, \texttt{div}, \texttt{c}, \texttt{square}, \texttt{x}\}$ in prefix notation. In

Table 7: Vocabulary used to pre-train the Multi-Set Transformer.

| Token | Meaning | Index |
|---|---|---|
| SOS | Start of sentence | 0 |
| EOS | End of sentence | 1 |
| c | Constant placeholder | 2 |
| x | Variable | 3 |
| abs | Absolute value | 4 |
| acos | Arc cosine | 5 |
| add | Sum | 6 |
| asin | Arc sine | 7 |
| atan | Arc tangent | 8 |
| cos | Cosine | 9 |
| cosh | Hyperbolic cosine | 10 |
| div | Division | 11 |
| exp | Exponential | 12 |
| log | Logarithmic | 13 |
| mul | Multiplication | 14 |
| pow | Power | 15 |
| sin | Sine | 16 |
| sinh | Hyperbolic sine | 17 |
| sqrt | Square root | 18 |
| tan | Tangent | 19 |
| tanh | Hyperbolic tangent | 20 |
| -3 | Integer number | 21 |
| -2 | Integer number | 22 |
| -1 | Integer number | 23 |
| 0 | Integer number | 24 |
| 1 | Integer number | 25 |
| 2 | Integer number | 26 |
| 3 | Integer number | 27 |
| 4 | Integer number | 28 |
| 5 | Integer number | 29 |
| E | Euler's number | 30 |

addition, each token in this sequence is transformed into a numerical index according to a pre-defined vocabulary that contains all unique symbols that appear in the dataset being processed. The vocabulary used in this work is provided in Table 7. According to this, the previous sequence in prefix notation would be expressed as the following sequence of indices: $\{0, 14, 11, 2, 3, 12, 11, 2, 18, 3, 1\}$.

During inference, each element $\hat{e}_i$ ($i \in [1, N_{out}]$) is generated in an auto-regressive manner. Specifically, the decoder $\psi$ produces a probability distribution over the elements of the vocabulary as follows:

$$\sigma\left(\psi\left(\mathbf{Z}, \boldsymbol{\theta}_d | \hat{e}_1, \ldots, \hat{e}_{i-1}\right)\right) = P\left(\hat{e}_i | \hat{e}_1, \ldots, \hat{e}_{i-1}, \mathbf{Z}\right),$$

where $\boldsymbol{\theta}_d$ represents the trainable weights of the decoder stack. This distribution is obtained by applying a softmax function $\sigma$ to the decoder's output. The element $\hat{e}_i$ is thus selected from the obtained probability distribution by using a sampling decoding strategy, which samples a token from the distribution to allow diversity in the generated sequence. Hence, the generation process can be written as:

$$\hat{e}_i = \text{sample}\left(\sigma\left(\psi\left(\mathbf{Z}, \boldsymbol{\theta}_d | \hat{e}_1, \ldots, \hat{e}_{i-1}\right)\right)\right).$$

The decoder keeps generating new sequence elements until the "end-of-sentence" token (EOS) is produced ($\hat{e}_i = 1$, according to Table 7) or the maximum output length allowed, denoted as $N_{max}$, is reached.

## A.2 Multi-Set Transformer Training

The Multi-Set Transformer is trained using a large collection of synthetically generated mathematical expressions, each stored in prefix notation. To control expression complexity, we impose a maximum length of 20 elements, following recommendations by Lample & Charton (2020) and Biggio et al. (2021). The training dataset contains one million symbolic skeletons, while an additional 100,000 are used for validation. The method used to generate this large dataset of symbolic skeletons is described in detail in [Ref. 1][2]. There, we outlined how skeletons are randomly drawn from a symbolic grammar that defines a space of valid expressions. Each skeleton is designed to be syntactically valid and structurally diverse.

---

**Algorithm 4** Multi-Set Transformer Training

---
```
 1: function TRAINMODEL(Q, g, N_S, n, B)
 2:     for each t ∈ range(1, maxEpochs) do
 3:         Batches ← getBatches(N_T, B)
 4:         for each batch ∈ Batches do
 5:             E^B, Ê^B ← [], []
 6:             for each j ∈ batch do
 7:                 D_j, e_j = generateSets(Q[j], N_S, n)
 8:                 ê_j = forward(g, D_j, e_j)
 9:                 E^B.append(e_j)
10:                 Ê^B.append(ê_j)
11:             L ← L(E^B, Ê^B)
12:             update(g, L)
13:     return g, Θ
```
---

Algorithm 4 outlines the core training routine of the Multi-Set Transformer. The function computed by the model is denoted as $g(\cdot)$, and $\mathbf{\Theta}$ denotes its weights. Note that $\mathbf{\Theta} = [\boldsymbol{\theta}_e, \boldsymbol{\theta}_d]$ contains the weights of the encoder and the decoder stacks. At each training iteration, a mini-batch of expression indices is drawn by shuffling the dataset. For each selected index $j$, the corresponding skeleton expression $\mathbf{e}_j$ is retrieved. A data collection $\mathbf{D}_j$ is then generated using the function `generateSets`, which produces multiple input–response sets derived from $\mathbf{e}_j$. The model processes each pair $(\mathbf{D}_j, \mathbf{e}_j)$ through the `forward` function to obtain a predicted skeleton $\hat{\mathbf{e}}_j$, following a "teacher forcing" strategy. The batch of predicted and ground-truth skeletons is used to compute the cross-entropy loss $\mathcal{L}(\mathbf{E}^B, \hat{\mathbf{E}}^B)$, and the model is updated via standard backpropagation and gradient descent routines encapsulated in the `update` function.

The function `generateSets`, detailed in [Ref. 1][2], plays a central role during training. Given a skeleton expression, it generates a multi-set collection of $N_S$ input–response sets, each containing $n$ samples. For each set, the skeleton is instantiated by sampling a new set of numerical constants. The resulting expression is then evaluated over a randomly sampled support vector to generate synthetic data. Repeating this process produces a collection of sets that reflect different realizations of the same symbolic structure, enabling the model to generalize across variations of functional forms that share the same skeleton.

Finally, as reported in [Ref. 1][2], we used a one-factor-at-a-time approach to tune the hyperparameters of the Multi-Set Transformer; the resulting configuration includes $N_S = 10$ input sets, each containing $n = 3000$ input–response pairs, a batch size of $B = 16$, and training performed using the Adadelta optimizer (Zeiler, 2012) with an initial learning rate of 0.0001. The architecture comprises $\ell = 3$ ISAB blocks in the encoder, $M = 5$ decoder blocks, an embedding size of $d = 512$, and $h = 8$ attention heads.

## B  Skeleton Equivalency Identification

This section lists representative cases of mathematically equivalent skeletons that our method can identify to avoid redundancy and reduce computational cost. The cases shown in Table 8 go beyond basic algebraic simplifications handled by SymPy, focusing instead on structural and parametric identities in trigonometric, hyperbolic, exponential, and logarithmic forms. This list is not exhaustive, and a more systematic approach to symbolic equivalence will be explored in future work.

At any point in SeTGAP's process, whenever a candidate skeleton matches a pattern in the first column of Table 8, it is rewritten into the corresponding form shown in the second column. If it already matches the form of the second column, it is left unchanged. After this normalization step, skeletons are compared using `SymPy.equal()`, so that skeletons that map to the same representation are detected as equivalent.

## C  Proof of Proposition 1

*Proof.* We prove the proposition by structural induction on the composition of operations that define $f(\mathbf{x})$. We begin with the base cases. If $f(\mathbf{x})$ is a constant, the required form is satisfied trivially by setting $c_0 = c$ with no additional terms. If $f(\mathbf{x}) = x_i$ is a single-variable function, the structure is preserved by setting $c_0 = 0$, $c_1 = 1$, and $T_{1,1}(\mathbf{x}) = x_i$.

Table 8: List of Equivalent Skeletons

| Skeleton A | Skeleton B | Condition / mapping of constants |
|---|---|---|
| $c_1 \cos(c_2 f(x) + c_3)$ | $c_1 \sin(c_2 f(x) + c_4)$ | $c_3 = c_4 + \pi/2$ |
| $c_1 \sin(c_2 f(x)) + c_3 \cos(c_2 f(x))$ | $c_5 \sin(c_2 f(x) + c_6)$ | $c_5 = \sqrt{c_2^2 + c_3^2}$, $c_6 = \mathrm{atan2}(c_3, c_1)$ |
| $c_1 \cos(c_2 f(x) + c_3) + c_4 \sin(c_2 f(x) + c_5)$ | $c_6 \sin(c_2 f(x) + c_7)$ | $c_6 = \sqrt{(c_1 \cos(c_3) + c_4 \cos(c_5))^2 + (c_1 \sin(c_3) + c_4 \sin(c_5))^2}$
$c_6 = \mathrm{atan2}(c_1 \sin(c_3) + c_4 \sin(c_5), c_1 \cos(c_3) + c_4 \cos(c_5))$ |
| $c_1 \sin(f(x)) + c_1 \sin(g(x))$ | $c_2 \sin(c_3(f(x) + g(x))) \cos(c_3(f(x) - g(x)))$ | $c_2 = 2c_1$, $c_3 = 0.5$ |
| $c_1 \sinh(f(x))$ | $c_2(\exp(f(x)) - \exp(-f(x)))$ | $c_2 = c_1/2$ |
| $c_1 \tanh(f(x))$ | $c_1 \frac{\exp(c_2 f(x)) - 1}{\exp(c_2 f(x)) + 1}$ | $c_2 = 2$ |
| $c_1 \log(f(x)^{c_2})$ | $c_3 \log(f(x))$ | $c_3 = c_1 c_2$ |
| $\log(\exp(c_1 f(x) + c_2))$ | $c_1 f(x) + c_2$ | Log-exp cancellation |
| $\log(c_1 \exp(f(x)))$ | $c_2 + f(x)$ | $c_2 = \log(c_1)$ |
| $c_1 f(c_2 x + c_3)$ | $c_1 f(c_2(x + c_4))$ | $c_4 = c_3/c_2$ |
| $\frac{c_1}{c_2 + c_3 f(x)}$ | $\frac{c_4}{1 + c_5 f(x)}$ | $c_4 = c_1/c_2$, $c_5 = c_3/c_2$ |

For the inductive hypothesis, assume the decomposition holds for functions $h(\mathbf{x})$ and $u(\mathbf{x})$, which are composed of unary or binary operations. That is, each function can be written in the form $h(\mathbf{x}) = c_0' + \sum_i c_i' \prod_j \nu_{i,j}'(T_{i,j}'(\mathbf{x}))$ and $u(\mathbf{x}) = c_0'' + \sum_i c_i'' \prod_j \nu_{i,j}''(T_{i,j}''(\mathbf{x}))$, with terms $T_{i,j}'$ and $T_{i,j}''$ themselves recursively decomposable in the same way.

For the inductive step, we consider the result of applying unary or binary operations to such functions. For a unary operation $f(\mathbf{x}) = \nu(h(\mathbf{x}))$, the expression satisfies the required structure by treating the composition as a single term ($i = 1$, $j = 1$) where $T_{1,1}(\mathbf{x}) = h(\mathbf{x})$, $\nu_{1,1} = \nu$, $c_0 = 0$, and $c_1 = 1$. Since $h(\mathbf{x})$ itself satisfies the recursive form by the inductive hypothesis, $f(\mathbf{x})$ does as well.

Now consider a binary operation $f(\mathbf{x}) = h(\mathbf{x}) \circ u(\mathbf{x})$, where $\circ$ is a binary operator. For the addition operation ($\circ = +$), substituting the expressions and grouping constants and summation terms yields:

$$f(\mathbf{x}) = (c_0' + c_0'') + \sum_i c_i' \prod_j \nu_{i,j}'(T_{i,j}'(\mathbf{x})) + \sum_i c_i'' \prod_j \nu_{i,j}''(T_{i,j}''(\mathbf{x})).$$

This expansion matches the desired structure $f(\mathbf{x}) = c_0 + \sum_i c_i \prod_j \nu_{i,j}(T_{i,j}(\mathbf{x}))$, considering that $c_0 = c_0' + c_0''$ and that the summation terms come directly from the sub-expressions of $h(\mathbf{x})$ and $u(\mathbf{x})$.

For the multiplication operation ($\circ = \cdot$), the product expansion gives:

$$f(\mathbf{x}) = c_0' c_0'' + c_0' \sum_i c_i'' \prod_j \nu_{i,j}''(T_{i,j}''(\mathbf{x})) + c_0'' \sum_i c_i' \prod_j \nu_{i,j}'(T_{i,j}'(\mathbf{x}))$$
$$+ \sum_{i,j} c_i' c_i'' \prod_j \nu_{i,j}'(T_{i,j}'(\mathbf{x})) \prod_j \nu_{i,j}''(T_{i,j}''(\mathbf{x})).$$

Each term fits into the structure $f(\mathbf{x}) = c_0 + \sum_i c_i \prod_j \nu_{i,j}(T_{i,j}(\mathbf{x}))$: the constant term can be expressed as $c_0 = c_0' \cdot c_0''$; the second and third terms are sums over products of unary operator applications, consistent with the required form; and the fourth term comprises products of sub-expressions from $h(\mathbf{x})$ and $u(\mathbf{x})$, which can be grouped as new terms $\prod_j \nu_{i,j}(Ti, j(\mathbf{x}))$. For other scalar-valued binary operations that are algebraically reducible, such as subtraction and division, we use the identities $h(\mathbf{x}) - u(\mathbf{x}) = h(\mathbf{x}) + (-u(\mathbf{x}))$ and $h(\mathbf{x})/u(\mathbf{x}) = h(\mathbf{x}) \cdot u(\mathbf{x})^{-1}$. Negation and inversion are unary operations, and since the unary case has been established, the result follows. Binary operations that involve non-scalar interactions (e.g., convolution or vector operations) are beyond the scope of this structural decomposition.

Thus, any function $f(\mathbf{x})$, defined by finite compositions of unary and binary operations, can always be expressed in the required form. Since the base case holds and the inductive step is proven, the proposition is established by structural induction. $\square$

Table 9: Tuning hyperparameter $n_B$.

| $n_B = 1$ | $n_B = 2$ | $n_B = 3$ | $n_B = 4$ |
|---|---|---|---|
| $14.02 \pm 19.64$ | $7.086 \pm 9.201$ | $1.021 \pm 1.325$ | $1.119 \pm 0.922$ |

Table 10: Tuning hyperparameter $n_{\text{cand}}$

| $n_{\text{cand}} = 1$ | $n_{\text{cand}} = 2$ | $n_{\text{cand}} = 3$ | $n_{\text{cand}} = 4$ |
|---|---|---|---|
| $4.487 \pm 4.919$ | $4.448 \pm 4.354$ | $1.021 \pm 1.325$ | $1.564 \pm 1.912$ |

Table 11: Tuning hyperparameter `rep`

| `rep` $= 50$ | `rep` $= 100$ | `rep` $= 150$ | `rep` $= 200$ |
|---|---|---|---|
| $8.564e+01 \pm 1.884e+02$ | $4.776e+01 \pm 1.252e+02$ | $1.021 \pm 1.325$ | $1.095 \pm 1.536$ |

Table 12: Tuning hyperparameter $\text{GA}_{\text{pop}}$

| $\text{GA}_{\text{pop}} = 300$ | $\text{GA}_{\text{pop}} = 400$ | $\text{GA}_{\text{pop}} = 500$ | $\text{GA}_{\text{pop}} = 600$ |
|---|---|---|---|
| $7.238e+02 \pm 1.213e+03$ | $1.137e+02 \pm 1.294e+02$ | $1.021 \pm 1.325$ | $0.959 \pm 1.827$ |

## D ==Merging Skeleton Expressions: Example==

In this appendix, we include an example demonstrating how all conditions in Algorithm 2 operate during a random merging process. The example considers two skeleton candidates: $e_1(x_1) = c_7 + c_8 x_1^3 + c_9 \sin(c_{10} x_1^2) \sin(c_{11} \sqrt{x_1} + c_{12} e^{x_1}) \tan(c_{13} x_1 \log(c_{14} x_1))$ and $e_2(x_2) = c_1 + c_2 x_2^2 + c_3 \sin(c_4 x_2) \tan(c_5 x_2 \log(c_6 x_2))$. Figure 4 presents the sequence of intermediate combinations, with key steps referencing the corresponding line numbers in Algorithm 2.

## E ==Hyperparameter Tuning==

This section presents hyperparameter tuning experiments aimed at assessing the effect of varying key hyperparameters on performance. Problem E5 was chosen for testing since it involves the largest number of variables among the tested problems and produced the highest extrapolation MSE values in Table 5. We adopted a one-at-a-time strategy, where a single hyperparameter was varied while the others were fixed to the default configuration: $n_B = 3$, $n_{\text{cand}} = 3$, `rep` $= 150$, and $\text{GA}_{\text{pop}} = 500$. In other words, whenever one hyperparameter was tuned, all others were kept at these default values.

For $n_B$, we tested the values $[1, 2, 3, 4, 5]$; for $n_{\text{cand}}$, we tested $[1, 2, 3, 4, 5]$; for `rep`, we tested $[50, 100, 150, 200]$; and for the population size used for all GA optimizations, denoted as $\text{GA}_{\text{pop}}$, we tested $[300, 400, 500, 600]$. Each setting was evaluated over 10 independent runs, each on a dataset generated with a different random seed. Fig. 5 shows the distribution of the extrapolation MSE values across runs, while Tables 9–12 report the rounded mean and standard deviation of the extrapolation MSE. In the box plots, the central line marks the median, box edges correspond to the 25th and 75th percentiles, whiskers span the full range excluding outliers, and outliers are defined as points outside $1.5 \times \text{IQR}$ beyond the interquartile range.

As observed in Tables 9–12, extrapolation errors do not decrease substantially when using hyperparameter values larger than $n_B = 3$, $n_{\text{cand}} = 3$, `rep` $= 150$, or $\text{GA}_{\text{pop}} = 500$. Since higher values incur additional computational cost without clear performance gains, increasing them is not justified.

## F Comparison of Predicted Mathematical Expressions

This appendix presents in Tables 13–39 the complete set of mathematical expressions learned by each symbolic regression method across iterations 2 to 9 of dataset generation. Results corresponding to the first iteration were reported in Tables 2–4.

Figure 4: Example of a random merging process, according to Algorithm 2.

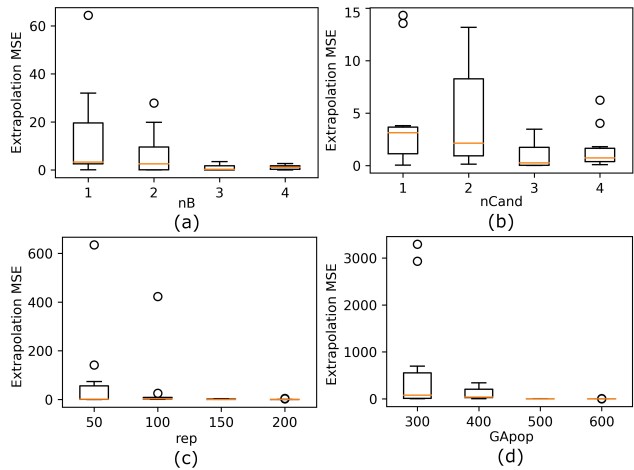

Figure 5: Hyperparameter tuning experiments for **(a)** $n_B$, **(b)** $n_{\mathrm{cand}}$, **(c)** `rep`, and **(d)** $\mathrm{GA}_{\mathrm{pop}}$.

Table 13: Comparison of predicted equations (E1—E4) with rounded numerical coefficients — Iteration 2

| Method | E1 | E2 | E3 | E4 |
|---|---|---|---|---|
| PySR | $0.607x_0x_1$ | $0.726x_0 - 0.759x_2 + 12.944\tanh(0.107x_2) - 4.572 + 11.225e^{-0.099x_0}$ | $0.15e^{1.5x_0} + 0.5\cos(3.0x_1)$ | $0.091\cosh(x_2) + \cosh(0.064x_0^2 - 0.052x_1 - 0.052x_3 + 1.154) - 1.914$ |
| TaylorGP | $0.597x_0x_1$ | $2.95(\sqrt{e^{\sqrt{\log(2x_1)}}} + 0.206 + e^{-0.294x_0} + e^{\tanh(x_2)} + e^{\tanh(\log(x_0))} + \sin^{0.5}(1.821((-e^{\log(x_0\tanh(x_0)+x_1)^{0.5}} - e^{-0.294x_0})^{0.5} + 0.179))^{0.5} + e^{\tanh(x_2)} + e^{\tanh(\log(x_0))} + \sqrt{\sin(x_2)} + e^{-0.294x_0}) + e^{-0.294x_0})^{0.5}$ | $-x_0 - (-x_0 - 0.868(e^{x_0})^{0.5} + e^{x_0})^{0.5} + e^{x_0} - \cos(\cos^{0.5}(\frac{x_1}{x_0})) + 0.092$ | $\sqrt{x_0}\log(x_0) + x_2 - e^{\tanh(\log(x_2))}$ |
| NeSymReS | $0.586x_0x_1 + \cos(0.593(x_0 - 0.979x_1)^2)$ | $-x_0 + 0.001e^{0.597x_1} + e^{\sin(0.175x_2)} + 7.267$ | $0.581e^{x_0} - 0.463\sin(1.104(0.003x_1 + 1)^2)$ | — |
| E2E | $(0.081x_0 - 17.975)(0.051\sqrt{0.008x_1 + 1} + 0.006)\cos(1.744(0.045(0.001 \cdot |15810201.601(0.292x_0 + 1)^3 + 0.016|+1)^2 + 1)^{0.5} - 0.007) + (2.709x_0 - 0.014)(0.223x_1 + 0.002)$ | $-0.002x_1 + ((0.016x_0 + 0.004)(1.231x_0 - 9.407) + \sin(0.235x_2 - 0.031) - 0.007)(0.004x_0 + 0.087x_1 - 0.081x_2 + 3.316) + 6.19$ | $0.177\sqrt{(0.58e^{3.132x_0} + 1}} + 0.486\cos(4.706(x_1 + 0.016)^2 - 0.066) - 0.104$ | $0.002|31.163x_3 + 5.524(x_0 + 0.023)^4 + 5.498(0.985x_1 - (x_2 - 0.038)^2 + 0.042)^2 - 1.464|$ |
| uDSR | $(0.001x_0^3 - 0.007x_0^2 - 0.001x_0x_1^2 + 0.611x_0x_1 + 0.002x_1^2 - 0.002x_1 - 0.006)\cos(e^{-5x_1^2+x_1})$ | $0.063x_0^2 - 0.5x_0 + 0.029x_1x_2 + 0.001x_1 - 0.001x_1^3 + 0.352x_2 + \sin(0.25x_2) + 6.499$ | $0.074x_0^4 + 0.237x_0^3 + 0.001x_0^2x_1 + 0.032x_0^2 - 0.001x_0x_1 - 0.112x_0 - 0.095x_1^4 + 0.001x_1^3 + 0.949x_1^2 - 0.005x_1 + e^{\cos(x_1/\log(2))} - 2.133$ | $(0.007x_0^4 - 0.013x_0^2x_1 + 0.007x_1^2 + 0.007x_2^4 - 0.013x_2^2x_3 + 0.007x_3^2)e^{\sin(e)}$ |
| TPSR | $-0.071x_1(-8.596x_0 - 0.119) - 0.113\cos(x_0 - 0.782) - 0.009$ | $(-1.362 + 0.321/(-9.597e^{2.221x_0 - 6.275x_2} - 0.665))(-0.044x_2 + 1.552 + 1/((0.008x_0 + 0.06)(0.043x_0 - 0.981) - 0.109))$ | $0.15e^{1.5x_0} + 0.5\cos(3.0x_1)$ | $0.044x_1 + 0.01(-x_2 - 0.013)^4 + 0.063(-0.355x_0^2 + 0.418x_1 + 0.209x_3 - 1)^2 - 0.126$ |
| SeTGAP | $0.61x_0x_1 + 1.041\sin((2.243x_0 - 1.512)(x_1 - 0.669) + 0.005)$ | $-0.497x_0 + 0.063x_0^2 + (0.648\sqrt{0.1x_1 + 1} + 0.002)(4.913\sin(0.205x_2) + 0.003) + 6.497$ | $0.151e^{1.499x_0} + 0.499\sin(3x_1 + 1.574)$ | $-0.02x_0^2x_1 - 0.004x_0^4 + 0.01x_0^4 + 0.01x_1^2 - 0.02x_2^2x_3 + 0.01x_2^4 + 0.01x_3^2 + 0.01$ |

Table 14: Comparison of predicted equations (E5—E8) with rounded numerical coefficients — Iteration 2

| Method | E5 | E6 | E7 | E8 |
|---|---|---|---|---|
| PySR | $e^{1.2x_3}$ | $\frac{\cos(0.2x_1^2)}{\cos(1.947\tanh(\tanh(0.281x_1)))} + \tanh(x_0)$ | $-x_1^2\sinh(0.492\cos(6.274x_0 + 1.574) + 0.778) + 1.049$ | $2.036\cosh(\tanh(x_0)) + 3.34\cosh(\tanh(\tanh(x_1\tanh(x_1)))) - 5.5$ |
| TaylorGP | $x_3^{0.5}e^{x_3} + \sin^{0.5}(0.056e^{x_3})$ | $\tanh(x_0) + 0.327 + \frac{4.831\sin(x_2)}{x_2}$ | $-0.974x_1^2 + |x_1|^{0.5} + 0.008$ | $\frac{\tanh(x_1)}{\cos(\tanh(x_0))}$ |
| NeSymReS | — | $0.149x_0 + x_1\sin(0.29x_1 + \frac{x_2}{x_1})$ | $\frac{0.289x_0 + x_1^2}{\cos(2.667(0.131x_1 - 1)^2) - 2.044}$ | $\cos(\frac{\sin(\frac{x_1}{x_1})}{x_0}) + 0.629$ |
| E2E | $0.96e^{1.218x_3} - 0.004 + 0.947\sin(221.795x_1 + 27.668)$ | $(5.971|0.169x_1 - 0.005|+0.14)\cos((1.531 - 17.089x_2)(-0.012x_2 - 0.009)) + 0.8\arctan(0.695x_0 + 0.196) - 0.01$ | $(0.009\sin(7.746x_0 + 0.025) + 7.454)(0.002(0.055x_0 - 1)^3 - 0.863(x_1 - 0.001)^2 + 0.696)/(6.13\sin(7.746x_0 + 0.025) + 9.52)$ | $-1.35\arctan(-0.096|2.433x_1 + (0.328x_0 - 0.01)(-2.094x_1 + (0.008 - 1.796x_0)(0.465 - 13.81x_1)(-0.835x_1 - 0.038) + 0.128) + 0.113|-0.04) - 0.035$ |
| uDSR | $-0.001x_0^2 - 0.001x_0x_1 - 0.002x_0x_3 - 0.002x_0 - 0.001x_1^2 - 0.001x_1x_2 - 0.001x_1x_3^2 - 0.002x_1x_3 - 0.002x_2x_3 + 0.004x_2 + 0.153x_3^4 + 0.57x_3^3 + 0.499x_3^2 + 0.609x_3 + \sin(x_0 + x_1x_2) + 1.107$ | $-0.001x_0^3 + 0.004x_0^2 - 0.01x_0x_1 - 0.001x_0x_2 + 0.24x_0 + 0.036x_1^2 - 0.002x_1x_2 - 0.012x_1 + 0.003x_2^4 - 0.262x_2^2 + 0.004x_2 + 3\cos(x_2) - \cos(2x_2) + 2.632$ | $-0.001x_0^4 - 0.002x_0^3x_1 - 0.015x_0^3 - 0.003x_0^2x_1^2 - 0.006x_0^2x_1 + 0.039x_0^2 - 0.008x_0x_1^2 + 0.019x_0x_1 + 0.404 + 0.235x_0 - 0.002x_1^4 - 0.827x_1^2 + 0.03x_1 + \cos(x_0x_1) + x_0\sin(x_0^2)\cos(x_0)$ | $0.008x_0^4 - 0.18x_0^2 - 0.004x_1^4 + 0.106x_1^2 + 0.001x_1 - e^{\cos(x_1)} - \cos(x_0) + \cos(x_1) + 2.728$ |
| TPSR | $0.003 + 0.987e^{0.001x_2 + 1.206x_3} - 0.036\cos(0.188x_0 - x_1 + 1.229)$ | $2795.262 - 2795.746\sin(0.008(-0.169x_0 + 0.025x_2 + 1)^2 + 1.529)$ | $-0.003(-0.124x_0 - 21.52)(x_1 - 0.202)(19.963(-0.322x_1 - 0.484\cos(6.286x_0 + 10.97) + 1)^2 - 56.092)$ | $2.008 - 1.496e^{-0.644|(0.071x_0 - 0.001)(7.973x_1 - 0.124)|}$ |
| SeTGAP | $e^{1.2x_3} - \sin(x_0 + x_1x_2 + 9.43)$ | $-(0.997|x_1|+0.008)\sin(0.201x_2^2 - 1.611) + \tanh(0.492x_0) - 0.008$ | $\frac{2.785x_1^2 + 0.05\sin(6.283x_0 - 3.149) - 2.793}{2.77\sin(6.283x_0 - 3.149) - 4.167}$ | $2 - \frac{12.162}{-0.026x_1 + 0.132x_1^2 + 0.084x_1^4 + 12.008x_1^4 + 12.144} - \frac{14.373}{14.113x_0^4 + 0.232(-x_0)^2 + 14.361}$ |

In each iteration, a new dataset was generated using a unique initialization seed, ensuring that all methods were trained and evaluated on the same data for a fair comparison. Each table reports the best expression found by each SR method, with shaded cells indicating cases where the learned expression's functional form matches the underlying ground-truth function. These results provide insight into the stability and consistency of each method in discovering accurate symbolic representations across different dataset realizations.

Table 15: Comparison of predicted equations (E9—E13) with rounded numerical coefficients — Iteration 2

| Method | E9 | E10 | E11 | E12 | E13 |
|---|---|---|---|---|---|
| PySR | $\tan(1.044\cos(0.627x_0)+0.242)+$ $\tan(\cos(e^{\cos(0.893x_1^{0.5})}))-2.117$ | $\sin(x_0 e^{x_1})$ | $2x_0\log(x_1^2)$ | $x_0\sin(\frac{1.0}{x_1})+1.0$ | $(x_0(\tan(\sinh(\sinh(\cosh(\tanh(0.309\cosh(x_1)))))))$ $+7.703))(x_0+8.766)$ |
| TaylorGP | $-0.906\log(|x_0|)+\log(|x_1|)\tanh(x_1)$ $-|x_0+0.119|^{0.5}$ | $\sin(x_0 e^{x_1})$ | $4x_0\log(|x_1|)$ | $0.752x_0\tanh(\frac{0.985}{x_1})+0.752$ | $2\sqrt{x_0}\log(|x_1|)$ |
| NeSymReS | $\log(|\frac{x_1}{x_0}|)-1.272$ | $\sin(x_0 e^{x_1})$ | $x_0\log(x_1^4)$ | $(x_0+x_1)\sin(\frac{1}{x_1})$ | $0.231x_0+\log(x_1^2)$ |
| E2E | $((1.9-0.6\log(13.214(x_0+0.001)^2$ $+0.6))(0.23x_1+0.231)-0.6)/$ $(0.23x_1+0.231)$ | $-0.02x_0-0.982\sin($ $(29.612x_0+0.106)$ $(10.592e^{0.571x_1}$ $-0.033)(0.017\cos($ $0.07e^{0.902x_1}-4.53)$ $+0.002))-0.003$ | $x_0(6.669\log(0.19$ $(|22.158x_1+0.255|+$ $0.111)^{0.5}-0.01)+0.1)$ | $0.997-7.62$ $\sin(\frac{(0.131x_0-0.001)(0.004x_1-6.7)}{6.661x_1+0.137})$ | $(0.091\log(0.174x_0+0.772)$ $+0.086)(27.6\log(0.711$ $(x_1+0.043)^2+0.13)$ $-0.099)$ |
| uDSR | $-0.002x_0^4-0.001x_0^3-0.052x_0^2$ $-0.001x_0x_1+0.024x_0-0.008x_1^4$ $+0.099x_1^3-0.504x_1^2+1.45x_1$ $+e^{x_0-e^{x_0}}+e^{\cos(x_0)}-3.3$ | $\sin(x_0 e^{x_1})$ | $x_0\log(1.0x_1^4)$ | $(1.0x_0x_1\sin(1/x_1)+x_1)/x_1$ | $\log(x_1^2)\log(0.004x_0^3+$ $0.06x_0^2+1.191x_0+$ $0.001x_1+1.42)$ |
| TPSR | $\left(-3.757-\frac{2.69}{10.828(x_0-0.002)^2+13.341}\right)$ $(-1.817\arctan(x_1+2.012)-10.316)$ $-0.045|39.196x_0+0.018|$ $(0.003x_1-0.256)|-48.763$ | $1.0\sin(0.9976x_0e^{x_1})$ | $-1.567x_0+(0.007+1/(-1.1$ $|12.472x_1-0.03|-0.988))$ $(12.636-27.802(x_1-0.002)^2)$ $(1.515x_0+0.013)-0.05$ | $6.9697135x_1+$ $1.654205-\frac{4.6\cdot10^{-5}}{x_1}$ | $(-0.053+$ $\frac{1}{0.228|6.18x_1+0.015|+0.15})$ $(0.088-0.092x_1^2)$ $\left(1.026-\frac{1.844}{x_0+1.941}\right)$ $(-2.037x_0-40.447)$ |
| SeTGAP | $-\log(30.042x_0^2+7.509)+$ $|\log(13.19x_1+6.59)|+0.13$ | $\sin(x_0 e^{x_1})$ | $2x_0\log(x_1^2)$ | $x_0\sin(\frac{1}{x_1})+1.0$ | $2.0\sqrt{x_0}\log(|x_1|)$ |

Table 16: Comparison of predicted equations (E1—E4) with rounded numerical coefficients — Iteration 3

| Method | E1 | E2 | E3 | E4 |
|---|---|---|---|---|
| PySR | $0.607x_0x_1+$ $1.1\cos(-2.25x_0x_1+$ $1.5x_0+1.5x_1+0.571)$ | $e^{e^{-\sin(0.128x_0)}}+$ $1.214e^{\sqrt{e^{\tanh(x_2)}}}+\tanh(\frac{x_1}{x_2})$ | $0.15e^{1.499x_0}+$ $0.497\cos(3.001x_1+\frac{1.557}{x_0+22.291})$ | $-0.02x_2^2(\frac{2.628x_1\sin(x_2)}{x_2}+x_3)-$ $0.007(x_1-13.401)(\cosh(x_0)+\cosh(x_2))$ |
| TaylorGP | $0.6x_0x_1$ | $-0.503x_0+0.001x_1+0.211x_2+8.607$ | $0.346e^{x_0}\log(1.348(-0.247x_0+e^{x_0}$ $-0.247e^{0.5|\sin(x_1)|^{0.5}})^{0.5})+0.107$ | $x_0+x_2-|x_0|^{\frac{1}{4}}|x_2|^{\frac{1}{4}}+0.778$ |
| NeSymReS | $0.586x_0x_1+$ $\cos(0.585(0.991x_0-x_1)^2)$ | $-x_0+0.384e^{0.024x_1}+e^{\sin(0.175x_2)}+6.813$ | $0.582e^{x_0}-$ $0.749\sin(0.57(1-0.001x_1)^2)$ | — |
| E2E | $0.255x_0(2.399x_1+0.034)+$ $1.11\cos(3089.014x_0+$ $1035.398)-0.006$ | $(0.776-0.111x_0)(0.171x_1+(0.174x_0+0.019)$ $(0.001x_2-0.07)(-0.499x_0+$ $0.042\cos(1.689x_1+50.388)$ $+42.271)+0.017)-2.94\cos((2.184x_1+86.243)$ $(0.025\sin(0.116x_2-0.038)+0.016))+6.62$ | $0.113e^{1.594x_0}+0.511\cos(3.08$ $(x_1+0.032)^2+0.008)+0.136$ | $0.0$ |
| uDSR | $(0.001x_0^3x_1+0.002x_0^3+2.455x_0^2x_1$ $+0.002x_0^2+0.002x_0x_1^3-0.002x_0x_1^2$ $-1.229x_0x_1-0.045x_0-0.004x_1^3$ $-0.007x_1^2+0.103x_1+0.104)$ $\sin\left(\frac{x_0}{2x_0+(2x_0^2-2x_0)/x_0}\right)/x_0$ | $\log(662.215e^{0.062x_0^2+0.001x_0x_2}$ $e^{-0.503x_0+0.029x_1x_2-0.002x_2^2+0.541x_2}+\cos(x_1/x_2))$ | $0.075x_0^4+0.001x_0^3x_1+0.235x_0^3+$ $0.023x_0^2-0.003x_0x_1-0.108x_0$ $+0.014x_1^4-0.228x_1^2-$ $\sin(x_1)\sin(2x_1)+0.706$ | $0.01x_0^4-0.02x_0^2x_1+0.01x_1^2+0.01x_2^4$ $-0.02x_2^2x_3+0.01x_3^2-0.526+\sin(e^e)$ |
| TPSR | $-0.141x_1(0.053-4.272x_0)+$ $0.002\sin(6.904x_1-7.997)$ $+0.05$ | $(-0.08+\frac{1}{-468.708e^{3.582x_0-9.1x_2}-11.692})$ $(-0.03x_1-7.596)((-0.427x_0-0.022x_2-$ $(-0.006x_0-0.303)(1.156x_0+2.022)-0.294)^2$ $+1.84)(-0.008x_1+1.435+\frac{1}{0.002x_0+0.394})$ | $0.15e^{1.5x_0}+0.5\cos(3.0x_1)$ | $0.011\left(0.705-(-x_0-0.011)^2\right)^2+$ $0.049(-0.003x_0-0.243x_1$ $+0.41x_2^2-0.407x_3+1)^2-0.065$ |
| SeTGAP | $0.607x_0x_1+$ $1.123\sin((2.243x_0-1.496)$ $(x_1-0.654)+0.014)$ | $-0.5x_0+0.062x_0^2+$ $(3.163\sqrt{0.1x_1+1}+0.001)\sin(0.2x_2)+6.513$ | $0.151e^{1.497x_0}-$ $0.5\sin(3.004x_1-7.854)$ | $-0.004x_0-0.02x_0^2x_1+$ $0.017(-x_0)^2+0.009(-x_0)^4+0.008(-x_1)^2+$ $0.01(-x_2)^4+0.011(-x_3)^2-(0.013x_3^2+0.003)$ $(1.575x_3+0.614)-0.024$ |

Table 17: Comparison of predicted equations (E5—E8) with rounded numerical coefficients — Iteration 3

| Method | E5 | E6 | E7 | E8 |
|---|---|---|---|---|
| PySR | $e^{1.2x_3}+\sin(x_0+x_1x_2)$ | $|x_1|\cos(0.2x_2^2)+\tanh(0.502x_0)$ | $-0.412x_1^2\cosh(1.055e^{-0.743\sin(6.278x_0)})+1$ | $2.046\cosh(\tanh(x_0))+$ $2.046\tanh(\frac{x_1}{\tanh(\sinh(1.875x_1))})$ $-3.196$ |
| TaylorGP | $2e^{x_3}-\log(3e^{x_3}+0.856\sqrt{e^{(x_3)}|x_3|}-$ $\log(2.32e^{x_3}+0.856\sqrt{e^{(x_3)}|x_3|})-0.26)-0.26$ | $\frac{x_1\sin(x_2)}{x_2\tanh(x_1)}+\cos(x_2)+\tanh(x_0)$ | $-0.974x_1^2+\sqrt{|x_1|}$ | $1.148|x_0|^{\frac{1}{4}}|x_1|^{\frac{1}{4}}-0.105$ |
| NeSymReS | — | $0.144x_0+x_1\sin(0.065x_1+\frac{x_2}{x_1})$ | $\frac{0.072x_0+x_1^2}{\cos(3.099(0.012x_1-1)^2)-0.252}$ | $\cos\left(\frac{\sin(\frac{x_0}{x_1})}{x_0}\right)+0.69$ |
| E2E | $0.871e^{1.253x_3}+0.981\cos(88.561(0.014|3.137x_1+$ $31752866.296(0.104\arctan(0.087x_2+$ $5.508)-1)^2-2.465|-1)^{0.5}-0.006)+0.13$ | $(0.002\sin(0.816x_1+0.616)+$ $0.151)(0.036x_1+\cos(0.711(0.014$ $-x_2)^2-0.573)+3.995)$ | $(0.002(1-0.256\sin(0.183(-32.468x_0+$ $0.047x_1+14.284|-0.293))^3$ $-0.026)(-0.043x_0+$ $19.294(x_1+0.017)^2-0.543)$ | $0.865$ |
| uDSR | $-0.001x_0^2+0.001x_0x_2-0.001x_0x_3+0.002x_0-$ $0.001x_1^2x_3+0.002x_1^2-0.001x_1x_3-0.004x_1+$ $0.003x_2^2+0.005x_2x_3-0.002x_2+0.153x_3^4+1.084$ $+0.567x_3^3+0.508x_3^2+0.619x_3+\sin(x_0+x_1x_2)$ | $-0.002x_0^3+0.001x_0^2-0.006x_0x_1+$ $0.004x_0x_2+0.26x_0+0.032x_1^2+$ $0.003x_1x_2-0.003x_1+0.003x_2^2-$ $0.266x_2^2+0.022x_2+3\cos(x_2)-$ $\cos(2x_2)+2.831$ | $0.001x_0^4+0.001x_0^3x_1-0.014x_0^3-0.002x_0^2x_1$ $-0.015x_0^2-0.002x_0x_1^3-0.009x_0x_1^2+0.002x_0x_1$ $+0.231x_0+0.001x_1^4+0.002x_1^3-0.913x_1^2$ $-0.004x_1+(x_0\cos(x_0)-\sin(x_0))\sin(x_0^2)+0.933$ | $0.008x_0^4-0.18x_0^2+0.001x_0x_2+$ $0.001x_0-0.004x_1^4+0.108x_1^2+$ $0.002x_1-e^{\cos(x_1)}-\cos(x_0)+$ $\cos(x_1)+2.722$ |
| TPSR | $(x_1+923.05)(0.001e^{1.2x_3}-0.01)+8.903$ | $(0.016\cos(0.771(x_1+0.145)^2$ $-0.875)-0.084)(-2.126x_0-$ $43.836\cos((3.209-\frac{8.357}{0.028x_2+0.476})$ $(0.058x_2+0.044))-17.68)$ | $-0.039x_0+294.76(-1+0.139/$ $(0.036-3.266e^{-7}/(-5.83e^{-9}$ $(-x_1-0.092)^2-6.046e^{-7})))^2-167.907$ | $(-0.239x_1-4.712)$ $(0.455x_1-8.997)$ $(\arctan(4.665(0.02-x_1)^2$ $(x_0-0.039)^2+31.821)-1.521)$ |
| SeTGAP | $1.006e^{1.201x_3}+1.006\sin(1.008x_0+$ $(1.005x_1-0.006)(x_2+0.029)+12.451)-0.007$ | $1.001\cos(0.2x_2^2+0.01)$ $|x_1|+1.0\tanh(0.5x_0)+0.002$ | $\frac{5.18x_1^2-5.18}{5.18\sin(6.283x_0+3.142)-7.77}$ | $2.0-\frac{14.485}{0.142x_0^2+14.37(-x_0)^4+14.466+\frac{13.074}{-13.081x_1^4-13.074}}$ |

Table 18: Comparison of predicted equations (E9—E13) with rounded numerical coefficients — Iteration 3

| Method | E9 | E10 | E11 | E12 | E13 |
|---|---|---|---|---|---|
| PySR | $1.172\sqrt{|x_1|} - 1.518\sinh(\sinh(\tan(\tanh($ $1.739e^{\cos(\cos(0.217x_0)} + \cos(\tanh(x_0))))))) + 2.232$ | $\sin(x_0 e^{x_1})$ | $2x_0 \log(x_1^2)$ | $x_0 \sin\left(\frac{1.0}{x_1}\right) + 1$ | $\sqrt{x_0} \log(x_1^2)$ |
| TaylorGP | $-\log(|x_0|) + \tanh(\log(|x_1|)) - \sqrt{\left|\log\left(\frac{0.568}{|x_0|}\right)\right|}$ | $\sin(x_0 e^{x_1})$ | $4.545 x_0 \log(|x_1|)$ | $\dfrac{2.77 \log\left(\sqrt{|x_0 + x_1|}\right)}{\tanh\left(\frac{0.5(x_0+x_1)}{x_1}\right)}$ | $2\log(|x_1|)\sqrt{|x_0|}$ |
| NeSymReS | $\log\left(\frac{0.286|x_1|}{|x_0|}\right)$ | $\sin(x_0 e^{x_1})$ | $x_0 \log(x_1^4)$ | $(x_0 + x_1)\sin\left(\frac{1}{x_1}\right)$ | $0.241 x_0 + \log(x_1^2)$ |
| E2E | $-0.098 - 0.052/((-0.003(0.029x_1 - 1)^3 + 0.005 + 6.94/(-0.001x_0 - 51.126x_1 + (3.21 - 0.085|32.081x_0 - 1.081|) (2.068x_1 - 69.713) + 126.399)))$ | $1.0\sin(0.035 x_0(26.26e^{1.16x_1} -0.1)) - 0.001$ | $(0.058|15.4\arctan( 0.258x_1 + 0.008) - 0.009| -0.005)(-0.006x_0 + 59.4 \sin(0.168x_0 - 0.001) + 0.011)$ | $1.0 + 7.5\sin(0.001x_0 (0.1 - \frac{24.6}{-0.159x_1-0.005}))$ | $1.08\log(0.02(-x_0 - 0.825)^2 (-x_1 - 0.057)^2 (-x_1 - 0.056)^2 + 0.101) + 0.09$ |
| uDSR | $\log\left(\frac{x_1(2x_1^2+x_1)}{4.0x_0^2x_1^2 + 1.0x_1^2}\right)$ | $1.0\sin(x_0 e^{x_1})$ | $x_0 \log(1.0x_1^4)$ | $x_0 \sin(1/x_1) + 1.0$ | $\log(x_1^2)\log(0.004x_0^3 + 0.061x_0^2 + 1.182x_0 -0.001x_1 + 1.433)$ |
| TPSR | $0.396x_1 + 0.004\left(1.464 + \frac{1}{19.402(0.002-x_0)^2 + 28.549}\right)$ $(0.873x_0 - 46.945)(-10.284x_0 + x_1 - 555.305) -149.911$ | $1.0\sin(1.001x_0 e^{x_1})$ | $(0.001 - 0.12x_0)(-0.971x_1^2 - 42.941 - 103.498(-0.605 |3.087x_1 - 0.011| - 0.633))$ | $(-1.292 - 0.14/(-0.179 (0.01x_0 + x_1 + 0.073)^2 - 0.106)(-0.005x_1 + \frac{0.195-0.883x_0}{x_1+0.066} - 0.809)$ | $(-0.104 + 1/(0.01|82.844x_1 -0.068| + 0.111)(49.998 - 50.042\cos(0.042x_1)) (159.942 + \frac{170.493}{-0.07x_0-1.16})$ |
| SeTGAP | $-0.999\log(11.668x_0^2 + 2.91) + 1.006|\log(13.4x_1 + 6.834)| - 0.861$ | $1.0\sin(1.0x_0 e^{x_1})$ | $2.0x_0 \log(x_1^2)$ | $1.0x_0\sin\left(\frac{1}{x_1}\right) + 1.0$ | $\sqrt{1.0x_0}\log(x_1^2)$ |

Table 19: Comparison of predicted equations (E1—E4) with rounded numerical coefficients — Iteration 4

| Method | E1 | E2 | E3 | E4 |
|---|---|---|---|---|
| PySR | $0.608x_0 x_1 + 1.1\cos(1.076(1.045x_0 -0.697)(2x_1 -1.333) - 1.571)$ | $(0.198x_1 + 2.985)\sin(0.201x_2) + \cosh(0.836e^{\tanh(2.02e^{0.162x_0})} - 4.76)$ | $13.457e^{\frac{1.753x_0 - 5.251}{\cosh(\tanh(\cos(1.525x_1)))}}$ | $0.091x_1(\cos(0.643x_0) - 1.803) - 0.044x_3 \cosh(0.65x_2) + 0.091\cosh(x_0) + 0.091\cosh(x_2) - 0.027$ |
| TaylorGP | $0.602x_0 x_1$ | $1.946\sqrt{x_0(\tanh(x_0) - 0.837)} + 1.946(\log(e^{x_2} + 11.111\sin(\cos(e^{x_0})) + 11.111\sqrt{|x_0(x_2 + 3.147\sqrt{|x_0|} + 0.984)|}))$ | $\dfrac{-x_0 + e^{x_0} -}{\sqrt{|x_0 - e^{x_0} - \sqrt{|x_0|}} + 0.785\sqrt{|e^{x_0} - 0.083|}}$ | $\frac{x_2}{\tanh(x_2)} + \log(|x_0|)\sqrt{|x_0|} - 1.571$ |
| NeSymReS | $0.586x_0 x_1 + \cos(0.586(x_0 -0.988x_1)^2)$ | $-x_0 + e^{\sin(x_2)} + 7.289$ | $0.584e^{x_0} - 0.411$ | — |
| E2E | $0.03x_0(19.977x_1 + 0.327) - 1.1\cos(31.768 x_1 + 2.118) + 0.016$ | $0.075x_0^2 - 0.443x_0 + 0.014x_1x_2 + 0.002x_1 - 0.002x_2^2 + 0.083x_2 + 3.07\sin(0.227x_2 + 0.025) + 6.23$ | $0.107x_0 + 0.156(0.318x_0 + 1)^2(x_0 - 0.002)^2 (|0.279x_0 + 0.578| + 0.133)^2 - 0.501\sin(0.091x_0 + 3.572x_1 - 1.273) + 0.152$ | $(0.095 - 0.001x_1)(0.1(0.559x_1 - (x_0 - 0.003)^2 + 0.037)^2 + 0.073) + 0.036 (-0.524x_3 + (0.204x_0 - 0.001)(0.142x_2 +0.013) + (0.017x_2 + 0.001) (32.565x_2 + 1.169) + 0.024)^2 - 0.001$ |
| uDSR | $(0.224x_0 x_1 + 0.001x_0 - 0.003x_1 -0.004)e^{\cos(e^{x_0(1-3x_0)})}$ | $\log(663.667e^{0.062x_0^2 - 0.001x_0x_1} e^{0.001x_0x_2 - 0.502x_0 + 0.03x_1x_2} e^{-0.004x_1 - 0.002x_2^3 + 0.541x_2} + 1)$ | $0.013x_0^4 - 0.032x_0^3 - 0.416x_0^2 - 0.002x_0x_1 -0.901x_0 - 0.013x_1^4 + 0.225x_1^2 -0.001x_1 + e^{x_0} + \cos(x_1)\cos(2x_1) - 1.295$ | $0.01x_0^4 - 0.02x_0^2x_1 + 0.01x_1^2 + 0.01x_2^4 - 0.02x_2^2x_3 + 0.01x_3^2$ |
| TPSR | $-0.069x_0(-8.798x_1 - 0.074) + 0.036\sin(5.606 x_1 + 2.258) + 0.009$ | $\dfrac{0.001(31.887x_1 + x_2 + 12721.866)}{e^{(-1.482(0.06x_0+0.651))}}$ $e^{\frac{78.544(0.06x_0+0.651)}{e^{-41.665x_0^{0.01x_1+0.693x_2-29.007}}}}$ | $0.15e^{1.5x_0} + 0.5\cos(3.0x_1)$ | $11.1441\left(1 - 0.944(-0.074x_2 - 1)^2\right)^2 -0.214569$ |
| SeTGAP | $0.607x_0 x_1 + 1.1\sin((2.25x_0 - 1.489) (x_1 - 0.663))$ | $-0.5x_0 + 0.063x_0^2 + (3.148\sqrt{(0.1x_1 + 1)} + 0.012) \sin(0.204x_2) + 6.493$ | $0.147e^{1.508x_0} + 0.5\sin(2.999x_1 + 1.571) + 0.002$ | $-0.02x_0^2x_1 + 0.011x_0^4 + 0.009x_1^2 + 0.011x_3^2 - 0.013(-x_0)^2 + 0.009(-x_2)^4 - (0.035x_2^2 - 0.002)(0.584x_3 - 0.46) - 0.016$ |

Table 20: Comparison of predicted equations (E5—E8) with rounded numerical coefficients — Iteration 4

| Method | E5 | E6 | E7 | E8 |
|---|---|---|---|---|
| PySR | $e^{1.2x_3} + \sin(x_0 + x_1 x_2)$ | $\sqrt{(x_1^2)}\cos(0.2x_2^2) + \tanh(0.5x_0)$ | $\frac{-1.0x_1^2 + \cos(0.011x_1\sin(6.283x_0))}{\sin(6.283x_0) + 1.5}$ | $4.404\tanh(\cosh(0.869x_0) \sqrt{\tanh(\cosh(x_1))}) \tanh(\cosh(0.869x_1)) - 2.427$ |
| TaylorGP | $e^{x_3}\sqrt{|x_3|} + \tanh(0.515e^{x_3} - 0.331\sin(\sqrt{e^{x_3}|x_3 - 0.662|}))$ | $\left(-\log(|x_2|) + |x_1|^{0.5}\right)(\cos(x_2) + 0.604) + \cos(x_2) + \tanh(0.142x_0)$ | $-0.974x_1^2 + |x_1|^{0.5}$ | $1.108|x_0|^{\frac{1}{4}}|x_1|^{\frac{1}{4}}$ |
| NeSymReS | — | $0.173x_0 + x_1\sin\left(0.284x_1 + \frac{x_2}{x_1}\right)$ | $\frac{0.059x_0 + x_1^2}{\cos(1.577(-0.668x_1 - 1)^3) - 1.862}$ | $\cos\left(\frac{\sin\left(\frac{x_0}{x_1}\right)}{x_0}\right) + 0.644$ |
| E2E | $0.96e^{1.21x_3} - 0.963\sin(-0.001x_0 \left(0.03 + \frac{90.6}{0.216x_1 - 0.235}\right) + 0.555x_1 + 0.03x_2 + (0.737 - 6.778x_0) (0.007x_1 - 1.38) + 0.898) + 0.055$ | $0.011x_1 - (1.57 - 0.001x_2)\arctan(-0.155x_0 + (0.055\sin(0.173x_1 + 0.031) - 0.005) (53.8\cos(0.171x_1 - 0.172(x_2 + 0.075)^2 + 0.011) + 0.006) + 0.004) - 0.006$ | $\left(11.393(x_1 - 0.128)^2 + 11.849\right) (-0.02(-0.021\cos(1254.632 (1 - 0.528x_0)^2 + 0.173) - 1)^2 -0.022)$ | $2.0 - 0.9(0.006|(3.121x_1 +0.11)(24.018(x_0 - 0.003)^2 + 6.0)| + 0.5)$ |
| uDSR | $-0.001x_0 x_3 + 0.001x_0 + 0.001x_1^2 x_3^2 - 0.002x_1^2 + 0.001x_1 x_2 + 0.003x_1 x_3 + 0.001x_1 + 0.001x_2 + 0.153x_3^4 + 0.569x_3^3 + 0.497x_3^2 + 0.599x_3 + \sin(x_0 + x_1 x_2) + 1.103$ | $-0.002x_0^3 - 0.007x_0 x_1 + 0.006x_0 x_2 + 0.26x_0 +0.039x_1^2 - 0.001x_1x_2 - 0.012x_1 + 0.003x_2^4 -0.262x_2^2 - 0.035x_2 + 3\cos(x_2) - \cos(2x_2) + 2.743$ | $0.002x_0^4 + 0.001x_0^3 x_1 - 0.029x_0^3 - 0.002x_0^2 x_1^2 + 0.002x_0^2 x_1 - 0.024x_0^2 - 0.009x_0 x_1^2 - 0.008x_0 x_1 - 0.55x_0 - 0.002x_1^4 - 0.003x_1^3 - 0.865x_1^2 + \log(e^{-x_0(\sin(x_0(x_0+1))\cos(x_0)-1)}) +0.01x_1 + 0.951$ | $-0.004x_0^4 + 0.107x_0^2 - 0.001x_0 x_2 +0.008x_1^4 - 0.18x_1^2 + 0.002x_2 -e^{\cos(x_0)} + \cos(x_0) - \cos(x_1) +2.729$ |
| TPSR | $0.988e^{1.203x_3} + 0.055\sin(x_0 -8.658x_1 + 1.093) + 0.029$ | $(0.001 + \frac{1}{x_0 - 411.157})(31.572 - 0.001x_2) (0.214\cos(2.182x_1 + 2.001) - 5.403) ((4.295 - 1.096x_0)(-0.044x_1 - 0.586) +14.767\cos(0.533x_2 - 0.05) + 9.177)$ | $-0.026x_0 - 1.254 + \frac{0.067 - 0.027(-x_1 - 0.023)^2}{0.006\sin(7.183x_0 + 2.841) + 0.03}$ | $2.01 - 1.478 e^{-0.00384|x_1(90.031x_0 + 2.401)|}$ |
| SeTGAP | $0.997e^{1.201x_3} + 0.999\sin(x_0 + 1.0x_1 x_2 + 0.017) + 0.004$ | $0.997\cos(0.202x_2^2 + 6.236)|x_1| + 1.0\tanh(0.5x_0)$ | $\frac{5.94x_1^2 - 5.979}{-5.941\sin(6.283x_0 - 6.283) - 8.91} -0.004$ | $2.0 - \frac{15.123}{15.039x_1^4 + 15.129} - \frac{15.086}{15.06x_0^4 + 15.09}$ |

Table 21: Comparison of predicted equations (E9—E13) with rounded numerical coefficients — Iteration 4

| Method | E9 | E10 | E11 | E12 | E13 |
|---|---|---|---|---|---|
| PySR | $1.141\sqrt{x_1} + 1.141(-0.895x_0 + (0.059x_0^2 - 1.08)\tanh(3.763x_0))\tanh(x_0)$ | $\sin(x_0 e^{x_1}\tanh(e^{e^{\tanh(\cosh(\tan(e^{x_1}+0.775)))}}))$ | $2x_0\log(x_1^2)$ | $x_0\cos\left(1.659 - \frac{0.088x_0x_1+x_0}{x_0x_1}\right) + 1.0$ | $\sqrt{x_0}\log(x_1^2)$ |
| TaylorGP | $0.26x_1 + \log\left(\frac{0.984}{\|x_0\|}\right) - \sqrt{\left\|\log\left(\frac{0.593}{\|x_0\|}\right) + \log\left(\left\|\tanh\left(\frac{x_1}{x_0}\right)\right\|\right)\right\|}$ | $\sin(x_0 e^{x_1})$ | $4x_0\log(\|x_1\|)$ | $\left(x_0\tanh\left(\frac{0.664}{x_1}\right) + 1\right)\sqrt{\|\tanh(x_1)\|}$ | $2\log(\|x_1\|)\sqrt{\|x_0\|}$ |
| NeSymReS | $\log\left(\frac{0.282\|x_1\|}{\|x_0\|}\right)$ | $\sin(x_0 e^{x_1})$ | $x_0\log(x_1^4)$ | $(x_0 + x_1)\sin\left(\frac{1}{x_1}\right)$ | $0.234x_0 + \log(x_1^2)$ |
| E2E | $-0.6\log(10.525(0.003 - x_0)^2 + 0.6) + 1.9 - \frac{0.6}{0.229x_1+0.235}$ | $0.001 - 1.0\sin(0.561x_0 (-0.002 + 17.5/((-88.065 - \frac{0.009}{7.629x_0+19.891})(0.001 - 0.025x_1) + 0.364(0.008 x_1 - 1)^2 - 7.514( 0.732x_1 - 1)^2 - 5.172)))$ | $-2.266x_0(0.018 - 0.26x_0(13.784(0.002 - (x_1 - 0.024)^2)^2 + 0.008))$ | $0.998 - 6.35\sin((0.006 + \frac{0.673}{0.173x_1+0.007})(-0.04x_0 - 0.002))$ | $(5.75 - 8.24/(1.265( \|x_1\|+0.008)^{0.5} + 0.135))(2.72\log(0.174x_0 + 0.991) + 0.188)$ |
| uDSR | $0.001x_0^4 - 0.095x_0^2 + 0.001x_0x_1^2 - 0.002x_0x_1 + 0.002x_0 - 0.007x_1^4 + 0.097x_1^3 - 0.497x_1^2 + 1.442x_1 - 2.803 + e^{\cos(x_0)}\sin(x_0)/x_0$ | $\sin(x_0 e^{x_1})$ | $x_0\log(7.389x_1^4) - 2x_0$ | $x_0\sin(1/x_1) + 1.0$ | $0.513e^{0.005x_0^3 - 0.076x_0^2} e^{-0.002x_0x_1} e^{0.588x_0+0.006x_1}\log(x_1^2)$ |
| TPSR | $-0.001x_0 + 1.386x_1 + 23.694\arctan(( 0.005x_1 + 0.059)(0.155x_0^2 + 37.746 (1 - 0.074x_1)^2 - 1.731\|x_0\| + 25.317) - 4.298) + 12.633$ | $0.9998\sin(0.9994x_0 e^{x_1}) - 0.000683$ | $7.994x_0 + 0.052$ | $(0.429 - 0.001x_0)(\arctan(224.422 ((x_1 + 0.121)^3 - 0.052)^2 + 2.305) - 1.24)(-0.001x_0(-0.049 - \frac{6021.0}{x_1}) + 0.081x_0 + 7.034)$ | $(0.064x_0 + 0.751 - 119.56/ ((-41.662 - 117.245e^{-0.175x_0}) (-0.108\|x_1\| - 0.014))) (0.313x_1 + 0.362(0.706 -0.561x_1)(x_1 + 2.798) - 0.513)$ |
| SeTGAP | $-1.0\log(14.156x_0^2 + 3.539) + 1.0\|\log(9.499x_1 + 4.75)\| - 0.294$ | $1.0\sin(1.0x_0 e^{x_1})$ | $2.0x_0\log(x_1^2)$ | $1.0x_0\sin\left(\frac{1}{x_1}\right) + 1.0$ | $\sqrt{1.0x_0}\log(x_1^2)$ |

Table 22: Comparison of predicted equations (E1—E4) with rounded numerical coefficients — Iteration 5

| Method | E1 | E2 | E3 | E4 |
|---|---|---|---|---|
| PySR | $0.608x_0x_1 + 1.1\sin(2.25x_0(x_1 - 0.667) - 1.5x_1 + 1.0)$ | $-0.5x_0 + (0.179x_1 + 2.984) \sin(0.2x_1) + e^{e^{\cos(0.111x_0)}} + 1.041$ | $0.15e^{1.5x_0} + 0.5\cos(3.0x_1)$ | $(\cosh(x_0) + \cosh(x_2)) (-0.005x_1 - 0.005x_3 + 0.091)$ |
| TaylorGP | $0.616x_0x_1$ | $-0.494x_0 - 0.005x_1 + 0.204x_2 + 8.593$ | $\frac{-x_0 + e^{x_0} - }{\sqrt{\left\|x_0 + 0.838\sqrt{e^{x_0}} - e^{x_0} - \sqrt{\|x_0\|}\right\|}}$ | $\frac{x_2}{\tanh(x_2)} + \log\left(\left\|\frac{x_0}{\tanh(x_0)}\right\|\right)\sqrt{\|x_0\|} - 1.797$ |
| NeSymReS | $0.59x_0x_1 + \cos(0.584(0.989 x_0 - x_1)^2)$ | $-x_0 + e^{\sin(x_2)} + 7.456 - 0.05e^{-0.102x_1}$ | $0.585e^{x_0} - 0.437\sin\left(1.612(0.04x_1 + 1)^2\right)$ | — |
| E2E | $0.021x_0(29.332x_1 - 0.194) + 1.1\cos(30.949 x_1 - 0.049) + 0.01$ | $-0.016x_0 + 0.049x_2 + (0.086x_0 - 0.704)(0.75x_0 - 0.015) + (0.003x_1 + 0.052)(-0.97x_1 + 53.2\sin(0.221x_2 + 0.035) + 0.064) + 6.444$ | $0.224e^{0.188x_0} e^{(0.004x_0+0.045)(22.712x_0+0.009x_1-5.15)} + 0.514\cos(3.152x_1 + 0.082) + 0.043$ | $0.002\|30.73x_3 + 5.334(x_0 + 0.019)^4 + 6.811(x_1 - 0.94(x_2 - 0.021)^2 + 0.028)^2 + 1.185\|$ |
| uDSR | $(0.001x_0^3 - 0.001x_0^2x_1^2 - 0.008x_0^2x_1 + 0.025x_0^2 - 1.824x_0x_1^2 + 0.608x_0x_1 - 0.024x_0 - 0.006x_1^3 + 0.022x_1^2 + 0.145x_1 - 0.225)\sin\left(\frac{1}{1-3x_1}\right)$ | $\log(666.967e^{0.063x_0^2 - 0.503x_0} e^{0.03x_1x_2 - 0.001x_1} e^{-0.002x_2^3+0.541x_2} + 1)$ | $0.075x_0^4 + 0.532x_0^3 + 0.001x_0^2x_1^2 + 0.023x_0^2 + 0.002x_0x_1^2 - 0.002x_0x_1 - 2.746x_0 - 0.129x_1^4 + 0.001x_1^3 + 0.97x_1^2 - 0.003x_1 + e^{\cos(2x_1)} + 3\sin(x_0) - 1.783$ | $0.01x_0^4 - 0.02x_0^2x_1 + 0.01x_1^4 + 0.01x_2^4 - 0.02x_2^2x_3 + 0.01x_3^2$ |
| TPSR | $-0.6\log(12.904(x_0 + 0.008)^2 + 0.6) + 1.9 - \frac{0.6}{0.227x_1+0.236}$ | $(0.372x_0 - 0.368x_2 + 123.99) ((0.001x_0 + 0.127)(x_0 - 0.045x_2 - 254.229) + 0.086 \arctan(0.002x_2(2.01x_1 + 71.9)) + 32.38)$ | $0.052e^{2.5x_0} + 0.5\cos(4.0x_1 - 0.03) + 0.09$ | $-0.13x_1 + 0.032x_3 + 0.081(0.067x_1 + 0.005x_2 + 0.363x_3 - (1 - 0.032 \|7.087x_0 + 28.484\|)^2 - 0.334 (x_2 + 0.005)^2 - 0.028)^2 + 0.319$ |
| SeTGAP | $0.607x_0x_1 - 1.1\sin((2.252x_0 - 1.492) (x_1 - 0.666) + 9.432)$ | $-0.5x_0 + 0.063x_0^2 + 3.16\sqrt{0.1x_1 + 1} \sin(0.202x_2) + 6.499$ | $0.15e^{1.5x_0} + 0.5\sin(3.0x_1 + 1.571)$ | $-0.02x_0^2x_1 + 0.001x_0^2 + 0.01x_0^4 + 0.01x_1^2 - 0.02x_2^2x_3 + 0.01x_2^4 + 0.01(-x_3)^2 - 0.001$ |

Table 23: Comparison of predicted equations (E5—E8) with rounded numerical coefficients — Iteration 5

| Method | E5 | E6 | E7 | E8 |
|---|---|---|---|---|
| PySR | $e^{1.2x_3} + \sin(x_0 + x_1x_2)$ | $0.52\cos(0.2x_2^2) + \tanh(x_0) \tan\left(\cos\left(\tanh\left(\frac{1.9}{x_1}\right)\right) + 0.534\right)$ | $4.26x_1^2(\tanh(\sin(6.283 x_0) + 1.8) - 1.1) + 0.967$ | $\sinh(0.406\cosh(\tanh(x_0 \sinh(\tanh(\sinh(x_0)))))) + \tanh(\cosh(0.895x_1)) + 0.69) - 3.031$ |
| TaylorGP | $e^{x_3}\|x_3\|^{0.5} + 0.487$ | $\frac{x_1\sin(x_2)}{x_2\tanh(x_1)} - 1.185\cos\left(e^{\|x_2\|^{0.5}}\right)$ | $-x_1^2 + \sqrt{\|x_1\|}$ | $1.07\|x_0x_1\|^{\frac{1}{4}}$ |
| NeSymReS | — | $-0.376x_0 + x_1\sin\left(\frac{x_0}{x_1}\right)$ | $\frac{-3967.593x_0 + x_1^2}{\cos(65757.558(-x_1 - 0.157)^3) + 28400.518}$ | $\cos\left(\frac{\sin\left(\frac{x_0}{x_1}\right)}{x_0}\right) + 0.653$ |
| E2E | $0.903e^{1.231x_3} + 0.105 + 1.0\cos(0.764/(0.062 \sin((-0.619x_0 - 0.079)(0.344x_1 - 0.212x_2 - 0.857)) - 0.007))$ | $(-0.001\|4.547x_1 - 0.068\| - 0.601) \arctan(8.32x_0 - 242.537(0.016x_0 + 1)^3 + 7.105) + 0.005\cos(0.667 (-x_2 - 0.162)^2 - 4.352) - 0.004$ | $(0.09 - 0.083(x_1 - 0.002)^2)(6.724(1 - 0.756 \sin(6.11x_0 - 0.012))^2 + 4.0)$ | $0.536 - 0.903\arctan((0.046 x_1 - 0.991)(-0.367\sin(2.487 \sqrt{(-\|32.358x_1 + 1.7\| - 0.017)^2 + 0.006)} -49.4) + 0.017\|(0.053(x_0 - 0.007)^2 + 0.016)(0.001x_1 + 425.696 (x_1 + 0.011)^2 - 22.805)\| - 0.286))$ |
| uDSR | $0.002x_0x_3 - 0.001x_0 + 0.003x_1x_2 - 0.001x_1x_3^4 + 0.001x_1x_3 + 0.002x_1 - 0.003x_2 + 0.151x_3^4 + 0.565x_3^3 + 0.51x_3^2 + 0.619x_3 + \sin(x_0 + x_1x_2) + 1.086$ | $-0.002x_0^3 + 0.004x_0^2 + 0.001x_0x_2^2 - 0.003x_0x_2 + 0.233x_0 + 0.034x_2^2 - 0.003x_1x_2 - 0.013x_1 + 0.003x_2^{\frac{1}{4}} - 0.264x_2^2 + 0.011x_2 + 3\cos(x_2) - \cos(2x_2) + 2.617$ | $-0.002x_0^4 - 0.001x_0^3x_1 - 0.042x_0^3 + 0.001x_0^2x_1^2 - 0.001x_0^2x_1 + 0.037x_0^2 - 0.001x_0x_1^3 - 0.012x_0x_1^2 + 0.026x_0x_1 + 0.587x_0 - 0.003x_1^3 - 0.915x_1^2 + 0.077x_1 - (x_0 - \sin(x_1))\sin(x_0^2) + 0.867$ | $-0.004x_0^4 + 0.109x_0^2 + 0.008x_1^4 - 0.179x_1^2 - 0.001x_1 - e^{\cos(x_0)} + \cos(x_0) - \cos(x_1) - 0.001x_0 + 2.72$ |
| TPSR | $0.97e^{1.211x_3} + 0.087\cos(x_1 + 3.732x_2 + 10.665) + 0.027$ | $(0.007x_1 + 0.004)(2.087x_1 + 1.792) + 3.762\sin(0.0044x_0 + 2.3489 \cos(0.523x_2 + 0.03) - 0.73097) + 1.077$ | $1.075 - 0.971(-0.022x_0 - x_1 + 0.096\cos(x_0 + 1.524) + 0.078)^2$ | $(-0.004x_1 - 0.001) (x_1 + 0.002) + 2.063 - 1.494e^{-0.001\|(5.976x_0 - 0.065)(59.997x_1 + 2.475)\|}$ |
| SeTGAP | $0.977e^{1.201x_3} + 1.0\sin( x_0 + x_1x_2 + 18.85) + 0.001$ | $1.0\cos(0.2x_2^2 + 6.282)\|x_1\| + 1.0\tanh(0.5x_0)$ | $(0.947 - 0.948x_1^2) ((\sin(6.283x_0) + 1.478)^{-1.0} + 0.025) + 0.02$ | $2.0 + \frac{15.997}{(0.026x_1^3 - 15.998x_1^4 - 15.994)} - \frac{16.587}{(-0.001x_0 + 0.077x_0^2 + 0.012x_0^3 + 16.459x_0^4 + 16.583)}$ |

Table 24: Comparison of predicted equations (E9—E13) with rounded numerical coefficients — Iteration 5

| Method | E9 | E10 | E11 | E12 | E13 |
|---|---|---|---|---|---|
| PySR | $\sqrt{1.148x_1} + \cosh(1.354\cos(x_0)$ $\cos(\tanh(x_0))) - 2.202\cosh(1.613$ $\tanh(0.481x_0))$ | $\sin\left(x_0 e^{x_1}\right)$ | $x_0(0.943 - 2.768\log($ $(\log(\cosh(1.303$ $\tanh(0.652x_1))))^{0.5})) +$ $3.385x_0$ $\log(\log(\cosh(x_1)))$ | $1.0x_0 \sin\left(\frac{1.0}{x_1}\right) + 1.0$ | $\sqrt{x_0}\log\left(x_1^2\right)$ |
| TaylorGP | $-\log(|x_0|) + \tanh(\log(|x_1|)) -$ $\sqrt{\left|\log\left(\frac{0.603}{|x_0|}\right)\right|}$ | $\frac{\tanh(\sin(e^{x_1}\sin(x_0))}{\sqrt{|\cos(\sin(x_1))|}}$ | $3.947x_0\log(|x_1|)$ | $\frac{1.217x_0}{1.229x_1 + \frac{0.471}{x_1}} + 0.961$ | $2\log(|x_1|)\sqrt{|x_0|}$ |
| NeSymReS | $\log\left(\frac{0.28|x_1|}{|x_0|}\right)$ | $\sin\left(x_0 e^{x_1}\right)$ | $x_0\log\left(x_1^4\right)$ | $(x_0 + x_1)\sin\left(\frac{1}{x_1}\right)$ | $0.252x_0 + \log\left(x_1^2\right)$ |
| E2E | $-0.6\log(12.904(x_0 + 0.008)^2 + 0.6)$ $+1.9 - \frac{0.6}{0.227x_1 + 0.236}$ | $0.001 - 1.0\sin(0.257x_0$ $(0.1 - 3.597e^{1.08x_1}))$ | $0.257x_0(30.0\log(0.19$ $(|21.905x_1 + 0.102| +$ $0.001)^{0.5} - 0.01) + 0.01)$ | $1$ | $(0.031 - 0.44\log(23.696$ $(-x_1 - 0.009)^2(x_1 + 0.201)^2$ $(0.033x_0 + 0.086|0.501x_0$ $-7.823| - 1)^2 + 0.02))$ $(-0.001x_0 - 8.705)$ $(0.032x_0 + 0.057)$ |
| uDSR | $\log((x_1 + \cos(1))(0.007x_0^2 x_1^2 + 0.809$ $-0.054x_0^2 x_1 + 2.112x_0^2 + 0.003x_0 x_1^2$ $-0.006x_0 x_1 + 0.002x_0 + 0.012x_1^4$ $-0.145x_1^3 + 0.588x_1^2 - 0.851x_1))$ | $\sin\left(x_0 e^{x_1}\right)$ | $2.0x_0\log\left(x_1^2\right)$ | $\left(1.0x_0^2 \sin(1/x_1) + x_0\right)/x_0$ | $(0.001x_0^3 - 0.025x_0^2 + 0.292x_0$ $+0.244)\log\left(x_1^2\right)/\cos(1)$ |
| TPSR | $(0.187 - 0.012x_1)(0.179x_1 - 2.902)$ $((-2.035 + \frac{4.906}{0.282|x_0 + 0.001| + 1.048})$ $(-0.655x_1 - 3.76) - 0.296|x_0| + 8.821)$ | $1.0\sin\left(1.0x_0 e^{x_1}\right)$ | $(-0.318x_0 - 0.004)$ $(-0.392x_1^2 - 15.221 -$ $\frac{30.882}{-0.017|98.801x_1 + 0.806| - 0.484})$ | $-10.226x_0 + 0.038x_1 + (x_0 +$ $4639.596 + \frac{371.436}{\arctan(440.318x_1 + 15.611)})$ $(0.006x_0(-0.002x_1 - 141.195)$ $+0.796x_0 - 0.001) + 6.082$ | $(-0.029 + 19.719/(0.422$ $|(0.45x_0 + 23.971)(4.248x_1$ $+0.008) + 7.133)(3.329 -$ $3.344(x_1 + 0.002)^2)$ $(-0.207x_0 - 0.669)$ |
| SeTGAP | $-1.001\log\left(18.044x_0^2 + 4.522\right) +$ $0.994|\log(18.83x_1 + 9.24)| - 0.705$ | $1.0\sin\left(1.0x_0 e^{x_1}\right)$ | $2.0x_0\log\left(x_1^2\right)$ | $1.0x_0\sin\left(\frac{1}{x_1}\right) + 1.0$ | $1.0\sqrt{x_0}\log\left(x_1^2\right)$ |

Table 25: Comparison of predicted equations (E1—E4) with rounded numerical coefficients — Iteration 6

| Method | E1 | E2 | E3 | E4 |
|---|---|---|---|---|
| PySR | $0.607x_0 x_1$ | $-0.499x_0 + e^{\tanh(e^{x_1})}\tanh(x_2) +$ $\log(\cosh(x_0) + 29.144) + 3.287$ | $0.15e^{1.5x_0} + 0.5\cos(3x_1)$ | $(0.09 - 0.009x_1)\cosh(x_0) +$ $(0.091 - 0.009x_3)\cosh(x_2)$ |
| TaylorGP | $0.62x_0 x_1$ | $-0.497x_0 - 0.002x_1 + 0.208x_2 + 8.571$ | $-x_0 + e^{x_0} - \sqrt{|x_0|} - 0.506 +$ $\tanh(x_0 - e^{x_0} + \sqrt{|e^{x_0} - 0.977|}$ $+0.977)$ | $\sqrt{\log(|x_0|)}\sqrt{\left|\frac{x_2\left(-x_2\sqrt{|x_0|} + \log(|x_2|) + 0.553\right)}{\tanh(x_2)}\right|}$ |
| NeSymReS | $0.667x_0 x_1 + \cos(0.066$ $(-x_0 - 0.869x_1)^2)$ | $93.845e^{0.002x_2} + e^{\sin\left(\frac{x_0}{x_1}\right)} - 86.601$ | $0.58e^{x_0} -$ $0.464\sin\left(1.482(-0.071x_1 - 1)^2\right)$ | — |
| E2E | $0.024x_1(25.373x_0$ $-0.001x_1 + 0.057) -$ $1.1\cos(76.2\tan(2.983$ $x_1 - 4.272) + 0.767)$ $+0.002$ | $(35.4 - 0.001x_0)(0.022(0.299x_0 - 1)^2$ $+(0.013x_1 + 0.301)(0.319\sin((0.145x_1$ $+3.599)(0.052x_2 - 0.004)) + 0.061)$ $+0.163)$ | $0.212e^{1.379x_0} +$ $0.475\cos(3.108x_1 - 0.004) - 0.013$ | $0.018(0.573x_1 - (0.033x_0 - 0.002)(21.569x_0$ $-1.005) + 0.036)^2 + 2.72(0.01\arctan(-$ $(0.079e^{\frac{0.005}{18.338x_2 - 4.735}} - 199.276)e^{-\frac{0.005}{18.338x_2 - 4.735}}$ $+0.001)((1.851x_2 - 2.372)(0.283x_3$ $-0.03) + 0.25((x_2 + 0.005)^2 - 0.003)^2$ $+0.075)| + 0.001$ |
| uDSR | $(-0.001x_0^4 + 0.001x_0^3 x_1 - 0.008x_0^3$ $-1.822x_0^2 x_1 + 0.023x_0^2 + 0.001x_0 x_1^3 +$ $0.001x_0 x_1^2 + 0.584x_0 x_1 + 0.089x_0 +$ $0.003x_1^3 - 0.015x_1^2 - 0.075x_1$ $+0.041)\sin\left(\frac{x_0}{-3x_0^2 + x_0}\right)$ | $0.062x_2^2 - 0.5x_0 + 0.029x_1 x_2 - 0.001x_1 -$ $0.002x_2^2 + 0.001x_2^2 + 0.471x_2 -$ $\cos\left((x_2 + \log(x_2 + e^{-x_2}))/x_2\right) + 7.094$ | $0.075x_0^4 + 0.335x_0^3 + 0.019x_0^2 +$ $0.001x_0 x_1^3 - 0.004x_0 x_1 - 0.987x_0 -$ $0.014x_1^4 + 0.001x_1^3 + 0.231x_1^2 - 0.004x_1 +$ $\sin(x_0) + \cos(x_1)\cos(2x_1) - 0.272$ | $0.01x_1^4 - 0.02x_0^2 x_1 + 0.01x_1^2 + 0.01x_2^4$ $-0.02x_2^2 x_3 + 0.01x_3^2 - \sin(1) + 0.842$ |
| TPSR | $0.074x_1(8.206x_0 +$ $0.136) + 0.043\sin(4.77$ $x_1 - 0.485) + 0.001$ | $0.037x_1 - 0.134x_2 + 0.766|9.705x_0 +$ $93.72 + \frac{94.187}{9.425e^{0.022x_0} + 0.035x_1 - 0.524x_2 + 9.862}|$ $-1026.309 + \frac{117.7}{0.001x_0 + 0.123}$ | $0.15e^{1.5x_0} + 0.5\cos(3.0x_1)$ | $0.003(-0.005x_2 + (0.363 - 0.003x_3)$ $((-0.941x_2 - 1.086)(-1.089x_1 + x_2 + 0.284) -$ $0.001(x_0(0.52x_3 + 69.999) + 0.001x_0 +$ $0.677)^2 - 0.238) - (1 - 0.025x_3)^2(0.148x_1$ $+x_2 - 0.552)^2 - 0.001)^2 + 0.524$ |
| SeTGAP | $0.608x_0 x_1 - 1.099$ $\sin((2.24x_0 - 1.544)$ $(x_1 - 0.67) - 9.46)$ | $-0.5x_0 + 0.063x_0^2 +$ $3.162\sqrt{0.1x_1 + 1}\sin(0.2x_2) + 6.498$ | $0.149e^{1.502x_0} -$ $0.5\sin(3.0x_1 + 4.712)$ | $-0.004x_0 - 0.02x_0^2 x_1 + 0.017(-x_0)^2 +$ $0.009(-x_0)^4 + 0.008(-x_1)^2 +$ $0.01x_1^4 + 0.011x_3^2 - (0.013x_2^2 + 0.003)$ $(1.575x_3 + 0.614) - 0.023$ |

Table 26: Comparison of predicted equations (E5—E8) with rounded numerical coefficients — Iteration 6

| Method | E5 | E6 | E7 | E8 |
|---|---|---|---|---|
| PySR | $e^{1.2x_3} + \sin(x_0 + x_1 x_2)$ | $\frac{2.535e^{\tan\left(1.058\cos\left(2.528\tanh\left(\frac{2}{x_1}\right)\right)\right)}}{\cos\left(0.2x_2^2\right) + \tanh(x_0)}$ | $\frac{1.0 - 1.0x_1^2}{\sin(6.283x_0) + 1.5}$ | $\sinh(\sinh(\sinh(\tanh(\sinh(\cosh($ $\tanh(x_0\tanh(x_0)))$ $\cosh(\tanh(x_1\tanh(x_1))) - 0.718)))))$ |
| TaylorGP | $e^{x_3}|x_3|^{0.5} + \tanh((x_3 -$ $0.891)e^{x_3} + 1.718e^{x_3})$ | $-1.185\cos\left(e^{|x_2|^{0.5}}\right) + \frac{4.854\sin(x_2)}{x_2}$ | $-x_1^2 +$ $\log(|x_1^2 - \log(|x_1^2 -$ $\sqrt{|x_1(x_1 - 0.073)|}|)|)$ | $1.102|x_0 x_1|^{\frac{1}{4}}$ |
| NeSymReS | — | $-0.368x_0 + x_1\sin\left(\frac{x_0}{x_1} - 0.001x_2\right)$ | $\frac{0.184x_0 + x_1^2}{\cos(1.965x_0 - x_1) - 1.94}$ | $\cos\left(\frac{\sin\left(\frac{x_0}{x_1}\right)}{x_0}\right) + 0.643$ |
| E2E | $0.002x_2 + 1.106e^{1.147x_3} +$ $0.774\cos(0.002x_1) + 0.942$ $\cos(28.703x_2 - 0.956) - 0.977$ | $0.942$ | $\left(6.284(x_1 - 0.007)^2 - 7.49\right)$ $(-0.082(0.01x_1 + \sin(6.457x_0 +$ $0.14) - 0.705)^2 - 0.098)$ | $2.06 - 30.5/(6.892(0.047 - x_0)^2$ $(x_1 - 0.003)^2 + 18.24((1 - 0.006x_0)^2)^2$ $-9.27\cos(1.329x_1 - 44.203) + 10.058)$ |
| uDSR | $0.001x_0 x_3^2 + 0.001x_0 x_3 + 0.002x_0$ $-0.001x_1^2 + 0.001x_1 x_2 x_3 + 0.001x_1 x_3 -$ $0.001x_1 + 0.002x_2^2 + 0.004x_2 x_3 +$ $0.003x_2 + 0.151x_3^4 + 0.567x_3^3 +$ $0.519x_3^2 + 0.619x_3 +$ $\sin(x_0 + x_1 x_2) + 1.083$ | $-0.002x_0^3 + 0.006x_0^2 + 0.003x_0 x_2$ $+0.238x_0 - 0.001x_1^2 x_2^2 + 0.001x_1^2 x_2$ $+0.042x_1^2 + 0.001x_1 x_2^2 + 0.002x_1 x_2 -$ $0.029x_1 + 0.003x_2^4 - 0.258x_2^2 -$ $0.007x_2 + 3\cos(x_2) - \cos(2x_2) + 2.536$ | $0.002x_0^4 - 0.004x_0^3 x_1 - 0.019x_0^3 -$ $0.003x_0^2 x_1 - 0.042x_0^2 - 0.001x_0 x_1^3 -$ $0.01x_0 x_1^2 + 0.061x_0 x_1 + 0.294x_0 -$ $0.003x_1^4 + 0.001x_1^3 - 0.851x_1^2 +$ $0.047x_1 + \sin(6x_0) + 0.909$ | $0.008x_0^4 - 0.18x_0^2 - 0.001x_0 - 0.004x_1^4$ $+0.108x_1^2 - e^{\cos(x_1)} - \cos(x_0) +$ $\cos(x_1) + 2.727$ |
| TPSR | $0.98e^{1.207x_3} - 0.039\sin(-x_1 +$ $3.302x_2 + 18.63) + 0.022$ | $(-0.068x_0 - 1.576\cos(0.538x_2)$ $-0.674)(0.016803x_0 + 0.001x_1 +$ $4.731\sin(0.015(0.326 - x_1)^2$ $+3.728) + 2.043)$ | $\frac{412.541 - 411.968}{\left(0.001(x_1 + 0.002)^2 + 1\right)^2}$ | $(0.001 - 0.003x_1)(x_1 - 0.002) + 2.057 -$ $1.494e^{-0.001|(5.993x_0 + 0.026)(59.999x_1 - 1.554)|}$ |
| SeTGAP | $0.995e^{1.201x_3} + 1.0\sin(x_0 +$ $x_1 x_2 - 6.283) + 0.002$ | $0.999\cos\left(0.2x_2^2 - 0.011\right)|x_1| +$ $1.0\tanh(0.5x_0)$ | $\left(-0.001 + \frac{0.999}{\sin(6.283x_0 - 9.424) - 1.5}\right)$ $\left(x_1^2 - 0.977\right) + 0.018$ | $-\frac{16.085}{\left(16.038x_1^2 + 2.016.09\right)} + 2.0 +$ $\frac{7.918}{\left(0.158x_0^2 - 8.067x_0^4 - 7.943\right)}$ |

Table 27: Comparison of predicted equations (E9—E13) with rounded numerical coefficients — Iteration 6

| Method | E9 | E10 | E11 | E12 | E13 |
|---|---|---|---|---|---|
| PySR | $\sinh(\sinh(2.7$ $e^{-\frac{0.259x_0^2}{x_1+0.727}-\frac{0.178x_0}{(x_1+1.017)\tanh(x_0)}}$ $-1.154))-1.787$ | $\sin(1.0x_0e^{x_1})$ | $2x_0\log(x_1^2)$ | $x_0\sin\left(\frac{1}{x_1}\right)+1.0$ | $\sqrt{x_0}\log(x_1^2)$ |
| TaylorGP | $-0.683\log(|x_0|)-$ $1.618\sqrt{|x_0|}+\sqrt{|x_1|}$ | $\sin(x_0e^{x_1})$ | $2.458x_0\log(|x_1|)+$ $(1.632x_0-0.908)\log(|x_1|)+$ $0.875\cos(3.872x_0+(2.732$ $x_0-2.481)\log(|0.754x_1$ $\tanh(x_1)|x_0|^{1.5}+0.307|)+$ $(4.458x_0-2.481)\log(|x_1|))$ | $0.806x_0\sin\left(\frac{1.229}{x_1}\right)$ $+0.806$ | $(\log(|x_0|)\log(|x_1|)+\tanh(\log(|x_1|)))$ $(\log(|x_0\tanh(\log(|x_1|))+\log(|x_1|)-$ $0.13|^{0.5})+|\tanh(x_0\tanh($ $\log(6.162|\log(|x_1|)|)))|^{0.5})$ |
| NeSymReS | $\log\left(\frac{0.281|x_1|}{|x_0|}\right)$ | $\sin(x_0e^{x_1})$ | $x_0\log(x_1^4)$ | $(x_0+x_1)\sin\left(\frac{1}{x_1}\right)$ | $0.245x_0+\log(x_1^2)$ |
| E2E | $2.964(0.2x_1-1)^3+4.352-$ $0.792\log(166.831$ $(x_0-0.052)^2+17.6)$ | $1.0\sin(0.005x_0$ $(167.868e^{1.143x_1}-$ $0.01))+0.001$ | $(2.024x_0-0.006)(0.75$ $\log(1.946(-x_1-0.005)^2$ $+0.05)+0.068)$ | $7.12\sin(0.001x_0$ $(0.003+\frac{31.2}{0.173x_1+0.001}))$ $+0.996$ | $(0.004-5.12\log(0.175x_0+1.109))$ $(0.945((|0.004-\frac{49.1}{0.326x_1-0.001}|+$ $0.045)^{0.5}-0.898)^{0.5}-3.44)$ |
| uDSR | $\log\left(\frac{2x_1+1}{4.0x_0^2+1.0}\right)$ | $\sin(x_0e^{x_1})$ | $x_0\log(1.851x_1^4\cos(1))$ | $\left(1.0x_0^2\sin(1/x_1)+x_0\right)/x_0$ | $\log(x_1^2)\log(0.004x_0^3+$ $0.06x_0^2+1.19x_0+0.001x_1+1.419)$ |
| TPSR | $\left(-0.01+\frac{0.001}{x_0+11.073}\right)(3.527-$ $0.23x_1)(x_1-15.288)((3.097$ $-6.725/(0.000341|(x_0-0.003)$ $(0.02x_1+0.399)(0.954x_1+$ $19.312)|+0.016)(-0.005x_1-$ $0.063)+0.665|x_0|-25.78$ | $1.0\sin(0.999x_0e^{x_1})$ | $(26.232-0.008x_1)$ $(-0.523x_0-0.01)$ | $2.867x_0+1.066$ | $(0.023-\frac{0.25}{0.829|x_1-0.001|+0.087})$ $(0.948-0.97x_1)(43.582-0.84x_0)$ $(0.055x_0+0.137)(0.494x_1+0.482)$ |
| SeTGAP | $-1.0\log(11.241x_0^2+2.809)+$ $1.0|\log(9.824x_1+4.914)|$ $-0.559$ | $1.0\sin(1.0x_0e^{x_1})$ | $2.0x_0\log(x_1^2)$ | $1.0x_0\sin\left(\frac{1}{x_1}\right)+1.0$ | $2.0\sqrt{x_0}\log(|x_1|)$ |

Table 28: Comparison of predicted equations (E1—E4) with rounded numerical coefficients — Iteration 7

| Method | E1 | E2 | E3 | E4 |
|---|---|---|---|---|
| PySR | $0.608x_0x_1+$ $1.1\sin((x_0-0.667)$ $(x_1-0.667))$ | $3.017\sin(0.2x_2)+$ $3.017\cosh(0.162x_0-0.723)+2.818$ | $0.15e^{1.5x_0}+0.5\cos(3.0x_1)$ | $-0.02x_0^2x_1-0.02x_2^2x_3+$ $0.091\cosh(x_0)+$ $0.091\cosh(x_2)$ |
| TaylorGP | $0.607x_0x_1$ | $-0.499x_0-0.008x_1+0.207x_2+8.586$ | $-2x_0+e^{x_0}-0.363$ | $0.756x_2\tanh(x_2)+$ $0.756\log(|x_0|)\sqrt{|x_0|}$ |
| NeSymReS | $0.61x_0x_1-\sin(0.001$ $(0.965x_0-1)^2)$ | $96.368e^{0.002x_2}+e^{\sin(x_0x_1)}-89.13$ | $0.579e^{x_0}-$ $0.437\sin\left(1.633(0.043x_1+1)^2\right)$ | — |
| E2E | $0.598x_0x_1-0.004x_0$ $+0.001x_1+0.143$ | $(0.011x_2+0.445)(0.174x_1-0.051x_2$ $+5.52)(0.387(0.235x_0-1)^2+\sin($ $0.225x_2+0.023)+0.144)+4.88$ | $((7.16(0.001x_0-1)^2+0.01)$ $(0.198e^{1.42x_0}+0.519\cos(3.037x_1$ $-0.034-0.072)+0.057)/$ $(7.16(0.001x_0-1)^2+0.01)$ | $0.002|5.04((x_0+0.033)^2-$ $0.001)^2+5.21(0.818x_1+$ $0.67x_3-(x_2+0.025)^2+$ $0.01)^2-0.11|+0.001$ |
| uDSR | $(-0.001x_0^2+0.606x_0x_1+0.009x_0$ $+0.001x_1^3-0.01x_1^2-0.018x_1$ $+0.022)\cos\left(e^{-3x_0^2+x_0}\right)$ | $0.063x_0^2-0.5x_0+0.029x_1x_2-0.002x_2^3$ $+0.001x_2^2+0.498x_2+\cos\left(\frac{x_2}{x_2e^{x_2}+x_2}\right)+5.677$ | $0.013x_0^4-0.033x_0^3-0.411x_0^2-$ $0.001x_0x_1-0.898x_0-0.014x_1^4$ $+0.23x_1^2+e^{x_0}+$ $\cos(x_1)\cos(2x_1)-1.306$ | $0.01x_0^4-0.02x_0^2x_1-$ $1.0x_0^2-1.0x_0x_3+x_0(x_0+x_3)$ $+0.01x_1^2+0.01x_2^4-$ $0.02x_2^2x_3+0.01x_3^2$ |
| TPSR | $0.011x_1(56.515x_0$ $+1.528)-0.031\sin($ $4.588x_1+0.776)$ $-0.005$ | $(2.621(0.060401x_0-0.019x_2-1.0-$ $\frac{0.1813}{-0.544e^{-2.096x_1+6.406x_2-1.507}})^2+0.508)$ $(0.073x_0+0.011x_1-0.028x_2+2.346)$ | $0.15e^{1.5x_0}+$ $0.5\cos(3.0x_1)$ | $0.198(-x_2-0.027)^2+$ $0.127(0.236x_1-0.234(0.006$ $-x_0)^2-1)^2-0.662$ |
| SeTGAP | $0.607x_0x_1+$ $1.098\sin((2.24$ $x_0-1.544)(x_1-$ $0.67))-0.002$ | $-0.501x_0+0.064x_0^2+(3.166$ $(0.1x_1+1)^{0.5}+0.001)$ $\sin(0.201x_2)+6.452$ | $0.151e^{1.498x_0}-$ $0.5\sin(3.001x_1+4.712)$ | $-0.02x_0^2x_1+0.01x_2^4-0.021$ $x_3(x_2)^2+0.005x_3+0.01(x_0)^4$ $+0.01(x_1)^2-(0.004x_0-0.708)$ $(0.014x_3^2-0.019)+0.028$ |

Table 29: Comparison of predicted equations (E5—E8) with rounded numerical coefficients — Iteration 7

| Method | E5 | E6 | E7 | E8 |
|---|---|---|---|---|
| PySR | $e^{1.2x_3}+\sin(x_0+x_1x_2)$ | $x_1\cos(0.2x_2^2)\tanh(236.582x_1)+$ $\tanh(0.496x_0)$ | $(1.863x_1^2-1.843)(\tanh(\sinh($ $\sin(6.281x_0))+1.345)-1.215)$ | $1.644\log(0.976/(\cos(\tanh(1.094x_0$ $\tanh(\sinh(\sinh(x_0)))))$ $\cos(\tanh(0.812\sinh(x_1)))))$ |
| TaylorGP | $2e^{x_3}-\log(|-3e^{x_3}+\log(|$ $2.287e^{x_3}-1.447\log(|\sin(x_3$ $)|)|)+1.447\log(|\sin(x_3)|)|)$ | $\frac{x_1\sin(x_2)}{x_2\tanh(x_1)}+\cos(x_2)+\tanh(x_0)$ | $-x_1^2+\log\left(\left|x_1^2-\log\left(\left|x_1^2-0.828\sqrt{|x_1^2|}\right|\right)\right|\right)$ | $\tanh((x_0+0.504)\tanh(x_0))+$ $\tanh(x_1^2-0.017)$ |
| NeSymReS | — | $-0.356x_0+x_1\sin\left(\frac{x_0}{x_1}-0.009x_2\right)$ | $\frac{0.08x_0+x_1^2}{\cos(2.445(0.924x_1-1)^2)-1.916}$ | $\cos\left(\frac{\sin\left(\frac{x_0}{x_1}\right)}{x_0}\right)+0.695$ |
| E2E | $1.138e^{1.163x_3}-0.123+0.953$ $\cos(22.045x_2-14.032+1.11/$ $(-0.742x_0+0.002x_1+19.256)+$ $2.11/((31.854x_2+0.793)(0.001x_3$ $+2.26))-0.033/(3.382x_2+0.147))$ | $0.806$ | $(0.368-\frac{4.78}{8.105-0.291x_1})(0.338x_1+23.594)$ $(0.348x_1-4.637)(0.136(0.019-x_1)^2$ $-0.117)(-0.219(1-0.329$ $\cos(6.186x_0-7.784))^3-0.046)$ | $2.1-0.9/(0.03(0.309x_0+0.074)$ $(0.687x_0-0.059)(2.086x_1-0.139)$ $(17.421x_1-0.429)|+0.6)$ |
| uDSR | $-1.0x_0^2-0.999x_0x_1-0.001x_0x_2+0.001x_0x_3$ $+x_0(x_0+x_1)-0.001x_1^2x_3-0.002x_1^2$ $-0.002x_1x_2-0.001x_1x_3-0.005x_1-0.001x_2^2$ $-0.003x_2x_3-0.001x_2+0.152x_3^4+0.567x_3^2$ $+0.512x_3^2+0.62x_3+\sin(x_0+x_1x_2)+1.1$ | $-0.002x_0^3-0.001x_0^2x_1+0.008x_0^2-$ $0.003x_0x_1+0.258x_0+0.035x_1^2+$ $0.004x_1x_2+0.029x_1+0.003x_2^4-$ $0.249x_2^2-0.009x_2+e^{\cos(x_2)-\cos(2x_2)}+$ $2\cos(x_2)+1.093$ | $-0.002x_0^3x_1-0.036x_0^3+0.002x_0^2x_1^2-0.004x_0^2+$ $0.001x_0x_1^3-0.006x_0x_1^2+0.028x_0x_1$ $+0.483x_0-0.001x_1^4-0.011x_1^3-0.892x_1^2$ $+0.114x_1-x_0\sin(x_0^2)+0.946$ | $0.008x_0^4-0.179x_0^2-0.004x_1^4$ $+0.108x_1^2+0.002x_1-e^{\cos(x_1)}$ $-\cos(x_0)+\cos(x_1)-0.001x_0+2.722$ |
| TPSR | $63.518$ $e^{(0.5-0.15x_3)(-0.068\cos(x_1+1.23x_2+1)-8.195)}$ $-0.024$ | $(-0.396\cos(0.876x_2-0.03)$ $-0.167((-0.016x_2-0.304)(x_0-$ $13.652x_2+269.341)+\cos(2.997$ $x_1+0.599)+70.235)$ | $-0.09x_0-0.01x_1-$ $(90.02x_1+6.533)+0.94$ | $2.011-$ $1.427e^{-0.0039|x_1(98.999x_0+0.009)|}$ |
| SeTGAP | $0.999e^{1.201x_3}+1.0\sin(x_0+$ $x_1x_2-12.566)+0.003$ | $1.0\cos(0.2x_2^2-6.275)|x_1|+$ $1.0\tanh(0.501x_0)$ | $\frac{2.907x_1^2-2.919}{2.904\sin(6.283x_0+3.148)-4.358}$ | $2.0-\frac{14.033}{13.947x_1^4-0.122x_0^3+14.032}-$ $\frac{14.438}{14.532x_1^4+14.434}$ |

Table 30: Comparison of predicted equations (E9—E13) with rounded numerical coefficients — Iteration 7

| Method | E9 | E10 | E11 | E12 | E13 |
|---|---|---|---|---|---|
| PySR | $0.226\cos(\cos(x_0)) + \sinh(\sinh(\sinh(0.974\cos(0.643x_0 + 0.167)))) - \frac{8.714}{x_1 + 2.537}$ | $\sin(1.0x_0 e^{x_1})$ | $2.0x_0\log(x_1^2)$ | $x_0\sin\left(\frac{1}{x_1}\right) + 1$ | $\sqrt{x_0}\log(x_1^2)$ |
| TaylorGP | $0.26x_1 + \log\left(\frac{0.952}{|x_0|}\right) - \sqrt{\left|\log\left(\frac{0.593}{|x_0|}\right) + \log\left(\left|\tanh\left(\frac{x_1}{x_0}\right)\right|\right)\right|}$ | $\sin(x_0 e^{x_1})$ | $4x_0\log(|x_1|)$ | $\frac{x_0\tanh(x_1^2)}{x_1} + 1$ | $2\log(|x_1|)\sqrt{|x_0|}$ |
| NeSymReS | $\log\left(\frac{0.274|x_1|}{|x_0|}\right)$ | $\sin(x_0 e^{x_1})$ | $x_0\log(x_1^4)$ | $(x_0 + x_1)\sin\left(\frac{1}{x_1}\right)$ | $0.23x_0 + \log(x_1^2)$ |
| E2E | $2.0 - 0.6\log(6.862(1 - \frac{0.129}{-0.056x_1 - 0.02})^2(x_0 - 0.009)^2 + 0.8)$ | $-0.998\sin((0.214 - 33.248x_0)(0.026\,e^{1.125x_1} + 0.001))$ | $0.241x_0(31.0\log(0.19(|25.419x_1 + 0.12|+0.011)^{0.5} - 0.01) + 0.5)$ | $(0.629 - 4.58\sin((-0.007 + \frac{9.78}{0.014 - 3.043x_1})(0.042x_0 + 0.001)))(1.581 - 0.006x_0)$ | $(-0.052 + \frac{0.004}{-0.019 - \frac{4.486}{5.98x_0 - 68.5}})(48.802\log(0.012|0.001 - 6.156/((0.243x_1 - 0.002)(0.806\sin(0.085x_0 - 0.168) + 0.054))| + 0.025) + 0.04)$ |
| uDSR | $\log\left(\frac{x_1 + 0.5}{2.0x_0^2 + 0.5}\right)$ | $\sin(x_0 e^{x_1})$ | $x_0\log(1.0x_1^4)$ | $\left(1.0x_0^2\sin(1/x_1) + x_0\right)/x_0$ | $(x_0^2\log(x_1^2))(x_0 + (0.089x_0^4 + 0.223x_1 + 2.369x_0^3 + 0.007x_0^2 x_1 - 11.117x_0^2 - 0.077x_0 x_1 + 25.02x_0 - 0.001x_1^4 + 0.002x_1^2 - 15.166)(x_0^2 + x_0))$ |
| TPSR | $0.612x_1 + (0.995 - 1.013\arctan(x_1 + 0.926))(0.808x_1 - 1.554) - 1.0\log(51.139x_0^2 + 12.793) + 2.955$ | $0.9998\sin(1.0x_0 e^{x_1})$ | $-0.03x_0(0.196x_1 - 261.804 + 40.943/(0.07|1.488x_1 + 0.003|+0.058))$ | $-0.772x_0 + (10.323 + \frac{1}{x_0 - 0.027})(0.029\arctan(15.95x_1 - 5.238) + 0.059)(x_0 + (0.128 - 0.034x_1)(1.088 - 0.098x_1)(4.661x_0 - 0.086) - 0.027) + 0.946$ | $\frac{(-1.365 + \frac{1.705}{0.0007x_0 - 0.55113(x_1 + 0.006)^2 + 0.376})}{(-0.457x_0 - 2.791)}$ |
| SeTGAP | $-1.0\log(2.785x_0^2 + 0.696) + 1.0\,|\log(13.845x_1 + 6.92)| - 2.296$ | $1.0\sin(1.0x_0 e^{x_1})$ | $4.0x_0\log(|x_1|)$ | $1.0x_0\sin\left(\frac{1}{x_1}\right) + 1.0$ | $2.0\sqrt{x_0}\log(|x_1|)$ |

Table 31: Comparison of predicted equations (E1—E4) with rounded numerical coefficients — Iteration 8

| Method | E1 | E2 | E3 | E4 |
|---|---|---|---|---|
| PySR | $0.609x_0 x_1 + 0.931\cos(1.488x_0 - 1.126x_1(2x_0 - 3.019) - 1.9x_1 + 0.556)$ | $\frac{2.278\tanh(0.356x_2) + 6.388}{\sqrt{(0.016x_0^2 + e^{-0.402x_0})^{0.5} + 0.127\tanh(x_1 x_2)}}$ | $0.15e^{1.499x_0} - 0.502\cos(2.999x_1 + 3.143)$ | $(\cosh(x_0) + \cosh(x_2))(-0.005x_1 - 0.005x_3 + 0.091)$ |
| TaylorGP | $0.613x_0 x_1$ | $-0.506x_0 - 0.001x_1 + 0.206x_2 + 8.575$ | $-0.612\left(e^{x_0}e^{2\tanh(x_0)}\right)^{0.5} + e^{x_0}$ | $\left(-\sin(x_0) + \sqrt{|x_2|}\right)e^{\tanh\left(\log\left(\sqrt{|x_0||x_2|}\right)\right)}$ |
| NeSymReS | $0.666x_0 x_1 + \cos(0.068(-x_0 - 0.83x_1)^2)$ | $e^{\sin(x_0 x_1)} + 124.329 - 116.894e^{-0.002x_2}$ | $0.579e^{x_0} - 0.444\cos(0.228x_1)$ | — |
| E2E | $(0.001x_0 - 1.463)\sin(0.482\cos(\frac{0.121x_0 + 78.526}{0.007x_0 + 0.29}) - 0.004) + (3.222x_0 + 0.007)(0.186x_1 + 0.003)$ | $(0.01x_0 - 0.099)(5.434x_0 + 0.438) + 6.3 + 3.0\sin((0.001x_1 + 0.02)(10.39x_2 + 0.291))$ | $0.143e^{1.497x_0} + 0.043 + 0.506\cos(3.559x_1 + 0.061)$ | $0.079|0.162(-x_1 + 0.885(x_0 + 0.05)^2 + 0.031)^2 + 0.123(0.135x_0 x_1 + 0.004x_0 + 0.011x_1 - x_2^2 - 0.022x_2 + 0.884x_3 + 0.024)^2 + 0.523| - 0.048$ |
| uDSR | $(0.001x_0^2 x_1 + 0.003x_0^2 + 0.604x_0 x_1 - 0.003x_0 + 0.001x_1^3 - 0.007x_1^2 - 0.021x_1 + 0.013)\cos\left(e^{-3x_0^2}\right)$ | $(0.062x_0^2 - 0.5x_0 + 0.001x_1^2 + 0.029x_1 x_2 - 0.001x_1 - 0.002x_2^2 + 0.539x_2 + 6.5)\cos\left(\frac{x_1}{e^4}\right)$ | $0.075x_0^4 + 0.236x_0^3 + 0.022x_0^2 - 0.001x_0 x_1^2 + 0.003x_0 x_1 - 0.104x_0 + 0.014x_1^4 - 0.001x_1^3 - 0.23x_1^2 + 0.003x_1 + \cos(x_1)\cos(2x_1) - \cos(x_1) + 0.712$ | $0.01x_0^4 - 0.02x_0^2 x_1 + 0.01x_1^2 + 0.01x_2^2 - 0.02x_2^2 x_3 + 0.01x_3^2$ |
| TPSR | $-2.937x_1(0.001 - 0.208x_0) + 0.04\sin(4.355x_1 - 0.296) - 0.002$ | $(-20.407 - \frac{0.002}{x_1 - 0.182})(-0.01x_2 - 1.038 + 1/(0.014(0.201x_0 + 0.007x_2 - 1)^2(0.442x_0 + 0.001x_2 - 1)^2 + 1.342))$ | $0.15e^{1.5x_0} + 0.5\cos(3.0x_1)$ | $-0.00034(0.347x_1 - 7.955)(x_3 - (90.12 - 0.35x_3)(0.111x_1 - 2.6) - 1.987) + 0.01(-0.001x_1 + 0.995x_2^2 - x_3 + 0.062)^2 - 0.621$ |
| SeTGAP | $0.608x_0 x_1 + 1.099\sin(2.251x_0 - 1.52)(x_1 - 0.655))$ | $\frac{-0.5x_0 + 0.062x_0^2 + 6.541 + }{\sqrt{0.1x_1 + 1}}(3.166\sin(0.201x_2) - 0.033)$ | $0.15e^{1.5x_0} + 0.5\cos(3.001x_1)$ | $-0.005x_0^2 + 0.01x_0^4 - 0.02x_1(-x_0)^2 + 0.01x_1^2 + 0.01x_2^4 - 0.02x_3(-x_2)^2 + 0.01x_3^2 + 0.015$ |

Table 32: Comparison of predicted equations (E5—E8) with rounded numerical coefficients — Iteration 8

| Method | E5 | E6 | E7 | E8 |
|---|---|---|---|---|
| PySR | $e^{1.2x_3} + \sin(x_0 + x_1 x_2)$ | $0.279e^{3.915\cos\left(\frac{\tanh(1.761\tanh(\sin(x_1)))}{x_1}\right)} \cdots \cos\left(\tanh\left(\frac{4.667}{x_1}\right)\right)\cos(0.2x_2^2) + \tanh(x_0)$ | $\cos(x_1) + \sinh(\sin(6.285x_0) + \cos(0.35x_1) + \cos(0.668x_1) - 2.288)$ | $\log(0.705\sinh(1.975\cosh(\tanh(x_0)) - \frac{1.184}{\sinh(\cosh(x_1))}))$ |
| TaylorGP | $0.855e^{x_3}\log(|2x_3 + \sqrt{|\tanh(e^{2x_3})|} + 0.748|)$ | $\cos(x_2) + 0.422 + \frac{5.65\log(|x_1|^{0.5})\sin(x_2)}{x_2}$ | $-x_1^2 + \log(|x_1^2 - \log(|x_1^2 - \log(|x_1^2 + 0.943|)|)|)$ | $\frac{0.829|x_0|^{\frac{1}{4}}}{\cos(\tanh(x_1))}$ |
| NeSymReS | — | $0.126x_0 + x_1\sin\left(0.294x_1 + \frac{x_2}{x_1}\right)$ | $\frac{0.042x_0 + x_1^2}{\cos(5.419(x_1 - 0.517)^2) - 1.896}$ | $\cos\left(\frac{\sin\left(\frac{x_0}{x_1}\right)}{x_0}\right) + 0.644$ |
| E2E | $1.392e^{1.085x_3} + 0.949\sin(-25.875x_2 + 100.707 + \frac{0.007}{0.055 - 0.1x_1}) - 0.484$ | $(6.048|0.168x_1 + 0.001|+0.14)\cos((0.21 - 17.355x_2)(-0.012x_2 - 0.01)) + 0.8\arctan(0.699x_0 + 0.269) - 0.01$ | $(-10.369(1 - 0.002x_0)^2(0.346x_1 + (0.01 - 0.003x_1)(0.029x_0 - 0.003) + 0.018)^2 + 1.7)(0.373(\sin(6.398x_0 + 0.342) - 0.814)^2 + 0.42)$ | $2.065(-1 - 0.556/(-0.361(x_0 + 0.028)^2(0.001x_1 - 1)^2(x_1 + 0.046)^2 + 0.409\cos(2.271x_1 + 0.076) - 1.369))^2 + 0.001$ |
| uDSR | $0.001x_0 x_1 - 0.001x_0 x_3 - 0.002x_0 + 0.001x_1^2 + 0.001x_1 x_2 + 0.001x_1 x_3^2 + 0.003x_1 x_3 - 0.002x_2^2 - 0.001x_2 x_3 - 0.004x_2 + 0.153x_3^4 + 0.569x_3^3 + 0.506x_3^2 + 0.602x_3 + \sin(x_0 + x_1 x_2) + 1.088$ | $-0.002x_0^3 - 0.002x_0^2 + 0.007x_0 x_1 + 0.004x_0 x_2 + 0.262x_0 + 0.039x_1^2 + 0.002x_1 x_2 + 0.006x_1 + 0.003x_2^4 - 0.262x_2^2 - 0.001x_2 + 3\cos(x_2) - \cos(2x_2) + 2.697$ | $0.001x_0^4 - 0.037x_0^3 - 0.001x_0^2 x_1^2 - 0.002x_0^2 x_1 - 0.018x_0^2 - 0.001x_0 x_1^3 - 0.013x_0 x_1^2 + 0.011x_0 x_1 - x_0\sin(x_0^2) + 0.501x_0 + 0.001x_1^3 - 0.896x_1^2 + 0.032x_1 + 0.976$ | $-0.004x_0^4 + 0.107x_0^2 - 0.002x_0 + 0.008x_1^4 - 0.179x_1^2 - 0.001x_1 - e^{\cos(x_0)} + \cos(x_0) - \cos(x_1) + 2.723$ |
| TPSR | $0.983e^{-0.001x_2 + 1.206x_3} e^{0.001\arctan((-59.047x_0} \cdots {}^{\cdots - 117.779)(-70.903x_1 -} \cdots {}^{\cdots 32.111x_3 + 8.033))} + 0.01$ | $(0.003\cos(0.044x_0 + 0.023x_1^2 - 2.266) + 0.003)(5.010^{-6}x_0 - 0.005x_1 - 465.626(0.025 - \arctan(-0.473 - \frac{9.002}{-0.668(x_2 - 0.082)^2 - 0.109}))^3 - 0.002)$ | $1.045 - 0.926(0.012x_0 - x_1 + 0.031)^2$ | $2.007 - \frac{1.09}{0.179(0.01 - x_1)^2(-x_0 - 0.03)^2 + 0.823}$ |
| SeTGAP | $0.999e^{1.2x_3} + 1.0\sin(x_0 + x_1 x_2 + 12.567) + 0.003$ | $1.0\cos(0.2x_2^2 + 12.564)|x_1| + 1.0\tanh(0.501x_0)$ | $\frac{3.767 - 3.787x_1^2}{(3.786\sin(6.283x_0) + 5.68)} + 0.003$ | $\frac{14.946}{0.121x_0^3 - 15.228x_0^4 + 0.176(-x_0)^2 - 14.95} - \frac{18.513}{0.135x_1 - 0.322x_1^3 + 18.515(-x_1)^4 + 18.515} + 2.0$ |

Table 33: Comparison of predicted equations (E9—E13) with rounded numerical coefficients — Iteration 8

| Method | E9 | E10 | E11 | E12 | E13 |
|---|---|---|---|---|---|
| PySR | $0.843e^{1.412\cos{(x_0)}} + \log{(x_1 + 0.504)}$ $-3.313 + 0.843\cos{(\cos{(0.866x_0)})}$ $-0.036\cosh{(x_0)}$ | $\sin{(x_0 e^{x_1})}$ | $2x_0\log{(x_1^2)}$ | $x_0\sin(\cos(0.001/$ $((1581.494x_1)/(\cosh(x_1))$ $+\cosh(x_1)))/x_1) + 1$ | $\sqrt{x_0}\log{(x_1^2)}$ |
| TaylorGP | $\log{(\frac{0.764}{|x_0|})} + \tanh{(x_1)} - (|\log{(\frac{0.792}{|x_0|}}$ $+\log{(|\tanh{(\frac{x_1}{x_0})}|)}|)|^{0.5}$ | $\sin{(x_0 e^{x_1})}$ | $4x_0\log{(|x_1|)}$ | $0.938 + \frac{-x_0 x_1 - 0.344}{-x_1^2 - 0.307}$ | $0.255x_0\log{(|x_1|)} + 2.807\log(0.943$ $|x_1|)|\log(0.826\sqrt{|x_0|}|\log(4.902$ $|x_0|(4.902|\log(|x_1|)|)|)|) + 0.13|^{0.5}$ |
| NeSymReS | $\log{(\frac{0.282|x_1|}{|x_0|})}$ | $\sin{(x_0 e^{x_1})}$ | $x_0\log{(x_1^4)}$ | $(x_0 + x_1)\sin{(\frac{1}{x_1})}$ | $0.239x_0 + \log{(x_1^2)}$ |
| E2E | $-6.44|0.81\arctan{(0.295x_0 + 0.004)}$ $+0.001|+3.33 + 96.1/((0.021x_0-$ $47.199)(0.062x_1 + 0.262)(0.023x_0$ $+0.692x_1 + 2.901))$ | $0.994\sin((-5.26x_0-$ $0.008)(0.273e^{0.573x_1}+$ $0.006)(-0.31x_1+$ $0.118\log(0.055x_1+$ $0.756) - 0.782))$ | $(0.331x_0 + 0.001)$ $(20.8\log(0.327(|20.18$ $x_1 + 0.198|+0.009)^{0.5}$ $-0.003) - 4.11)$ | $7.72\sin(0.003x_0(0.008+$ $\frac{6.35}{0.173x_1 + 0.002})) + 0.078$ $|0.006x_1 - 0.247|+0.987$ | $(0.034 - 0.052\sqrt{|0.603x_0 + 1|})$ $(0.08 - 57.9\log(|7.09|0.162x_1$ $-0.004|-0.001|))$ |
| uDSR | $\log{(\frac{2x_1 + 1}{4.0x_0^2 + 1.0})}$ | $\sin{(x_0 e^{x_1})}$ | $x_0\log{(1.0x_1^4)}$ | $(1.01x_0 + 0.008x_1^2 + 1.017x_1 - 0.161)\sin{(1/x_1)}$ | $\log{(x_1^2)}\log(0.004x_0^3 + 0.061x_0^2$ $+1.186x_0 - 0.001x_1 + 1.427)$ |
| TPSR | $0.13x_1 + 2.261\arctan{(x_1 + 0.751)}$ $-1.0\log(13.419x_0^2 + 3.359) - 0.1994$ | $1.0\sin{(1.0x_0 e^{x_1})}$ | $-17.697x_0 + (0.185x_0+$ $0.002)(0.673(0.001 - x_1)^2$ $+122.071+$ $\frac{112.993}{-0.116|29.974x_1 + 0.112|-1.036})$ $-0.157$ | $(7.035\cdot 10^{-9}x_0 + 8.736\cdot 10^{-5})$ $(5.096x_1 + 11119.99+$ $\frac{(0.002 - 0.068x_0)(288275.2x_1 + 22166.6)}{-1.46(x_1 + 0.084)^2 - 0.985})$ | $(0.089 - \frac{0.153}{0.125|x_1 + 0.01|})(-15.537x_0$ $-72.042)(0.013(0.004 - x_1)^2$ $(1 - 0.005x_0)^2 - 0.01)$ |
| SeTGAP | $-0.999\log{(3.882x_0^2 + 0.963)}+$ $0.999|\log(10.173x_1 + 5.052)| - 1.656$ | $1.0\sin{(1.0x_0 e^{x_1})}$ | $2.0x_0\log{(x_1^2)}$ | $1.0x_0\sin{(\frac{1}{x_1})} + 1.0$ | $1.0\sqrt{x_0}\log{(x_1^2)}$ |

Table 34: Comparison of predicted equations (E1—E4) with rounded numerical coefficients — Iteration 9

| Method | E1 | E2 | E3 | E4 |
|---|---|---|---|---|
| PySR | $0.607x_0 x_1-$ $1.1\cos(x_0(2.25x_1-$ $1.5) - 1.5x_1 + 2.571)$ | $0.178x_1\sin{(0.199x_2)} + 2.983\sin{(0.2x_2)}+$ $2.983\log{(\cosh{(0.464x_0 - 1.809)} + 5.42)}$ | $0.15e^{1.5x_0} + 0.5\cos{(3x_1)}$ | $(0.092 - 0.008x_1)(\cosh{(x_0)} + 6.478)+$ $(0.092 - 0.009x_3)\cosh{(x_2)} - 0.643$ |
| TaylorGP | $0.613x_0 x_1$ | $\tanh{(x_2)} + |(x_0(-x_0 + (e^{x_0})^{\frac{1}{4}} + \log{(55.556|x_0|)})$ $+1.704 + \cos{(e^{1.424\log{(55.556|x_0|)^{\frac{1}{4}}}})}(\tanh{(x_0)})|^{0.5}$ $+1.305|-0.587x_0 + 0.587(e^{x_0})^{\frac{1}{4}} + 0.587$ $\log{(55.556|x_0|)} + 0.587\cos{(x_0)} + 1|^{0.5}$ | $-x_0 + e^{x_0} + \cos{(x_0)} - 0.815$ | $x_0 + 0.118x_2^2 - 0.412$ |
| NeSymReS | $0.61x_0 x_1 + \sin($ $0.002(-x_0 - 0.698)^2)$ | $93.728e^{0.002x_2} + e^{\sin{(x_0 x_1)}} - 86.434$ | $0.583e^{x_0} - 0.465\cos{(0.204x_1)}$ | — |
| E2E | $0.021x_1(29.356x_0+$ $0.027) + 1.1\cos(30.874$ $x_1 - 1.378) - 0.001$ | $0.019x_1 + (0.007x_0 - 0.056)(8.915x_0 + 0.563)+$ $(0.437x_1 + 8.091)$ $(0.371\sin{(0.194x_2 + 0.034)} - 0.007) + 6.279$ | $0.042x_0 - 0.004x_1 + (0.001x_0$ $+0.216)(2.33\cos(3.43x_1+$ $0.075) - 0.07) + 0.166$ $\sqrt{0.779e^{2.954x_0} + 1} - 0.008$ | $0.004|2.358(x_3 - 0.048)^2 + 2.231((x_0+$ $0.028)^2 + 0.039)^2 + 2.177(-0.921x_1+$ $(x_2 + 0.01)^2 + 0.024)^2 - 0.255|+0.003$ |
| uDSR | $(-0.002x_0^4 + 0.001x_0^3 x_1 + 0.001x_0^3-$ $1.265x_0^2 x_1 + 0.017x_0^2 + 0.001x_0 x_1^3+$ $0.001x_0 x_1^2 + 0.545x_0 x_1 - 0.011x_0+$ $0.001x_1^4 + 0.003x_1^3 - 0.027x_1^2-$ $0.05x_1 + 0.04)\sin{(\frac{x_0^2}{-2x_0^2 + x_0})}/x_0$ | $\log{(660.383e^{0.063x_0^2 - 0.001x_0 x_1}}$ $e^{0.001x_0 x_2 - 0.504x_0 + 0.03x_1 x_2}$ $e^{-0.005x_1 - 0.002x_2^3 + 0.543x_2} + 1)$ | $0.013x_0^4 - 0.033x_0^3 - 0.001x_0^2 x_1$ $-0.414x_0^2 - 0.899x_0 - 0.13x_1^4+$ $0.001x_1^3 + 0.979x_1^2+$ $0.001x_1 + e^{x_0} + e^{\cos{(2x_1)}} - 2.821$ | $0.01x_0^4 - 0.02x_0^2 x_1 + 0.01x_1^2+$ $0.01x_2^2 - 0.02x_2^2 x_3 + 0.01x_3^2$ |
| TPSR | $0.072x_1(8.488x_0+$ $0.065) + 0.056\sin(6.353$ $x_1 + 1.19) - 0.032$ | $(-0.009 + \frac{1}{-89.564(0.13x_2 - 1)^2 - 79.916})$ $(-1126.481((-0.001+$ $\frac{0.002}{0.283(0.259x_2 + 0.033)(9.015x_2 - 0.006)|+0.093})$ $(0.441x_2 + 0.076)(-0.947x_0 - 0.688x_1$ $-0.205x_2 + 0.213) + 1)^3 - 7.999)$ $(-0.002x_1 + 0.122(0.134x_0 - 1)^2 + 0.303)$ | $0.007x_1(82.733x_0 + 27.9)+$ $0.351(0.166x_1 - 1)^3(-0.361$ $\cos((0.014 - 0.032x_1)(33.244$ $-77.329x_0)) - 1)^3 - 0.553$ | $-0.1167x_1 - 0.149x_3 + 0.0016(x_2-$ $0.123)^2 + 0.00512(-x_2 - (-5.082x_0$ $-0.035)(-0.255x_0 - 0.072)$ $-0.002)^2 + 1.551$ |
| SeTGAP | $0.608x_0 x_1 + 1.1$ $\sin((2.256x_0 - 1.491)$ $(x_1 - 0.655) + 0.047)$ $-0.001$ | $-0.499x_0 + 0.064x_0^2+$ $\sqrt{0.1x_1 + 1}(3.161\sin{(0.2x_2)} + 0.011) + 6.46$ | $0.15e^{1.5x_0} + 0.5\cos{(3.0x_1)}$ | $-0.02x_0^2 x_1 + 0.004x_0^4 + 0.01x_0^4 + 0.01x_1^2-$ $0.02x_2^2 x_3 + 0.01x_2^4 + 0.01x_3^2 - 0.009$ |

Table 35: Comparison of predicted equations (E5—E8) with rounded numerical coefficients — Iteration 9

| Method | E5 | E6 | E7 | E8 |
|---|---|---|---|---|
| PySR | $(e^{1.2x_3})+$ $\sin{(x_0 + x_1 x_2)}$ | $\tanh{(x_0)} + 9.715\cos{(0.2x_2^2)}$ $\cos{(\frac{5.902}{\tanh{(\cosh{(0.095x_1)})}})}$ | $(2.482x_1^2 - 2.467)$ $(\cos{(0.435e^{\cos{(6.278x_0 + 1.57)}})} - 1.16)$ | $0.8\tan(\sinh(\tanh(\cosh(0.827$ $\cosh(\tan(\tanh(x_1))))\tanh(\cosh(x_0)))))$ $\cosh{(\tanh{(x_0)})} - 0.957$ |
| TaylorGP | $(\cos(\sin(\tanh(1.515e^{x_3})))$ $+\tanh(1.119|(0.799\cos(\sin($ $\tanh(1.515e^{x_3}))) + 0.799\tanh($ $0.534e^{0.5(x_3)}))e^{x_3} - e^{x_3}|^{0.5}))e^{x_3}$ | $\cos{(x_2)} + \tanh{(x_0)}+$ $\frac{7.407\log{(|x_1|^{0.5})}\sin{(x_2)}}{x_2}$ | $-x_1^2 + \log(|x_1^2+$ $\frac{-x_1^2 + \sqrt{|x_1|} + \sqrt{|x_1 - 0.479\tanh{(e^{-0.538x_1^2})}|}}{x_1^2 + 0.392}|)$ | $e^{\tanh\left(\tanh\left(\log\left(0.864\sqrt{|x_0||x_1|}\right)\right)\right)}$ |
| NeSymReS | — | $0.145x_0 + x_1\sin{(0.302x_1 + \frac{x_2}{x_1})}$ | $-0.05632x_0 - 0.8403x_1^2$ | $\cos{(\frac{\sin{(\frac{x_0}{x_1})}}{x_0})} + 0.634$ |
| E2E | $-0.001x_1 + 0.002x_3 + 0.023$ $(0.024 - x_3)^2 + 0.947e^{1.206x_3}+$ $0.933\cos(12297.635(0.404x_1$ $+1)^2 + 0.225(0.001x_0 - 0.001$ $x_2 + 1)^{0.5} - 15.41) + 0.032$ | $(0.035x_1 - 0.002)(4.984\arctan($ $(-0.03 - \frac{4.462}{-0.539x_0 - 0.118})(0.004$ $x_0 + 0.001)) - 0.16) + 0.929$ | $\left(0.09 - 0.082(x_1 + 0.007)^2\right)$ $(6.131(0.766 - \sin(6.516$ $x_0 + 0.216))^2 + 5.0)$ | $2.1-$ $\frac{0.9}{0.002|(3.087x_1 - 0.168)(37.54(x_0 + 0.029)^3 + 6.0)|+0.6}$ |
| uDSR | $0.001x_0 x_1 + 0.001x_0 - 0.002x_1^2-$ $0.002x_1 x_2 - 0.001x_1 x_3^2 - 0.004x_1 x_3+$ $0.007x_1 + 0.003x_2^2 - 0.001x_2 x_3+$ $0.152x_3^4 + 0.569x_3^3 + 0.504x_3^2+$ $0.601x_3 + \sin{(x_0 + x_1 x_2)} + 1.088$ | $-0.002x_0^3 + 0.003x_0^2 - 0.003x_0 x_1+$ $0.002x_0 x_2 + 0.238x_0 + 0.038x_1^2+$ $0.002x_1 x_2 + 0.029x_1 + 0.003x_2^4-$ $0.264x_2^2 + 0.022x_2 + 3\cos{(x_2)}$ $-\cos{(2x_2)} + 2.717$ | $0.001x_0^4 - 0.037x_0^3 + 0.001x_0^2 x_1^2+$ $0.002x_0^2 x_1 - 0.019x_0^2 + 0.001x_0 x_1^3-$ $0.009x_0 x_1^2 - 0.009x_0 x_1 - x_0\sin{(x_0^2)}+$ $0.485x_0 + 0.004x_1^3 - 0.897x_1^2$ $-0.04x_1 + 0.951$ | $-0.004x_0^4 + 0.107x_0^2 + 0.008x_1^4$ $-0.18x_1^2 - 0.002x_1 - e^{\cos{(x_0)}}$ $+\cos{(x_0)} - \cos{(x_1)} + 2.725$ |
| TPSR | $2.627214 - 0.004x_2$ | $(0.397\cos(0.017x_1 + 0.524x_2+$ $0.076) + 0.24)(0.234x_0 + \cos($ $1.716x_1 - 2.174) + 6.18)$ | $-0.934(0.086 - x_1)^2(0.001x_0$ $+1 + \frac{0.116}{x_1})^2 + 1.098$ | $0.001x_1 + 0.61(0.002 - \arctan($ $-0.01x_0(89.997x_1 - 2.242))^2 + 0.654$ |
| SeTGAP | $1.001e^{1.2x_3} + 1.0\sin($ $x_0 + x_1 x_2 + 12.566) - 0.008$ | $-1.0\cos{(0.2x_2^2 - 9.425)}|x_1|+$ $1.0\tanh{(0.5x_0)}$ | $\frac{8.18x_1^2 - 8.247}{(-8.185\sin{(6.283x_0 - 6.283)} - 12.273)}$ $-0.006$ | $\frac{16.205}{(-0.015x_0 - 16.207x_0^4 - 0.033(-x_0)^3 - 16.206)} + 2.0+$ $\frac{11.914}{(0.013x_1 - 0.118x_1^2 - 0.016x_1^3 - 11.817x_1^4 - 11.901)}$ |

Table 36: Comparison of predicted equations (E9—E13) with rounded numerical coefficients — Iteration 9

| Method | E9 | E10 | E11 | E12 | E13 |
|---|---|---|---|---|---|
| PySR | $\log\left(\frac{x_1+0.5}{2x_2+0.5}\right)$ | $\sin(x_0 e^{x_1})$ | $2.0x_0\log(x_1^2)$ | $x_0\sin\left(\frac{1}{x_1}\right)+1.0$ | $\sqrt{x_0}\log(x_1^2)$ |
| TaylorGP | $-\log(|x_0|)+\tanh(x_1)-0.298-\sqrt{\left|\log\left(\frac{0.873}{|x_0|}\right)+\tanh(x_1)-\sqrt{|x_0|}\right|}$ | $\sin(\tanh(x_0))\cos(\sqrt{|x_1|})$ | $4x_0\log(|x_1|)$ | $\dfrac{0.998x_0\sin\left(\frac{x_0}{x_1(x_0+0.048)}\right)}{\sqrt{|\log(\sqrt{|x_0|})|}}+0.998$ | $(\sqrt{x_0}\log(\sqrt{|x_1|})^2+0.689)(\log(\sqrt{|x_1|})+0.446)+\log(|x_1|)(\log(|x_0|)+1)+\log(|x_1\tanh(x_1)|)$ |
| NeSymReS | $\log\left(\frac{0.281|x_1|}{|x_0|}\right)$ | $\sin(x_0 e^{x_1})$ | $x_0\log(x_1^4)$ | $(x_0+x_1)\sin\left(\frac{1}{x_1}\right)$ | $0.243x_0+\log(x_1^2)$ |
| E2E | $-0.6\log(10.053(0.009-x_0)^2+0.6)+1.9-\frac{0.7}{0.274x_1+0.301}$ | $-0.994\sin(3.608x_0+0.013)(-0.009-0.234e^{(0.595-0.053x_1)}...(1.976x_1+0.061)))-0.001$ | $(2.922x_0+0.006)(4.14\arctan(0.336-0.008|0.26x_1+0.695-\frac{84.4}{3.242x_1-0.071}|)-0.047)$ | $0.588+7.1\sin(0.058-9.51/((0.01-\frac{103.409}{13.96x_0-0.071})(-0.001x_0+10.994x_1-0.059)))$ | $(0.741-4.65\arctan(0.175 x_0+0.35))(0.174e^{0.175x_0}-20.8)(0.008\log(0.001x_1+12.9)+0.072)$ |
| uDSR | $\log((0.002x_0^4+0.002x_0^3x_1+0.001x_0^3+0.046x_0^2x_1-0.016x_0^2+0.002x_0x_1^2-0.009x_0x_1^2+0.213x_0x_1+0.101x_0-0.001x_1^4+0.01x_1^3-0.035x_1^2+2.146x_1+1.143)(4x_0^2+e^{x_0}))$ | $\sin(x_0 e^{x_1})$ | $x_0\log(1.0x_1^4)$ | $x_0\sin(1/x_1)+1.0$ | $(\log(x_1^2))(x_0+x_1+(0.003x_0^3-1.047x_0^2-0.999x_0x_1+0.54x_0+0.455)/x_0)$ |
| TPSR | $59.31(-0.003x_1-1)^2-66.389-82.366/(-4.5|(31.371+\frac{17.165}{0.218x_1+0.124})(0.301x_0-0.001)(0.007x_1+0.021)|-10.235$ | $1.0\sin(0.9992x_0 e^{x_1})$ | $7.953x_0-0.188$ | $0.827+5.558\sin((0.056x_0-4.862)(0.006x_0(0.1+\frac{8.807}{0.002-1.613x_1})-0.009))$ | $-0.684x_0+97.57+\frac{50.936}{-0.009x_0-0.535}+15.513/((66.734+207.296e^{-0.219x_0})(-0.012(x_1-0.002)^2-0.007))$ |
| SeTGAP | $-1.001\log(13.041x_0^2+3.264)+0.998|\log(7.365x_1+3.647)|-0.108$ | $1.0\sin(1.0x_0 e^{x_1})$ | $4.0x_0\log(|x_1|)$ | $1.0x_0\sin\left(\frac{1}{x_1}\right)+1.0$ | $2.0\sqrt{x_0}\log(|x_1|)$ |

Table 37: Comparison of predicted equations (E1—E4) with rounded numerical coefficients — Iteration 10

| Method | E1 | E2 | E3 | E4 |
|---|---|---|---|---|
| PySR | $0.608x_0x_1$ | $x_0+2.969\sin(0.2x_2)+\cosh(0.083x_0-3.519)-10.302$ | $0.15e^{1.5x_0}+0.5\sin(3.0x_1+1.571)$ | $-0.165x_1-0.165x_3+1.564\log((\cosh(\sinh(0.083x_0^2))-0.553)\cosh(x_2))+1.564\cos(x_2)$ |
| TaylorGP | $\frac{0.604x_0x_1(x_0^2x_1^2)^{\frac{1}{4}}}{\sqrt{|x_0x_1|}}$ | $-0.513x_0-0.001x_1+0.204x_2+8.572$ | $-x_0\tanh(0.632e^{x_0}-1)+e^{x_0}-\log(e^{x_0})-0.682$ | $x_0+x_2-2$ |
| NeSymReS | $0.589x_0x_1+\cos(0.598(0.966x_0-x_1)^2)$ | $-x_0+e^{\sin(x_1)}+63.876-56.599e^{-0.004x_1}$ | $0.575e^{x_0}-0.418\sin(1.451(0.052x_1+1)^2)$ | — |
| E2E | $(23.064x_0-0.034)(0.026x_1+0.001)-0.001+1.09\sin(0.158x_0+(1.872x_1+0.125)(0.003|0.021x_1-2.239|+81.7)+0.021)$ | $-0.008x_0(0.172x_1-1.607)(0.007x_1-40.299)+6.26+(0.008x_1+0.149)(19.2\sin((0.001x_2+0.073)(3.077x_2+0.001))+0.003)$ | $0.003x_0+0.06e^{1.804x_0}-0.499\cos(0.003x_0-2.985x_1+59.557)+0.159$ | $0.843|0.07x_3-0.013(0.836x_1-(x_0+0.059)^2(0.004x_1-1)^2+0.039)^2-0.01(0.96x_3-(x_2-0.046)^2+0.046)^2+0.017|$ |
| uDSR | $(-0.003x_0^4-0.002x_0^3x_1+0.002x_0^3-0.004x_0^2x_1^2+0.002x_0^2x_1+0.084x_0^2-0.002x_0x_1^3+2.43x_0x_1^2-1.168x_0x_1-0.003x_1^4-0.003x_1^3+0.069x_1^2+0.022x_1-0.329)\sin\left(\frac{x_1}{4x_1^2-2x_1}\right)$ | $\log(658.504e^{0.062x_0^2-0.001x_0x_1}e^{0.001x_0x_2-0.502x_0+0.03x_1x_2}e^{-0.002x_1-0.002x_2^2+0.542x_2}+\cos(x_1/x_2))$ | $0.013x_0^4-0.033x_0^3-0.412x_0^2+0.001x_0x_1-0.898x_0-0.129x_1^4-0.001x_1^3+0.973x_1^2+0.004x_1+e^{x_0}+e^{\cos(2x_1)}-2.814$ | $0.01x_0^4-0.02x_0^2x_1+0.01x_1^2+0.01x_2^4-0.02x_2^2x_3+0.01x_3^2$ |
| TPSR | $-x_0+(1.009-73.098x_0)(-0.008x_1-0.013)+0.007$ | $(0.098+\frac{1}{x_0+44.83})(0.217-0.991x_2)(-0.006x_0-(-0.303+\frac{23.261}{x_2-0.219})(0.033x_1+38.009)(-0.003x_2-0.213)(-0.613x_2+(0.499-0.007x_0)(27.627(-0.01x_0-1)^2+7.021)-17.446)+111.696+\frac{0.001}{x_1})$ | $-0.008x_0(5.0-75.398x_1)+0.04x_0+1.1\cos(8.906(0.168-0.253x_1)(x_0-0.667)+1.567)$ | $-0.182x_1+0.005(1-0.015(0.007-|9.838x_0+0.02|)^2)^2+0.04(0.011x_0^2+0.45x_2^2-0.425x_3+1)^2+0.045$ |
| SeTGAP | $0.607x_0x_1+1.099\sin(2.242x_0-1.492)(x_1-0.685))$ | $-0.5x_0+0.062x_0^2+3.162\sqrt{0.1x_1+1}\sin(0.201x_2)+6.501$ | $0.147e^{1.507x_0}-0.5\sin(3.001x_1+4.712)$ | $-0.02x_0^2x_1+0.01x_0^4+0.01x_1^2+0.01x_2^4+0.01x_3^2-(0.02x_2^2+0.002)(x_3+0.212)+0.009$ |

Table 38: Comparison of predicted equations (E5—E8) with rounded numerical coefficients — Iteration 10

| Method | E5 | E6 | E7 | E8 |
|---|---|---|---|---|
| PySR | $1.0e^{1.2x_3}+\sin(x_0+x_1x_2)$ | $|x_1|\cos(0.2x_2^2)+\tanh(0.5x_0)$ | $\frac{2.119-2.118x_1^2}{2.118\sin(6.283x_0)+3.176}$ | $\log(\tan(1.442(\tanh(\cosh(x_1)\tanh(\sqrt{\cosh(x_0)}))\tanh(\cosh(x_0)))^{0.5})-0.635)$ |
| TaylorGP | $2e^{x_3}-0.24-\tanh(e^{x_3}+\sin((1.598-0.126\sqrt{|x_3|})(e^{x_3}-0.24)))\sqrt{|x_3+e^{x_3}|}$ | $\cos(x_2)+\tanh(x_0)+\frac{4.95\sin(x_2)}{x_2}$ | $-x_1^2+0.802\sqrt{|x_1^2+\sin(x_0)|}$ | $2.037\tanh(0.521\sqrt{|x_0||x_1|})$ |
| NeSymReS | — | $-0.368x_0+x_1\sin(\frac{x_0}{x_1}+0.005x_2)$ | $\frac{0.166x_0+x_1^2}{\cos(2.498(0.137x_1-1)^2)-2.073}$ | $\cos(\frac{\sin(\frac{x_0}{x_1})}{x_0})+0.64$ |
| E2E | $0.002x_2+0.273+0.845e^{1.25x_3}e^{0.008\arctan(-0.065x_0+0.002x_1-14.76)}+0.993\cos(-16.985x_2+(-0.008x_0-37.102)(67.0|0.925+\frac{0.008}{-0.573x_3-0.015}|-59.1)+0.088+\frac{46.3}{0.012-0.001x_0})$ | $(-0.007x_1-0.015)(9.191\sin((0.021-0.906x_2)(21.206|3.697x_0+11.042(-x_2-0.077)^2+0.689|+0.008))-0.121)-0.219\arctan(-0.489x_0-0.094)+0.566$ | $0.003x_0+0.144+(0.002-0.005(-x_1-0.014)^2)/(0.006-0.001(-\sin(7.343x_0+3.499)-0.628)^2)$ | $2.0-0.9/(0.006(2.068x_0-0.4)(3.413x_0+0.229)(2.073x_1+0.038)(3.109x_1+0.206)|+0.5)$ |
| uDSR | $0.001x_0x_1+0.001x_0+0.001x_1^2+0.003x_1x_3+0.002x_1+0.003x_2^2+0.001x_2x_3^2+0.004x_2x_3+0.001x_2+0.153x_3^4+0.568x_3^3+0.503x_3^2+0.613x_3+\sin(x_0+x_1x_2)+1.086$ | $-0.001x_0^3-0.002x_0^2-0.002x_0x_1-0.002x_0x_2+0.242x_0+0.037x_1^2+0.01x_1+0.003x_2^2-0.259x_2^2+3\cos(x_2)-\cos(2x_2)+2.688$ | $0.001x_0^4-0.022x_0^3+0.002x_0^2x_1^2-0.001x_0^2x_1-0.028x_0^2-0.001x_0x_1^3-0.007x_0x_1^2+0.02x_0x_1+0.321x_0+0.002x_1^4+0.005x_1^3-0.933x_1^2-0.051x_1+e^{\sin(6x_0)}-0.234$ | $0.005x_0^4-0.001x_0^3-0.306x_0^2-0.001x_0x_1-0.816x_0+0.008x_1^4-0.18x_1^2+\log(-x_0+e^{2x_0})-\cos(x_1)+0.914$ |
| TPSR | $14.264(0.045e^{x_3}+0.002\cos(7.628x_2-12.078)+1)^2-14.456$ | $1.56(0.05x_0+1)^2-0.88$ | $1.0-0.89(-x_1-0.05)^2$ | $2.015-\frac{0.793}{0.124(0.001-x_0)^2(-x_1-0.005)^2+0.595}$ |
| SeTGAP | $1.002e^{1.2x_3}+0.999\sin(x_0+x_1x_2-6.281)$ | $0.998\cos(0.201x_2^2-18.884)|x_1|+1.0\tanh(0.501x_0)$ | $\frac{1.005x_1^2-0.873}{(-\sin(6.283x_0-6.283)-1.507)}+0.084$ | $-\frac{11.534}{(11.542x_1^4+11.535)}+2.0-\frac{11.534}{(0.045x_0+0.062x_0^2+11.474x_0^4+11.528)}$ |

Table 39: Comparison of predicted equations (E9—E13) with rounded numerical coefficients — Iteration 10

| Method | E9 | E10 | E11 | E12 | E13 |
|---|---|---|---|---|---|
| PySR | $\sqrt{1.148x_1} - 4.723 + \frac{4.49}{\cosh\left(x_0\tanh\left(\frac{2.296}{1.849-0.771\cos(x_0)}\right)\right)}$ | $\sin(x_0 e^{x_1})$ | $2x_0\log(x_1^2)$ | $1.0x_0\sin\left(\frac{1.0}{x_1}\right) + 1.0$ | $\log(3.711\sinh(\sqrt{|\log(|\cosh(0.381x_1)|)|}))\tanh(\sqrt{e^{-2.032\cosh(0.209x_1)}} + 2.45)2.015\sqrt{|x_0|}$ |
| TaylorGP | $\log\left(\sqrt{\left|\frac{x_1}{x_0}\right|}\right) - 1.486\sqrt{|x_0|} + 0.889$ | $\frac{\cos(\sqrt{|x_1|})}{\tanh(\tanh(x_0))}$ | $4x_0\log(|x_1|)$ | $\log(e^{\tanh(\frac{x_0}{x_1})}) + \tanh(\frac{x_0}{x_1}) + \sqrt{|\tanh(x_1)|}$ | $2\log(|x_1|)\sqrt{|x_0|}$ |
| NeSymReS | $\log\left(\frac{0.284|x_1|}{|x_0|}\right)$ | $\sin(x_0 e^{x_1})$ | $x_0\log(x_1^4)$ | $(x_0 + x_1)\sin\left(\frac{1}{x_1}\right)$ | $0.23x_0 + \log(x_1^2)$ |
| E2E | $-0.986\log(26.495(-0.179 - \frac{1}{0.691x_1+7.596})^2(-0.063 - \frac{1}{0.691x_1+0.626})^2(0.031-x_0)^2 + 0.224) - 0.087$ | $-1.0\sin(0.026x_0(1.0 - 36.638e^{1.091x_1})) - 0.001$ | $(-0.214x_0 - 0.001)(6.0\log(1742.4(0.02 - \frac{1}{1.244x_1+0.023})^2 + 0.01) - 50.0)$ | $0.996 - 7.12\sin((0.01 + \frac{19.5}{0.006x_0-9.741x_1-0.305})(0.067x_0 + 0.001))$ | $(2.27 - 0.465(0.086(|(92.8\arctan(0.172x_0 + 20.71) + 29.3)(-0.015x_1 + (0.913 + \frac{61.081}{x_1})(0.056x_0 - 2.916) + 0.736)|+0.001)^{0.5} - 1)^{0.5})(3.87\log(|21.928|0.027x_0 - 1|^{0.5} - 26.9|) + 0.05)$ |
| uDSR | $\log\left(\frac{2x_1+1}{4.0x_0^2+1.0}\right)$ | $\sin(x_0 e^{x_1})$ | $x_0\log(0.368x_1^4) + x_0$ | $x_0\sin(1/x_1) + 1.0$ | $(0.003x_0^3 - 0.048x_0^2 + 0.547x_0 + 0.001x_1 + 0.443)\log(x_1^2)$ |
| TPSR | $-97.127x_1 - 0.029(-x_0 - 0.024)^2 + (-0.013x_0 + x_1 - 1169.375(-0.084x_1 - 0.002\arctan(0.29 - 5.115|0.129x_0 + 0.001|) + 0.118) + 137.562$ | $1.003\sin(0.99667x_0e^{x_1})$ | $(-0.524x_0 - 0.02)(-0.277x_1^2 - 7.971 - \frac{32.033}{-5.252|0.701x_1+0.001|-0.726})$ | $0.026x_0(-0.068 + \frac{1}{-0.634(-x_1-0.964)^2-0.887})(-36.911\arctan(44.002x_1 - 16.916) - 34.003) - 0.075x_0 + 0.978$ | $(-0.015 + \frac{1}{0.539|9.48x_1+0.014|+0.511})(0.417 - 0.44(0.001 - x_1)^2)(1.325 - \frac{2.356}{x_0+1.916})(-1.316x_0 - 22.185)$ |
| SeTGAP | $-1.0\log(10.831x_0^2 + 2.707) + 1.0|\log(21.911x_1 + 10.961)| - 1.399$ | $1.0\sin(1.0x_0e^{x_1})$ | $4.0x_0\log(|x_1|)$ | $1.0x_0\sin\left(\frac{1}{x_1}\right) + 1.0$ | $1.0\sqrt{x_0}\log(x_1^2)$ |

Table 40: Comparison of expressions learned by SeTGAP Under Noisy Conditions

| Problem | $\sigma_a = 0.01$ | $\sigma_a = 0.03$ | $\sigma_a = 0.05$ |
|---|---|---|---|
| E1 | $0.608x_0x_1 + 1.089\sin((2.263x_0 - 1.525)(x_1 - 0.665) - 0.011)$ | $(0.611x_0 + 0.001)(x_1 - 0.043) - 0.885\sin((2.253x_0 - 1.544)(x_1 - 0.721) - 3.214) + 0.006$ | $0.61x_1(x_0 - 0.006) - 0.47\sin((2.381x_0 - 1.171)(x_1 - 0.798) + 8.501) + 0.006$ |
| E2 | $0.063x_0^2 - 0.499x_0 + (3.078\sqrt{0.098x_1 + 1} + 0.062)(\sin(0.199x_2) - 0.002) + 6.481$ | $0.063x_0^2 - 0.499x_0 + (3.222\sqrt{0.099x_1 + 1} - 0.058)(\sin(0.201x_2) - 0.007) + 6.511$ | $(-(0.507x_0 - 6.557)(0.181x_1 - 18.566) - (0.374x_0^2 + 18.893\sin(0.198x_2))\log(x_1 + 19.824))/(0.181x_1 - 18.566)$ |
| E3 | $0.143e^{1.516x_0} + 0.506\sin(2.999x_1 + 1.572) + 0.009$ | $0.148e^{1.505x_0} - 0.498\sin(3.0x_1 - 1.572) + 0.003$ | $0.149e^{1.502x_0} - 0.496\sin(3.0x_1 - 1.581) + 0.003$ |
| E4 | $0.01x_0^4 - 0.02x_0^2x_1 - 0.007x_0^2 + 0.009x_1^2 + 0.011x_2^4 - 0.02x_2^2x_3 - 0.02x_2^2 + 0.008x_3^2 + 0.113$ | $0.01x_0^4 + 0.01x_2^4 - 0.02x_2^2x_3 + 0.01x_3^2 + 0.242\cosh(10.69\sqrt{1 - 0.016x_1} - 8.186) - 1.473$ | $0.01x_1^4 - 0.021x_3^2x_4 + 0.01x_4^4 + 0.011x_4 + 0.01x_4^2 + (0.093x_1^2 + 0.007|x_4 + 1|)\sin(0.248x_2 - 2.981) + 0.124$ |
| E5 | $1.004e^{1.199x_3} + 0.997\sin(1.0x_0 + 1.0x_1x_2) - 0.005$ | $(-(0.015\sin(0.508x_2 + 1.574) - 0.001)(|4.779\sin(1.0x_0 + 3.141) - 6.283| - 6.283) + (6.226\sin(0.508x_2 + 1.574) - 6.283)(0.002e^{1.202|x_3+5.282|} + 0.145\sin(1.0|x_1 - 6.283| - 1.847) + 0.067\sin(1.0|x_1 + 6.202| - 4.425)|2.649\sin(1.0x_0 + 3.24) - 2.097|+0.005))/(6.226\sin(0.508x_2 + 1.574) - 6.283)$ | $0.996e^{1.202x_3} + 1.0\sin(1.0x_0 + 1.0x_1x_2) + 0.004$ |
| E6 | $1.0\cos(0.2x_2^2 + 0.008)|x_1| + 1.0\tanh(0.5x_0) + 0.005$ | $\cos(0.199x_2^2 + 0.028)(1.001|x_1| - 0.001) + 1.002\tanh(0.486x_0) + 0.02$ | $1.0\cos(0.2x_2^2)|x_1| + 0.997\tanh(0.502x_0)$ |
| E7 | $\frac{1.001x_1^2 - 0.9914}{\sin(6.283x_0 + 3.15) - 1.501} + 0.011$ | $\frac{-1.001x_1^2 + 1.006}{\sin(6.283x_0) + 1.501} + 0.007$ | $-0.982\frac{x_1^2 - 0.993}{\sin(6.283x_0 + 3.139) - 1.492} + 0.019$ |
| E8 | $2.0 - 11.356/(11.313x_1^4 - 0.116x_1^3 + 0.047x_1^2 + 0.078x_1 + 11.352) - \frac{10.636}{10.804x_0^4 - 0.113x_0^2 + 10.646}$ | $2.0 - \frac{15.531}{15.414x_1^4 - 0.051x_1^3 + 0.165x_1^2 + 0.018x_1 + 15.517} - \frac{12.561}{12.523x_0^4 - 0.085x_0^3 + 0.062x_0^2 + 0.061x_0 + 12.553}$ | $2.0 - \frac{12.837}{12.956x_0^4 + 0.007x_0^3 - 0.125x_0^2 + 12.861} - \frac{16.19}{16.217x_1^4 - 0.007x_1^3 + 16.202}$ |
| E9 | $-1.0\log(4.884x_0^2 + 1.22) + 1.0|\log(9.415x_1 + 4.709)| - 1.35$ | $-1.0\log(17.004x_0^2 + 4.251) + 0.999|\log(10.158x_1 + 5.072)| - 0.176$ | $5.965\sqrt{0.625\log(9.36x_1 + 6.35) + 1} - 0.999\log(11.259x_0^2 + 2.817) - 7.713$ |
| E10 | $1.0\sin(1.0x_0e^{x_1})$ | | $1.0\sin(1.0x_0e^{1.0x_1})$ |
| E11 | $2.0x_0\log(x_1^2)$ | $(4.001x_0 - 0.004)\log(|x_1|) + 0.003$ | $(4.001x_0 - 0.008)\log(|x_1|) + 0.006$ |
| E12 | $1.0x_0\sin(1/x_1) + 1.0$ | $1.0x_0\sin(1/x_1) + 1.0$ | $1.0x_0\sin(1/x_1) + 1.0$ |
| E13 | $(1.001\sqrt{x_0} - 0.001)\log(x_1^2)$ | $(0.993\sqrt{x_0} - 0.002)\log(x_1^2) + 0.029$ | $(1.001\sqrt{x_0} - 0.003)\log(x_1^2)$ |

## G   SeTGAP Under Noisy Conditions

This appendix presents the expressions learned by SeTGAP under noisy conditions, shown in Table 40, where shaded cells indicate an incorrectly identified functional form. While the expressions obtained in the noiseless setting are provided in Tables 2–4, here we focus on analyzing the impact of noise on the recovered

Table 41: SRBench++ synthetic equations

| Eq. | Underlying equation | Domain range |
|---|---|---|
| F1 | $0.4x_0x_1 - 1.5x_0 + 2.5x_1 + 1$ | $[-5,5]^2$ |
| F2 | $0.4x_0x_1 - 1.5x_0 + 2.5x_1 + 1 + + \log(30x_2^2)$ | $[-5,5]^3$ |
| F3 | $\frac{0.4x_0x_1 - 1.5x_0 + 2.5x_1 + 1}{0.2(x_0^2 + x_1^2) + 1}$ | $[-20,20]^2$ |
| F4 | $\frac{0.4x_0x_1 - 1.5x_0 + 2.5x_1 + 1 + 5.5\sin(x_0 + x_1)}{0.2(x_0^2 + x_1^2) + 1}$ | $[-20,20]^2$ |

Table 42: Expressions learned by SeTGAP on SRBench++ functions

| It. | F1 | F2 | F3 | F4 |
|---|---|---|---|---|
| 1 | $0.403x_0x_1 - 1.504x_0 + 2.5x_1 + 1.0$ | $0.401x_0x_1 - 1.5x_0 + 2.5x_1 + 0.994\log(x_2^2) + 4.411$ | $\frac{(16.43x_0 + 101.498)(x_1 - 3.751) + 420.837}{8.23x_0^2 + 8.145x_1^2 + 40.317}$ | $\frac{37.563(-0.232x_0 - 1.467)(x_1 - 3.255) - 192.406}{-0.519x_0 - 4.32x_1^2 + (0.354x_0^2 + 0.224)(-0.127x_1 - 13.04) - 10.342} + 0.023$ |
| 2 | $0.403x_0x_1 - 1.501x_0 + 2.5x_1 + 1.0$ | $0.4x_0x_1 - 1.5x_0 + 2.501x_1 + 0.994\log(x_2^2) + 4.415$ | $\frac{(5.225x_0 + 32.182)(x_1 - 3.679) + 131.061}{2.599x_0^2 + 2.591x_1^2 + 12.653}$ | $\frac{45.613 \cdot (0.73x_0 + 4.5)(x_1 - 3.22) + 713.964}{16.491x_1^2 + (0.339x_0^2 + 0.744)(0.556x_1 + 51.947) + 11.18} + 0.027$ |
| 3 | $0.402x_0x_1 - 1.5x_0 + 2.5x_1 + 1.0$ | $0.4x_0x_1 - 1.5x_0 + 2.5x_1 + 1.999\log(|x_2|) + 4.402$ | $\frac{(3.518x_0 + 22.14)(x_1 - 3.804) + 92.972}{1.755x_0^2 + 1.771x_1^2 + 8.998}$ | $0.019 - \frac{14.122(-2.3x_0 - 14.317)(x_1 - 3.208) - 716.019}{16.077x_1^2 + (0.578x_0^2 + 1.851)(0.251x_1 + 29.437)}$ |
| 4 | $0.4x_0x_1 - 1.5x_0 + 2.5x_1 + 1.0$ | $0.4x_0x_1 - 1.5x_0 + 2.5x_1 + 1.006\log(x_2^2) + 4.39$ | $\frac{-20.847(-0.437x_0 - 2.699)(x_1 - 3.68) + 229.698}{4.523x_0^2 + 4.524x_1^2 + 22.284}$ | $0.25 - \frac{50.0}{-3.823x_0^2 - 4.555x_1^2 + (8.072x_0 + 66.932)(x_1 - 6.951) - 327.87}$ |
| 5 | $0.4x_0x_1 - 1.5x_0 + 2.5x_1 + 1.0$ | $0.4x_0x_1 - 1.5x_0 + 2.5x_1 + 0.995\log(x_2^2) + 4.411$ | $\frac{8.168(0.507x_0 + 3.191)(x_1 - 3.779) + 109.02}{2.076x_0^2 + 2.078x_1^2 + 10.51}$ | $0.256 - \frac{50.0}{-3.411x_0^2 - 4.472x_1^2 + (7.662x_0 + 64.337)(x_1 - 6.769) - 332.132}$ |
| 6 | $0.4x_0x_1 - 1.5x_0 + 2.5x_1 + 1.0$ | $0.4x_0x_1 - 1.5x_0 + 2.5x_1 + 1.987\log(|x_2|) + 4.411$ | $\frac{4.819(2.48x_0 + 15.215)(x_1 - 3.665) + 298.03}{5.929x_0^2 + 5.917x_1^2 + 28.938}$ | $0.254 - \frac{50.0}{-3.174x_0^2 - 3.987x_1^2 + (7.231x_0 + 60.398)(x_1 - 6.93) - 355.795}$ |
| 7 | $0.4x_0x_1 - 1.5x_0 + 2.5x_1 + 1.0$ | $0.4x_0x_1 - 1.5x_0 + 2.5x_1 + 1.004\log(x_2^2) + 4.395$ | $\frac{-6.653(0.808x_0 + 5.044)(x_1 - 3.708) - 137.9}{-2.675x_0^2 - 2.687x_1^2 - 13.337}$ | $0.26 + \frac{50.0}{4.443x_0^2 + 5.439x_1^2 - (9.653x_0 + 79.204)(x_1 - 6.977) + 362.483}$ |
| 8 | $0.4x_0x_1 - 1.5x_0 + 2.5x_1 + 1.0$ | $0.4x_0x_1 - 1.5x_0 + 2.501x_1 + 0.993\log(x_2^2) + 4.413$ | $\frac{29.656(0.745x_0 + 4.661)(x_1 - 3.756) + 574.829}{11.069x_0^2 + 11.043x_1^2 + 55.471}$ | $0.236 + \frac{50.0}{2.314x_0^2 + 2.99x_1^2 - (5.12x_0 + 43.79)(x_1 - 6.84) + 277.643}$ |
| 9 | $0.4x_0x_1 - 1.5x_0 + 2.5x_1 + 1.0$ | $0.4x_0x_1 - 1.501x_0 + 2.5x_1 + 1.975\log(|x_2|) + 4.424$ | $\frac{6.325(2.199x_0 + 13.381)(x_1 - 3.567) + 335.564}{6.827x_0^2 + 6.865x_1^2 + 32.789}$ | $0.274 - \frac{50.0}{-6.504x_0^2 - 8.41x_1^2 + (14.504x_0 + 119.585)(x_1 - 6.792) - 474.742}$ |
| 10 | $0.4x_0x_1 - 1.5x_0 + 2.5x_1 + 1.0$ | $0.4x_0x_1 - 1.5x_0 + 2.499x_1 + 0.995\log(x_2^2) + 4.41$ | $\frac{34.748(-0.581x_0 - 3.638)(x_1 - 3.715) - 519.981}{-10.04x_0^2 - 10.11x_1^2 - 50.332}$ | $\frac{42.93(-0.59x_0 - 3.686)(x_1 - 3.267) - 562.393}{-12.503x_1^2 + (0.977x_0^2 + 2.913)(-0.118x_1 - 13.727) + 1.471} + 0.021$ |

functional forms. These expressions correspond to the results reported in Table 6, where different noise levels were introduced to assess the robustness of SeTGAP.

## H    Experiments on SRBench++ Functions

In this section, we evaluate SeTGAP on synthetic functions from the SRBench++ benchmark (de Franca et al., 2025). Specifically, we focus on the four functions F1–F4 from the "Rediscovery of Exact Expressions" task, presented in Table 41. Here, SeTGAP follows the same configurations described in Sec. 4.

Table 42 shows the expressions learned for each problem, where shaded cells indicate an incorrectly identified functional form. The evaluation was repeated ten times, each on a newly generated dataset with a different random seed. The results indicate that SeTGAP identifies the correct functional form of problems F1–F3 across all iterations, while it fails to recover the functional form of problem F4. This outcome is expected, as the corresponding univariate skeleton with respect to variable $x_0$ is $\mathbf{e}(x_0) = (cx_0 + \sin(x_0 + c))/(cx_0^2 + c)$,

Table 43: Extrapolation MSE on SRBench++ functions

| F1 | F2 | F3 | F4 |
|---|---|---|---|
| 6.480e-03 ± 1.055e-02 | 5.949e-04 ± 5.012e-04 | 4.759e-05 ± 7.400e-05 | 6.172e-01 ± 4.915e-01 |

which involves eight operators, including three unary operators (`sin`, `sqr`, and `inv`). Such complexity exceeds the representational capacity of our approach, since the Multi-Set Transformer used in the experiments was trained only on expressions with up to seven operators, at most two unary operators, and up to one nested unary operator [Ref. 1][2]. Finally, we assessed the extrapolation capability of the learned expressions by evaluating them on an extended domain. As in Sec.4, the extrapolation range was defined as twice the size of the original domain, excluding the original interval. Each extrapolation set consisted of 10,000 points sampled within this range. Table 43 reports the mean and standard deviation of the extrapolation MSE.

