# OpenReview forum: "Decomposable Neuro Symbolic Regression"
_TMLR — Rejected by TMLR_

### Review · Reviewer_YyeS · 2025-07-14

**Summary Of Contributions:**

The authors consider the symbolic regression problem. They extend their earlier work that uses a transformer to generate univariate 'skeleton' (templated) functions. Their question is how to combine these univariate skeletons. For this, they use the following process: It creates an initial population of multivariate functions by merging univariate functions with some random elements. Then, it uses genetic programming to combine and mutate the different merges. Finally, it finds values for the constants in the resulting skeleton. The method finds very similar forms to that in 13 synthetic tasks.

**Audience:**

Yes

**Claims And Evidence:**

No

**Requested Changes:**

Critical:
- After section 3.1, please give an overview of the method to help follow the story
- Section 3.2: Please highlight more clearly where existing work (Ref 1.) ends and your method begins.
- Section 3.2: "we discard identitcal or mathematically equivalent skeletons": How is this done? This is not easy!
- Proposition 1: This proposition is missing that it assumes the binary operators are +, -, *, or /. The proof requires this. That highly limits the statement, as it states it works on _any *finite*_ composition involving binary operators. I doubt this is the case for operators that involve nonlinearities/transcedental functions. (In addition, please state what are the operators used in the SR algorithm)
- Algorithm 2 is very complex and section 3.3.1 explaining it is hard to follow. I found it hard to relate this algorithm to Proposition 1, which I assume is the motivation. Please add an example for each of the conditions in the algorithm, and relate better to the proposition.
- Please provide more details on the GA and GP algorithms used. These seem to be custom (eg, what's the mutation/crossover operator that respects the structure of the skeletons?)
- Please add your previous work [Ref 1.] to the results as an ablation to show what the combination contributes.

Minor:
- Section 3.2: "fixed to random values": Please highlight the underlying distribution
- Bottom of page 5: $c_q$ should be $c_2$ I think.
- Algorithm 2: How is isAllSymbols defined?
- Algorithm 2, line 20: I don't understand what .func does and how it's possible to go in recursion on the arguments of the skeletons. I thought ex1 and ex2 are a single skeleton each, but .args would return multiple (sub)-skeletons.

Questions:
- Isn't the choice of test set for univariate parameter fitting quite influential? Isn't that somewhat limiting if it is only on a single set of random values for the other params?
- The choice of Pearson correlation for testing rather than MSE was a bit confusing to me. Do you have an ablation for this?
- Why use genetic algorithms for parameter fitting rather than gradient descent? (This might be a naive question - I'm not from this area).
- Section 3.3.2: Am I correct that each candidate is evaluated with random coefficients? Isn't this cause of a lot of variance in evaluation?
- How are the 13 equations for the experiments selected?

**Strengths And Weaknesses:**

Strengths:
- The paper is pretty well written, especially the introduction and related work for someone who is not familiar with SotA in symbolic regression
- The method impressively finds the resulting forms

Weaknesses
- The method is quite complex, with many moving parts while many details are missing
- The method extends the author's previous work, however this work is not available and the paper is not self-contained without having it accessible.
- There are no ablation studies to highlight the necessity of all parts in the method.
- There are no experiments on tasks where the method fails, and all tasks have at most 4 variables and are synthetic. This raises questions about transfer to real-world, cherry picking, and scalability.

---

> ### Author Response · Authors · 2025-08-30
> **Response to Reviewer YyeS**
>
> Thank you for your comments. We address your questions and comments below:
>
> Q1:  We do not use a single set of random values for the variables not under analysis, but multiple sets. Our MSSP approach tackles this problem. For ex., if we have a 3-variable system and want to analyze the relationship between x1 and the response, y, multiple sets (Ns=10) will be sampled in which variable x1 is allowed to vary while x2 and x3 are fixed to random values (which will be different for each set). The task of the MSSP solver is then to find which functional form (skeleton) explains the common behavior among these generated sets.
>
> Q2: We haven’t done an ablation study for this. The reason is that, at this stage, we’re only interested in evaluating the identified functional form. E.g., suppose we’re analyzing x1 and its underlying functional relationship with respect to y is sinusoidal. Then, using our MSSP approach, we have generated two candidate skeletons: e1 = c1 cos(c2 x_1 + c3) + c4, and e2 = c1 sin(c2 x_1^2 + c3) + c4. Using a GA, if we optimize for Pearson correlation, we just need to fit the coefficients c2 and c3. If the fitted skeletons match the behavior of the data, the correlation will be high, favoring e1. Optimizing MSE would require fitting all coefficients and much larger GA populations and time.
>
> Q3: We use GAs because the search is typically non-differentiable, non-convex and may include discontinuities (abs, nested trig). Gradient-based or quasi-Newton methods assume smoother objectives and can struggle with such expressions.
>
> Q4: All candidate skeletons (“candSks”) are evaluated on the same test set $\tilde{\mathbf{D}}_{S'}^{(test)}$ (Alg. 3, line 1).
>
> Q5: Equations E1, E3, E4, E7, E8, and E9 have been in previous SR works (refs in the paper). E10-E13 have been derived from the paper “The Metric is the Message,” and show increased complexity compared to previous benchmarks, such as the Nguyen benchmark, which consists of simple polynomial expressions. In previous work, we included equations E2, E5, and E6 to increase the number of non-separable problems, which is where the compared Neural SR approaches tend to fail.
> We also tested SeTGAP on functions from the SRBench++ benchmark (App. H). SeTGAP fails on problem F4, given that the univariate skeleton of its underlying function with respect to x0 is $(c x_0 + \sin(x_0 + c)) / (c x_0^2 + c)$, which requires 8 operators, including 3 unary operators (sin, sqr, and inv). The complexity of this skeleton could not be reproduced by our approach because our Multi-Set Transformer was trained using expressions with up to 7 operators and 2 unary operators.
>
> Comments:
>
> C1: We extended the final paragraph of the Introduction. Fig. 1 and that paragraph now overview the approach.
>
> C2: We added Sec. 3.2.1 and a new App. A to clarify this. Previously, we focused on formulating the Multi-Set Transformer (i.e., the solid green box in Step 1 of Figure 1) for univariate skeleton prediction. Here, we present SeTGAP, which leverages the Multi-Set Transformer to generate a set of univariate skeletons per variable and also presents a method for merging them into multivariate expressions.
>
> C3: We only identify identical skeletons (using SymPy’s method “equal”) and some trivial trig-equivalences, as explained in the new App. B. This reduces obvious redundancy and cost but does not affect final performance. For example, this happened in Tables 15 and 18, where SeTGAP produced equivalent expressions for problem E13: \sqrt{x_0}log(x_1^2) and 2\sqrt{x_0}log(|x_1|).
>
> C4: Thanks for noticing the oversight. We updated the Proposition to reflect this. We’re also adding a new Appendix A specifying the list of operators used by the Multi-Set Transformer.
>
> C5: Algorithm 2 was clarified and its link to Proposition 1 made explicit.
>
> C6: Our “constrained GP” approach is custom and is detailed in Alg. 2 (identifies admissible combinations) and 3 (evolves combination candidates in parallel and selects the best). The mutation and crossover operations are described in Sec 3.3.2.
>
> C7: An ablation removing the Multi-Set Transformer is not possible since it produces the univariate skeletons; it cannot be omitted. Currently, the Multi-Set Transformer is the only method that produces skeletons to explain the functional form between each independent variable and the system’s response.
> However, we added tuning experiments in App. E to describe the effect that different hyperparameters have, including those that interact with the Multi-Set Transformer.
>
> C8: “.func” is the head operator node and “.args” its arguments. E.g., (x1^2 + log(x2)).func = “Sum” and (x1^2 + log(x2)).args = [x1^2, log(x2)], or (tanh(x1^2 + log(x2))).func = “tanh” and (tanh(x1^2 + log(x2))).func = [x1^2 + log(x2)]. If two skeletons are compatible (i.e., they share the same head operator node), their inner arguments can be merged. We provided an example in Appendix D.
>
> All changes are highlighted in yellow in the paper.

---

### Review · Reviewer_i47B · 2025-07-21

**Summary Of Contributions:**

This paper contributes a new symbolic regression method to extract governing equations that generate observed data. The core method uses a pretrained opaque regression model to generate univariate symbolic skeletons, feeds these skeletons into a genetic algorithm for selection and combination, then optimizes any remaining constants. Experiments on synthetic benchmark problems with ground-truth equations shows that the proposed method outperforms certain baselines at recovering the correct functional form and also has better numerical accuracy in extrapolation tests.

**Audience:**

Yes

**Claims And Evidence:**

Yes

**Requested Changes:**

* Mandatory: explicitly list out the SOTA method for each of the benchmark problems and compare with them.
* In Prop 1, the $c_{i,j}$ has a free index $j$ that needs explanation.

**Strengths And Weaknesses:**

Strengths:
* The core intuition of decomposing the full task into univariate skeleton prediction subtasks is well-motivated.
* Overall method and algorithm is rigorously explained in detail.

Weaknesses
* As presented, the method appears to have a high number of design choices and it is not clear which are crucial ones that must be tuned well for the overall approach to work.
* The neural SR methods used as baselines was limited to those with publicly available trained models, so it is not clear that the method was compared with SOTA. The explanation that some published methods would require the authors to conduct computationally-prohibitive training is not an excuse to exclude those methods, because the authors could have just applied their method to the benchmark problems used in those previously published works and directly compare with previously published results without running the training for those baselines.
* The method relies on using some pretrained regression model to produce the artifical dataset when varying $x_v$ for each univariate subtask. Not clear how sensitive the method is to the pretrained model. In the experiment section, the pretrained regression models had different architectures (i.e., different number of layers) for each problem. Why, and how would anyone know how to choose that in practice?

Not a weakness strictly, but an unusual behavior that I have not seen in many years of reviewing in ML: when describing some relevant prior work, the authors intentionally did not provide citations (claiming to abide by the double-blind review). So it is near impossible for reviewers to understand that part unless one already was familiar with the hidden prior work. Better to just cite it and simply just rephrase the writing to exclude claiming that this work was by the authors.

---

> ### Author Response · Authors · 2025-08-30
> **Response to Reviewer i47B**
>
> Thank you for your comments. We address your questions and comments below:
>
> **W1: _As presented, the method appears to have a high number of design choices..._**
>
> W1: Thanks for this comment. We added tuning experiments in Appendix E to describe the effect that different hyperparameters have, including those that interact directly with the Multi-Set Transformer, such as n_B and n_cand.
>
> **W2: _The neural SR methods used as baselines was limited to those with publicly available trained models..._**
>
> W2: We stated that “The comparison is limited to neural SR methods with publicly available models.” This statement does not imply that some methods were left out; instead, it excludes one method in particular,  Bertschinger et al. (GECCO 2024), which was tested on only five univariate problems. Their set of problems is not a benchmark used by the other methods used for comparison in our work and, what is more, escapes the scope of this work, as we focus on multivariate SR. This was clarified in the paper. In addition, we added the unified deep SR (uDSR) as another compared method.
>
> Please note that not all the compared methods have been tested on the same set of functions. Therefore, our synthetic datasets ensure fair comparison. They allow us to inspect the mathematical expressions obtained by the different methods (not only their metrics) and, importantly, analyze their consistency after multiple iterations (Appendix F), which is not done in any major benchmark. The main objective of previous benchmarks is to compare the predictive power of different SR methods. Rather than achieving SOTA performance, our work aims to address the limitations of existing methods that struggle to correctly identify the functional relationships of all variables in a multivariate system.
>
> **W3: _The method relies on using some pretrained regression model ..._**
>
> W3: We would like to clarify that $\hat{f}$ is not a pre-trained model, but rather a generic opaque regression model trained on each problem to approximate the system’s input–output behavior. The SeTGAP framework does not rely on any particular design of $\hat{f}$; instead, it treats $\hat{f}$ as an input and aims to distill its learned function into interpretable mathematical expressions $\tilde{f}$. In practice, $\hat{f}$ may take the form of a neural network with any architecture, a random forest, or other regression models.
>
> In our experiments, NNs were selected for $\hat{f}$, and their architectures were tuned in the standard way to minimize MSE for each problem. Different problems present different levels of functional complexity, which naturally require architectures of varying depth. This tuning step is not part of the contribution of SeTGAP, but merely ensures that the opaque model reasonably captures the problem’s dynamics. Once trained, SeTGAP is applied to extract symbolic models that interpret $\hat{f}$. Note that, if $\hat{f}$ contains modeling errors (e.g., due to noise or imperfect fitting), SeTGAP’s output reflects this mismatch, as shown in Table 6. Thus, the sensitivity to the opaque model is not a limitation of SeTGAP itself but an expected property of any interpretability method that seeks to explain a given model.
>
> **C1:  _It is near impossible for reviewers to understand the prior work..._**
>
> C1: We appreciate the reviewer’s concern. We added a new Section 3.2.1, to clarify the difference with respect to the previous work, and a new Appendix A. These sections were written to review our prior work and provide the necessary background for understanding the Multi-Set Transformer introduced in Ref. 1.
>
> **C2:  _In Prop 1, the_** $c_{i,j}$ **_has a free index  that needs explanation..._**
>
> C2: Thanks for noticing this oversight. We fixed Prop 1.
>
> All changes are highlighted in yellow in the paper.

---

> > ### Comment · Reviewer_i47B · 2025-09-15
> > **Acknowledgement of rebuttal**
> >
> > I appreciate the additional hyperparameter experiments and the improved explanation of prior work and the prior problem-specific regression models.

---

### Review · Reviewer_CFao · 2025-08-25

**Summary Of Contributions:**

This paper presents SeTGAP, a symbolic regression method that combines Multi-Set Transformers with genetic algorithms and genetic programming. The approach first generates univariate symbolic skeletons for each variable independently, then merges them incrementally using evolutionary techniques to produce multivariate expressions. The authors claim their method consistently recovers correct functional forms across synthetic problems, outperforming existing GP-based and neural SR methods like PySR, E2E, TPSR, and NeSymReS.

**Audience:**

Yes

**Claims And Evidence:**

No

**Requested Changes:**

I suggest authors to consider at least adding these to paper:
- Provide discussion and experimental results on the comparison with uDSR baseline, which appears to share significant overlap with the proposed method. This should include
- Expand evaluation datasets to include SRBench problems to test on higher-dimensional problems and enable meaningful comparison with state-of-the-art baselines already included in this standard benchmark
- Ablation study on the proposed components of methodology to validate each component's contribution

**Strengths And Weaknesses:**

## Strengths:
- The decomposable approach of analyzing variables separately before merging is interesting and shows promise for maintaining interpretability during the search process

## Weaknesses:
- I'm concerned on the novelty of the method compared to similar method uDSR. The framework focus on the main contribution of combining neural transformer models with the power of genetic algorithms and GP for search, however hasn't compared with the known method uDSR which has very similar structure and also builds on the idea of combining neural models with GP for the same purpose.
- I also think the evaluation testbed is very limited. The datasets used for comparison experiments are uncommon for SR and mostly limited to 1D problems. I'd recommend using SRBench datasets instead for more standard evaluation compared to baselines.
- The proposed framework consist of multiple components which authors claim to be important, however, there's no ablation study for components in the proposed approach to validate impact of each of them on performance. Ablation experiments are needed to understand why each of the component is necessary and helpful.
- As evaluations are only conducted on very low-dimensional datasets, the scalability of the proposed method not higher-dimensional problems is not clear.

---

> ### Author Response · Authors · 2025-08-30
> **Response to Reviewer CFao**
>
> Thank you for your comments. We address your questions and comments below:
>
> **W1: Novelty compared to uDSR...**
>
> W1: We have included uDSR in the Related Work section and as a baseline in the Experimental section. We do not believe there is substantial overlap between uDSR and our approach. uDSR is a hybrid method that first attempts to divide the problem into sub-problems using AI-Feynman. For each sub-problem, it uses an RNN-based optimization, where, at each step, a neural SR model is used to generate candidate mathematical expressions. These candidates are then evolved and refined using GP, the resulting expressions are then fed back into the RNN controller. Notably, all components process all available variables jointly.
>
> In contrast, our approach is decomposable. SeTGAP employs a Multi-Set Transformer to generate multiple univariate skeletons that capture the relationships between individual system variables and the response. These skeletons are then progressively merged through a constrained GP (i.e., it explicitly maintains the identified skeleton structures throughout the process) in a cascade process until a complete multivariate expression is obtained. Thus, although both uDSR and SeTGAP leverage transformer architectures and evolutionary techniques, they fulfill fundamentally different roles within their respective frameworks.
>
> Tables 2-4 and App. F show that uDSR tends to produce the largest expressions. This occurs because when the skeletons generated by their neural SR model are inaccurate, they are evolved with GP and augmented with additional nodes (typically polynomials) to better fit the data.
>
>
> **W2: Evaluation testbed**
>
> W2: First, we would like to correct the reviewer’s statement that “The datasets used for comparison experiments are mostly limited to 1D problems.” Please note that none of the tested problems is one-dimensional.
>
> The SR problems were chosen to cover a range of functional forms and difficulties. E1, E3, E4, E7, E8, and E9 correspond to expressions used in prior SR studies (refs. in the paper), while E10–E13 were adapted to the multivariate setting from the suite proposed by Bertschinger et al. (2023). We also included E2, E5, and E6 [Ref. 1] to increase the proportion of non-separable problems, a class where neural SR approaches have been observed to struggle. All equations were evaluated over extended input ranges rather than the narrow domains commonly used in earlier works, thereby increasing problem difficulty. This set of equations shows various structures and increasing complexity compared to previous benchmarks, such as the Nguyen benchmark and most of the AIFeynman equations, which consist of simple polynomial expressions.
>
> The main objective of previous benchmarks is to compare the predictive power of different SR methods. Rather than achieving SOTA performance, our work aims to address the limitations of existing methods that struggle to correctly identify the functional relationships of all variables in a multivariate system. Therefore, our evaluation allows us to inspect the expressions obtained by the different methods (not only their metrics) and, importantly, analyze their consistency after multiple iterations (App. F).
>
> Nevertheless, we tested SeTGAP on functions from SRBench++ (App. H). Due to the time constraints, we only reported SeTGAP’s results to demonstrate its behavior. As expected, SeTGAP fails on problem F4, given that the univariate skeleton of its underlying function with respect to $x_0$ is $(c x_0 + \sin(x_0 + c)) / (c x_0^2 + c)$, which requires 8 operators, including 3 unary operators (sin, sqr, and inv). The complexity of this skeleton could not be reproduced by our approach because the Multi-Set Transformer employed for the experiments was trained using expressions with up to 7 operators and 2 unary operators.
>
>
> **W3: Multiple components.**
>
> W3: We added tuning experiments in App. E to describe the effect that different hyperparameters have.
>
> **W4: Low-dimensional datasets....**
>
> W4: We would like to clarify that the dimensionality of the datasets used in our evaluation is consistent with what is commonly addressed in SR works (especially in neural SR). E.g., the “Rediscovery of Exact Expression” task in SRBench++ includes problems with up to 3 variables. Likewise, the compared methods NeSymRes and E2E are restricted to problems with at most 3 and 5 variables, respectively. Our framework does not impose such hard constraints on dimensionality.
>
> That said, our evolutionary merging process, while effective in consistently recovering ground-truth functions in low-dimensional settings, becomes computationally infeasible as dimensionality grows. To address this, we plan to investigate faster and more scalable merging strategies, grounded in a formal theoretical framework, to determine when correctly inferred skeletons can be reliably composed into coherent multivariate expressions.
>
> All changes are highlighted in yellow in the paper.

---

### Decision · Action_Editor_yeEu · 2025-11-05

**Recommendation:** Reject

**Additional Comments:**

The proposed approach appears to be solid and the authors are clearly experts. The main issue is the empirical evaluation, which appears undersized and underscoped compared to analogous efforts/publications.

**Audience:**

Yes

**Audience Explanation:**

Symbolic regression is a relatively niche but important problem. Recent works have been picked up by major publication venues, highlighting how this research area enjoys sufficient interest.

**Claims And Evidence:**

No

**Claims Explanation:**

The paper introduces a symbolic regression framework for discovering/recovering multivariate expressions that exploits a multi-set transformer paired with a genetic optimizer. This was evaluated on 13 (17?) multivariate equations.

Reviewers were generally skeptical of the generality of the evaluation setup. Specifically, it is unclear whether the reported results are sufficient to empirically showcase the generalization ability of the method. The authors claim the equations they chose are representative, which may well be the case. But it seems that more are available in the literature (or could be easily created) and more are use to test symbolic regression models. For instance, Udrescu and Tegmark have tested about 50 equations, which are all available. Another option would be to test recovery of random equations (in addition to well-known formulas). It is unclear why this was not done.

As highlighted by one reviewer, ablation experiments are also missing.

I am confident that improved evaluation would benefit the paper substantially.

**Resubmission Of Major Revision:**

The authors may consider submitting a major revision at a later time.